# Insights into architecture, growth dynamics, and biomineralization from pulsed Sr-labelled *Katelysia rhytiphora* shells (Mollusca, Bivalvia)

Laura M. Otter[1], Oluwatoosin B. A. Agbaje[1], Matt R. Kilburn[2], Christoph Lenz[3,4], Hadrien Henry[1,3], Patrick Trimby[5], Peter Hoppe[6], Dorrit E. Jacob[1,3]

[1]Department of Earth & Planetary Science, Macquarie University, Sydney, NSW 2109, Australia
[2]Centre for Microscopy Characterisation and Analysis, University of Western Australia, Perth, WA 6009, Australia
[3]Australian Research Council Centre of Excellence for Core to Crust Fluid System (CCFS) / GEMOC
[4]Institute of Mineralogy and Crystallography, University of Vienna, Althanstr. 14, 1090 Vienna, Austria
[5]Oxford Instruments NanoAnalysis, High Wycombe, HP12 3SE, United Kingdom
[6]Particle Chemistry Department, Max Planck Institute for Chemistry, Hahn-Meitner-Weg 1, 55128 Mainz, Germany

*Correspondence*: Laura M. Otter (laura.otter@mq.edu.au)

**Abstract.** The intertidal bivalve *Katelysia rhytiphora*, endemic to south Australia and Tasmania, is used here for pulsed Sr-labelling experiments in aquaculture experiments to visualize shell growth at the micro- to nano-scale. The ventral margin area of the outer shell layer composed of (i) an outermost outer shell layer (oOSL) with compound composite prismatic architecture with three hierarchical orders of prisms and an (ii) innermost outer shell layer (iOSL) with crossed-acicular architecture consisting of intersecting lamellae bundles. All structural orders in both layers are enveloped by an organic sheath and the smallest mineralized units are nanogranules. Electron Backscatter Diffraction reveals a strong preferred orientation of the aragonite c-axes perpendicular to the growth layers, while the a- and b-axis are scattered within a plane normal to the local growth direction and >46 % twin grain boundaries are detected. The Young's modulus shows a girdle-like maximum of elastically stiffer orientations for the shell following the inner shell surface.

The bivalves were subjected for 6 days twice to seawater with an increased Sr concentration of 18x mean ocean water by dissolving 144 $\mu g \cdot g^{-1}$ Sr (159.88 Sr/Ca mmol/mol) in seawater. The pulse labelling intervals in the shell are 17x (oOSL) and 12x (iOSL) enriched in Sr relative to the Sr spiked seawater. All architectural units in the shell are transected by the Sr label, demonstrating shell growth to progress homogeneously instead of forming one individual architectural unit after the other. $D_{Sr/Ca}$ for labelled and unlabelled shell are similar to shell portions formed in the wild (0.12 to 0.15). All $D_{Sr/Ca}$ are lower than values for equilibrium partitioning of Sr in synthetic aragonite.

## 1 Introduction

The shells of bivalves are bio-composites with a complex, hierarchical 3D arrangement of crystalline calcium carbonate (aragonite and/or calcite), intimately conjoined by organic macromolecules that control nucleation and growth of the mineral entity across all length scales (Weiner and Traub, 1980; Rodriguez-Navarro et al., 2012; Simkiss, 1965; Addadi et al., 2006;

Cusack et al., 2008; Weiner et al., 1984). This arrangement significantly enhances the physical and mechanical properties of the shell and explains its high mechanical strength and fracture resistance (Currey and Kohn, 1976; Jackson et al., 1988; Kamat et al., 2000).

Trace elements incorporated in the carbonate phase of shells are used to monitor and reconstruct (paleo)environmental parameters, e.g. water salinity (Klein et al., 1996b), temperature (Zhao et al., 2017a; Klein et al., 1996a; Schöne et al., 2011), and pH (Zhao et al., 2017b). While the incorporation mechanisms of trace elements in mollusc shells are not yet fully understood, we do know that the incorporation of some trace elements, such as strontium, are influenced by local growth rates between different growth axes, shell curvature along the same axis, and physiological effects (Urey et al., 1951; Gillikin et al., 2005; Carré et al., 2005; Gillikin et al., 2008; Foster et al., 2009). Organic carboxyl-groups play a critical role for the incorporation of Mg into the shell (Stephenson et al., 2008; Wang et al., 2009; Shirai et al., 2012), but direct evidence for a similar role of organic molecules in the incorporations of other trace elements is lacking. Indeed, different trace elements (e.g. Mg, K, Ca, Sr) show distinctly different interaction with organic molecules and influence on mineral growth (Sand et al., 2017) showing that generalisation for the role of trace elements in biomineralisation are not straightforward.

A critical step forward in our understanding of how trace elements are incorporated into the growing biomineral is to gain better insight across all spatial scales into how different shell architectures are formed. Traditionally, studies on shell formation have been focusing on the nacreous ultrastructure (Checa et al., 2006; Nudelman, 2015), while more recently other ultrastructures, such as the crossed-lamellar architecture received increasing attention (Böhm et al., 2016; Almagro et al., 2016; Agbaje et al., 2017b). Here we present data and detailed characterization of two rarely investigated ultrastructures, namely the compound composite prismatic and the crossed-acicular ultrastructure, which are common to bivalves of the Veneridae family (Shimamoto, 1986).

We are using a combination of pulse Sr labelling aquaculture experiments and high-resolution microanalytical methods to gain insight into submicron architecture and growth dynamics in the two different portions of the outer shell layer. Pulse Sr labelling experiments have contributed significantly to our understanding of submicron scale growth mechanisms in marine calcifiers such as scleractinian corals, echinoderms and foraminifera (Shirai et al., 2012; Nehrke et al., 2013; Domart-Coulon et al., 2014; Gorzelak et al., 2014; Gutner-Hoch et al., 2016). Since pulse Sr labelling experiments provide time gauges for shell growth at high spatial resolution, this method enables study of time-resolved growth of individual submicron sized architectural units in the shell relative to local growth, which, due to the curvature of the shell, can vary by up to 90˚ in direction from the direction of dorso-ventral shell extension in bivalves.

## 2 Materials and methods

### 2.1 Aquaculture and pulsed Sr-labelling experiments

The "common cockle" or "ridged venus" *Katelysia rhytiphora* (Lamy, 1935) is a temperate, shallow burrowing, intertidal species that occurs along the shorelines of Tasmania and south-eastern to south-western Australia (Edgar, 2000). Some species

of the genus *Katelysia* are edible (*K. peronei, K. rhytiphora,* and *K. scalarina)* and have been a historical food source in Australia as seen by their occurrence in aboriginal shell middens (Cann et al., 1991). Today, *Katelysia* are produced in aquaculture (Nell et al., 1994), and shells in the wild are used to extract environmental parameters (Nell and Paterson, 1997). *K. rhytiphora* shells were collected alive at Port Lincoln, South Australia from fine- to medium-grained sand in the intertidal zone. Twenty-nine bivalves were placed in polyethylene boxes (20 x 40 x 10 cm, 7 bivalves per box) filled with sterilized beach sand and placed within 50 litre polyethylene tanks at the seawater facility at Macquarie University in September 2016. All tanks were connected to a recirculating system with filtered, sterilized natural seawater. Temperature and water chemistry, including salinity, pH matched ocean values. The setup in smaller sand-filled boxes enabled easy and quick transfer of the bivalves between the larger tanks, thus minimizing handling stress. Indeed, the bivalves were observed to continue filter-feeding while being transferred, which is a reliable sign for the absence of handling stress. Acclimatisation period was 3 weeks and experiments lasted 36 days. A 12h/12h day/night light cycle was maintained throughout the experiment and the water was homogenized using an air-stone. The bivalves were fed daily with a mix of microalgae "Shellfish Diet 1800" (Reed Mariculture Inc., USA) containing *Isochrysis sp., Pavlova sp., Tetraselmis sp., Chaetocerous calcitrans, Thalassiosira weissflogii,* and *Thalassiosira pseudonana*. After acclimatisation, bivalves were transferred twice for 6 days each to labelling seawater conditions at 18 x mean ocean water average of 144 µg·g$^{-1}$ Sr (4.380 g SrCl$_2$ x 6H$_2$O in 10 l seawater). Between labelling events, bivalves spent 12 days at normal seawater conditions (ca. 8 µg·g$^{-1}$ Sr). After the last labelling event, some bivalves were collected after 6 days at ambient conditions, while the remaining specimens were collected after 12 days. The pulsed Sr labelled periods are referred to as "labelling events", LE1 and LE2, whereas "normal events" NE1 and NE2 refer to background conditions, with ambient marine Sr levels. The water quality was maintained by fully renewing the spiked seawater every 48 hours with a freshly produced batch (using 4.380 g SrCl$_2$ x 6H$_2$O per 10 l seawater). Over the entire course of aquaculture an effort was made to keep the conditions (temperature, salinity, pH, lighting), including food availability, as stable as possible, so that Sr-concentration in the seawater was the only altered variable. After the experiments, bivalves were deep-frozen at -20 °C. After thawing and removing of soft tissues, shells were rinsed in deionized water and air-dried.

## 2.2 Sample preparation

Valves were cut along the maximum growth axis using an IsoMet low speed precision sectioning saw (Buehler, IL, USA). Left valves were mounted using EpoFix epoxy resin (Struers, Australia), while 3 mm-thick shell slabs from right valves were fixed on microscopy glass slides using metal bisphenol-A-epoxy resin (Permatex, Hartford, CT, USA). After curing at room temperature, sample surfaces were ground and polished using sandpaper (P400-P2000) as well as 3 and 1 µm diamond pastes. Left valves were further polished using a final chemical polishing step with a diluted suspension of colloidal silica (0.05 µm) for one minute on a neoprene polishing cloth to ensure optimum conditions for high-resolution analyses. Additional shell pieces were immersed in a solution of 1% wt./vol. ethylenediaminetetracetic acid disodium salt dihydrate (EDTA; Sigma-Aldrich), ultra-sonicated for 6 minutes, rinsed with Milli-Q water and air-dried. For SE-images un-etched broken shell pieces

and some etched with EDTA (1% wt./vol) were mounted on aluminium stubs using carbon glue, and gold-coated with a thickness of 15 nm.

## 2.3 Optical microscopy

A Leica M205C binocular stereomicroscope with reflective light was used to image shell slabs along the entire shell cross-section. Images were stitched and contrast improved in Adobe Photoshop CS5. To obtain greyscale line profiles, the image part containing the prismatic oOSL was cropped and further improved in contrast. Greyscale line profiles were acquired using ImageJ (Schindelin et al., 2015).

## 2.4 Electron probe micro analyser (EPMA), field emission gun scanning electron microscopy (FEG-SEM) and electron backscatter diffraction (EBSD)

Quantitative wavelength-dispersive X-ray spectroscopy (WDS) was carried out using a JEOL JXA 8200 electron probe micro analyser (EPMA) at the University of Mainz, Germany, with a defocused beam in rastering mode at 20,000 x magnification to obtain concentrations of Na, Mg, P, S, Cl, K, Ca, Mn, Fe, Sr, and Ba calibrated against a variety of minerals and synthetic reference materials (Table S1). Backscattered electron (BSE) images at lower magnification were acquired from carbon-coated polished cross-sections. Specimens were imaged with 15 kV acceleration voltage and 8 nA beam current at 11 mm working distance. Epoxy mounts and broken pieces of shells were imaged with field-emission gun scanning electron microscopes (FEG-SEM), namely a JEOL JSM- 7100F and a Phenom XL at Macquarie University (BSE images at 15 kV and 8 nA), and a ZEISS Leo 1530 at the Max Planck Institute for Chemistry, Germany, for secondary electron (SE) images (at 3 kV and 2 nA).

Electron backscatter diffraction (EBSD) data were acquired at Oxford Instruments NanoAnalysis, High Wycombe, United Kingdom, using a Hitachi SU70 FE-SEM equipped with an Oxford Instruments AZtec 3.4 EBSD-EDS system, with an X-Max 150 mm$^2$ EDS detector and a CMOS-based Symmetry EBSD detector. Three EBSD maps were collected along the axis of maximum growth in different regions of interest using 15 kV accelerating voltage, a beam current of 10 nA and a step size of 0.1 μm. The EBSD pattern resolution was 156 x 128 pixels at a collection rate of 195 patterns per second. Noise reduction was performed using the HKL software and datasets were processed using the MTex toolbox in Matlab (Bachmann et al., 2010; Mainprice et al., 2011) following the protocol in Henry et al. (2017). All EBSD data points were used for the calculation of the Young's modulus.

## 2.5 Micro-Raman spectroscopy

Raman spectra were recorded at room temperature using a Horiba Jobin Yvon LabRAM HR Evolution spectrometer coupled to an Olympus optical microscope with the laser beam path aligned through the microscope objective (quasi-backscattering configuration). A diode-pumped solid-state laser with 473 nm (~15 mW at sample surface) and a He-Ne laser with 633 nm (~10 mW at sample surface) excitation wavelength were used. Spectra recorded in the red spectral range ($\lambda_{exc} = 633$ nm) have

a spectral resolution of 0.8 cm$^{-1}$ and a pixel resolution of 0.3 cm$^{-1}$; those recorded in the blue spectral range ($\lambda_{exc}$ = 473 nm) have a spectral resolution of 1.6 cm$^{-1}$ and 0.6 cm$^{-1}$ pixel resolution using a grating with 1800 lines /mm.

Hyperspectral images were obtained using a software-controlled x-y table and a step width of 0.6 µm. All instrument set-up parameters and measurement conditions were kept constant during automated point-by-point spectra acquisition to guarantee subtle changes of Raman band parameters to be recorded reliably. Minute modification of Raman band parameters as obtained from hyperspectral mapping were interpreted only qualitatively. Data reduction included background subtraction and peak fitting using Lorentzian-Gaussian (pseudo-Voigt) function. All FWHM values were corrected for the instrumental apparatus function using the empirical correction published in Váczi (2014).

## 2.6 NanoSIMS analyses

Epoxy mounts were gold-coated prior to introduction into a new generation CAMECA NanoSIMS 50L ion probe equipped with a Hyperion RF plasma oxygen ion source, at the University of Western Australia. The primary oxygen ion beam was focused to a diameter of 100 nm and images were acquired from 100 × 100 µm$^2$ areas at a resolution of 1024 × 1024 pixels with a dwell time of 3.6 ms/pixel. $^{24}$Mg, $^{40}$Ca, and $^{88}$Sr were measured on electron multipliers at a mass resolving power of 5000. The imaged areas were pre-sputtered at a slightly larger map area prior to acquisition. Images were processed using the OpenMIMS plugin for ImageJ/FIJI, where a correction for detector dead time was applied and the ratio of $^{88}$Sr/$^{40}$Ca are expressed as a Hue-Saturation-Intensity (HSI) colour scale – min (blue) = 10, max (magenta) = 100.

## 2.7 Thermal gravimetric analysis (TGA)

Using a DREMEL power tool, fractions of both portions of the outer shell layer were obtained by removing the iOSL in one shell fragment and the oOSL in an other. Both samples were soaked in H$_2$O$_2$ (Merck KGaA, Darmstadt; Germany) for 1 hr at room temperature and washed with Milli-Q water. After air-drying, each sample was powdered using an agate mortar and pestle. Total amounts of organics were determined with a TGA 2050 thermogravimetric analyser (TA Instruments, USA). About 10 mg of powdered sample was measured (two replicates). The analysis was carried out under a nitrogen atmosphere, at a linear heating rate of 10°C/min, between 25–1000°C.

## 3 Results

### 3.1 Ultrastructure and shell growth

The outer surface of *K. rhytiphora* shells show prominent, concentric ridges (Fig. 1A) and a yellow and purple to brown pigmentation on the inside (Fig. 1B). The shell is fully aragonitic (Fig. S1) and our study focussed on the two architecturally different outer layers of the shell, to the outside of the pallial line. Underneath a very thin periostracum (see subchapter 3.4), the outermost outer shell layer (oOSL) consists of a compound composite prismatic architecture, while the innermost outer shell layer (iOSL) has a crossed-acicular ultrastructure. General thickening of the whole shell is achieved by the inner layer

beyond the pallial line (Fig. 1C). The oOSL of *K. rhytiphora* shells studied here is characterised by three dark bands near the ventral margin (Fig. 1C- E). A minor dark band at the very tip of the shell corresponds to growth in November when the bivalves were sacrificed (Fig. 1C-E).

The compound composite prismatic ultrastructure, which is considered to be one of the most complex shell architectures known represents an umbrella-term for a family of differently arranged hierarchical prismatic ultrastructures (Taylor, 1969; Popov, 1986; Shimamoto, 1986). First-order prisms in the oOSL have thicknesses between 10-30 µm and run parallel to the outer shell surface, with the long axis of the prisms oriented parallel to the main growth axis (Popov, 1986). These first-order units consist of 0.3 µm thick second-order prisms (Shimamoto, 1986) that protrude radially from the central axis of first-order prisms creating a feather-like appearance when viewed in cross-section (Taylor, 1969; Shimamoto, 1986). Each prism in both hierarchical orders is covered by a thin organic sheath (Taylor, 1969; Shimamoto, 1986). Taylor (1969) also observed smaller units within second-order prisms delineated by organic matrix and we refer to these units as third-order prisms here.

Two different schools of thought group the crossed-acicular ultrastructure with other structurally related architectures: Shimamoto (1986) classifies the crossed-acicular ultrastructure as a subtype of the homogeneous ultrastructure, while Marin et al. (2012) groups the crossed-lamellar, complex crossed-lamellar, and crossed-acicular ultrastructures together. The crossed-acicular ultrastructure has previously been comprehensively described for the marine gastropod *Cuvierina* (Carter, 1989) and consists of single lamellae that are arranged into bundles intersecting at angles of 120-150° with dipping angles of 30 to 40° relative to the inner shell surface (Carter, 1989).

## 3.2 Validation of Sr incorporation

Qualitative NanoSIMS mapping revealed two distinct bands of elevated Sr concentration in the oOSL at the ventral margin (Fig. 2A) as well as in the iOSL ca. 0.5 mm away from the ventral margin (Fig. 2B). Correlation of NanoSIMS maps with BSE images verify that light greyscales in BSE images are indeed caused by higher concentrations of Sr in the shell.

EPMA-based WDS analyses (Tables 1 and S2) show that Sr contents are generally higher in the oOSL than in the iOSL, averaging 19,500 µg·g$^{-1}$ for the oOSL and 12,000 µg·g$^{-1}$ for the iOSL (note that the value for iOSL is a minimum value as the analysed area slightly exceeds the label width). Strontium concentrations in growth regions formed before aquaculture (pre-aqua), during acclimatisation (pre-LE 1) and between labelling events (NE 1) are around 1,120 µg·g$^{-1}$ for oOSL and again lower (1,010 µg·g$^{-1}$) for the iOSL (Tables 1 and S2). Likewise, average molar ratios of Sr/Ca (mmol/mol) range from 1.32 (oOSL) and 1.18 (iOSL) in shell sections grown during ambient conditions in aquaculture to 14.55 (iOSL) and up to 23.60 (oOSL) in shell portions grown during pulse labelling. Hence, the increase of 18x mean ocean water concentrations (144 µg·g$^{-1}$ Sr) in seawater resulted in a 17x increase in Sr in the oOSL (19,500 µg·g$^{-1}$) in the labelled compared to the unlabelled conditions and in about 12x increase for the iOSL (>10,970 µg·g$^{-1}$). Concentrations of other minor elements (Na, Mg, S, Cl) are generally lower in the iOSL and were identical within uncertainty between labelling and non-labelling experiments. Molar

ratios for Na/Ca and Mg/Ca range from 13.56 to 25.91 and from 0.76 to 1.05, respectively, and do not correlate with high Sr concentrations. Concentrations of Mn, Ba, P, K, and Fe in the shells are below detection limits.

## 3.3 Raman spectroscopy

Raman spot analyses in both studied shell layers show peak positions characteristic for aragonite (Fig. S1), namely a doublet at 701 and 705 cm$^{-1}$ ($v_4$, CO$_3$ in-plane bending), a peak at 1084.8 cm$^{-1}$ ($v_1$, CO$_3$ symmetric stretching), and several modes between 170 and 300 cm$^{-1}$ that are due to rotations and translations of Ca$^{2+}$ and CO$_3^{2-}$ units (Urmos et al., 1991; Wehrmeister et al., 2010; Carteret et al., 2013). In addition, broad bands centred at 1134 and 1532 cm$^{-1}$ represent C-C single bond ($v_2$ stretching mode) and C=C double bond ($v_1$ stretching mode) vibrations of polyene chains in organic pigments in the shell (Otter et al., 2017).

Micro-Raman hyperspectral mapping of the most intensive peak at 1084.8 cm$^{-1}$ revealed that band widths (full-width at half-maximum, FWHM) differ between Sr labelled and unlabelled areas (Fig. 3). Two regions with systematic peak broadening in both ultrastructures correspond to the Sr labels seen as bands of light greyscale in BSE images and represent a change in concentration from 19,500 µg·g$^{-1}$ Sr in labelled to 1,120 µg·g$^{-1}$ in shell portions grown in ambient conditions. Although Sr concentrations in the seawater and duration of labelling conditions were identical for all labelling periods, the more recent outer label (LE 2) is narrower and brighter than in the earlier label (LE 1), reflecting different shell growth rates. Band width distribution shows distinct narrow increments within both labels.

Highest FWHM values within each labelled area are 2.2 and 2.7 cm$^{-1}$ for the oOSL (Fig. 3A) and 1.8 and 2.4 cm$^{-1}$ for the iOSL (Fig. 3B, Table S3), while FWHMs in unlabelled areas are less than 1.8 cm$^{-1}$. The $v_1$ symmetric stretching band shows a shift in peak position to lower wavenumbers in areas of high Sr concentration (Fig. S2, Table S3). Note that FWHMs and peak positions do not vary among different architectural features in the unlabelled shell architecture, and hence, are not influenced by grain size effects.

## 3.4 Architecture of the shell layers

### 3.4.1. The compound composite prismatic architecture (oOSL)

As visible in radial sections of the oOSL (Fig. 4A) first-order prisms are oriented with their long sides parallel to the umbo-ventral margin axis and form a fan-like arrangement resulting in the ridged outer surface (Taylor, 1969; Popov, 1986; Shimamoto, 1986). First-order prisms originate and end at the organic-rich growth checks (Fig. 4A) and can reach sizes of >700 µm (projected 2D length) and widths of 17µm (aspect ratio of >40). Growth checks can be organic rich as observed here, or are fully mineralized with a different morphology, such as a thin layer of prisms (see below; Ropes et al., 1984). In contrast to studies that reported first-order prisms to exhibit square shapes in longitudinal cross-sections (e.g., Taylor, 1969), we observed irregular six-sided prism cross-sections in these *Katelysia* shells (Fig. S4). Measured widths of around 17 µm compare to literature values of around 10 µm for other venerid shells (Shimamoto, 1986). First-order prisms consist of second-order

prisms arranged radially around their central axis at an angle of 68° (Figs. 4B, S5), resulting in a feathery arrangement of second-order prisms in cross-sections (Popov, 1986; Shimamoto, 1986). Individual second-order prisms have projected lengths and widths of 3 ±0.3 μm and 0.3 ±0.06 μm (n=8), with an aspect ratio of 10. The widths are in accordance with values provided by Shimamoto (1986) for the shells of other venerid shells. Both first- and second-order prisms were found to be enveloped by organic sheaths as indicated by darker greyscales in the BSE images (Figs. 4A, B, S4) supporting literature findings for this ultrastructure (Shimamoto, 1986; Taylor, 1969).

Second-order prisms consist of third-order prisms (Fig. 4C), which are arranged with their long axes parallel to each other. They have lengths of 496 ±129 nm and widths of 67 ±16 nm (n=8, Fig. S6) with a lower aspect ratio of 8 compared to first- and second-order prism. Lastly, the smallest building blocks revealed by SEM images in etched shell samples are nano-granules with sizes in the range of 70 nm (Fig. 4E and F).

### 3.4.2. The crossed-acicular architecture (iOSL)

The acicular-prismatic boundary is marked by a ca. 30 μm wide transitional layer of granular texture comprising high numbers of short first-order prisms and spherulitic grains (Fig. S3). The growth check, which is organic rich in the oOSL, continues as a thin prismatic layer into the crossed-acicular ultrastructure of the iOSL (Fig.4A, green arrow). Bundles of cross-layered lamellae in the iOSL are enveloped by organic sheaths (dark grey, Figs. 4D, S7) and measure up to 1.4 x 0.8 x 0.2 μm (Fig. S8). Individual acicular lamellae are 1.8 ±0.4 μm long and 0.22 ±0.05 μm (n=19) wide with aspect ratios of about 8 (Figs. 4D, S7). The angle between acicular lamellae is 81 ±8° (n=6). Similarly to the oOSL, etching revealed a nano-granular texture in this layer (Fig. 4F).

### 3.4.3. Organic content

Thermal gravimetric analysis (TGA) was used to determine the total amount of organic macromolecules in the shell, which amounts to 1.42 ±0.03 wt.% and 2.19 ±0.04 wt.% for the iOSL and oOSL, respectively (Fig. S9). The organic phases are visible after etching the mineral phase and exhibit fibre- and sheet-like shapes (Fig. 4E, F).

### 3.5 Crystallographic preferred orientations

The ultrastructure of the aragonite grains in the oOSL and iOSL shell layers is shown in the orientation map in Fig. 5A. The map is colour-coded using an inverse pole figure colour scheme and shows the crystal direction in the orientation map facing the reader with blue, green, and red for the crystallographic a- [100], b- [010], and c-axis [001], respectively. Fig. 5A shows the feathery arrangement of the second-order prisms within the first-order prisms (outlined in green) as described above. The rims of the first-order prisms in the oOSL are well-resolved in the orientation map (Figs. 5, S10), while most of their cores remains dark and unindexed, indicative of poor or non-existent diffraction patterns as measured during the EBSD indexing cycle. We believe this effect is an artefact of sample polishing arising from preferential removal of the nearly vertically oriented

second-order prisms in these areas (Fig. S11). The alternative explanation, namely reduced crystallinity in these areas is highly unlikely as this would have been detected in Raman maps via significant band-width changes (Figs. 3, S2).

The iOSL with crossed-acicular architecture shows ca. 17 µm by ca. 10 µm large areas of lamellae, where the aragonite crystallographic axis are well co-orientated (Fig. 5A, outlined in yellow) and have high amounts of crystallographic twin boundaries (Figs. 5, S12).

Pole figures (Fig. 5B) show a strong preferred orientation of the aragonite c-axes perpendicular to the growth layers in the crossed-acicular architecture, while the crystallographic a and b axes are scattered on a plane normal to the local growth direction. The local growth direction in the crossed-acicular ultrastructure (green arrow in Fig. 5B) is perpendicular to the light grey Sr labels in the underlying BSE image and at this locality differs by ca. 90° from the general shell growth direction (white arrow). In comparison, the local growth direction of the compound composite prismatic layer (oOSL: purple arrow in Fig. 5B) has a smaller angle with the general shell growth direction. We identified a high abundance of twinning with 46% (oOSL) and 56% (iOSL) of the grains showing at least one twin (i.e. 63.8° ±5° rotation around the 001 axis).

## 3.6 Local growth rates

The pulsed Sr labels are easily visible in both the oOSL and the iOSL in BSE images as bands of bright greyscale (Fig. 4A). In general, greyscale values in BSE images cannot be relied on for trace element quantification. In this study, however, we have calibrated the BSE grey scale using quantitative WDS-based EPMA measurements in the same analytical session. Moreover, correlative mapping of the Sr distribution with NanoSIMS and micro-Raman spectroscopy (Figs. 2, 3, S2) clearly correlates the bright greyscales in the BSE images spatially with the Sr-labelled areas. Thus, in this study, greyscales in the BSE images reflect variations in Sr concentrations on the shells without any doubt.

Commonly, bivalve shell growth rates are reported as the macroscopic linear dorso-ventral shell extension ("general growth direction" in this study). Our high magnification images require us to take into account that local growth directions of the architectural units differ from the macroscopic linear dorso-ventral shell extension axis. Previously, these have been referred to as "crystal growth rate" (e.g., Gillikin et al., 2005; Carré et al., 2006). Instead, we use the term "local growth rate", because "crystal growth rate" does not reflect recent research that established the mesocrystalline nature of the material, initially formed as amorphous calcium carbonate.

Table 2 summarises the average local daily growth rates for all experiments (for detailed dataset see tables S4, S5). Length measurements were acquired in triplicate at five different locations on cross-sections along the maximum growth axis using the software ImageJ (Figs. S14, S15, Tables S4, S5). Although sizes and ages of the bivalve shells are similar, absolute local growth rates vary among specimens, especially for the oOSL (Fig. 6A, B). On a daily average within 6 days of pulsed Sr-labelling procedure, layer LE1 grew 0.93 ±0.15 µm (range: 0.37 – 2.22 µm), while layer LE2 grew 0.60 ±0.12 µm (range: 0.43 – 0.80 µm; Tables 2, S4). A 12-day ambient period (NE1) resulted in an average daily growth of 1.02 ±0.09 µm (range: 0.31 – 1.86 µm). The last 12-day ambient period (NE2) resulted in an average daily growth of 0.76 ± 0.08 µm (range: 0.47 – 1.41 µm). In comparison, the crossed-acicular ultrastructure (iOSL) grew only 0.88 ±0.10 µm (range: 0.58 – 1.17 µm) during

LE1 and 0.72 ±0.05 μm (0.62 – 0.92 μm) during LE2. Twelve days of ambient conditions (NE1) resulted in 0.75 ±0.04 μm (0.31 – 1.13 μm) daily average growth and for NE2 in 0.47 ±0.03 (0.43 –0.84). Based on average daily growth rates, oOSL grew 17 % faster than iOSL, which showed steadier growth (i.e. smaller standard deviations). Growth rates decrease with increasing distance to the ventral margin along iOSL (Fig. 4A). Individual Sr labels offer further detail and comprise several narrow increments of varying width and greyscale intensity in both ultrastructures (Figs. 4B, D). A systematic shift towards faster or slower local growth rates during Sr incubation was not observed (Fig. 6C).

## 4 Discussion

### 4.1 Multiscale architecture

Compared to simple prisms in the nacroprismatic bivalve ultrastructure the compound composite prismatic ultrastructure of *K. rhytiphora* is far more complex, containing three orders of prisms with sizes ranging from mm (first-order prisms) to nm (third-order prisms; Figs. 4A-C, S11). With respect to the number of hierarchically distinct units, the compound composite prismatic ultrastructure shares more similarity with the crossed-lamellar architecture than with the simple prism ultrastructure (Agbaje et al., 2017b). In *K. rhytiphora*, the first-order prisms run perpendicular to the growth checks and radially with respect to the radial cross-section (Fig. 4A) and comprise two orders of acicular prisms with high aspect ratios that are arranged feathery (radially in 3D around the central prism axis) in the case of second-order prisms and parallel in the case of third-order prisms (Fig. 4B, C).

Organic contents of both shell layers, namely 2.2 wt.% in the oOSL and 1.4 wt.% in the iOSL (Figs. 4E-F, S9), are intermediate between nacroprismatic shells (3-5 wt.% total organic content) and the highly mineralized crossed-lamellar shells with less than 1 wt.% organic content (Agbaje et al., 2017a).

Second-order prisms in the oOSL are co-oriented across their thin organic envelopes and, likewise, lamellae in the iOSL show co- orientation over 10 μm (Figs. 6A, S10). Co-orientation across the delineating organic sheath in the shell is a general observation for all bivalve shell architectures (e.g., Gilbert et al., 2008; Agbaje et al., 2017b) and is the result of the epitaxial growth mechanism via mineral bridges across the organic scaffolding (Checa et al., 2011). This model involving mineral bridges was developed for growth mechanisms in nacre, which has comparatively thick organic interlamellar sheets of ca. 30 nm and where 150-200nm sized mineral bridges are indeed visible (Checa et al., 2011; Nudelman, 2015). The organic sheaths in the *K. rhytiphora* shells are significantly thinner than the interlamellar membranes in nacre and mineral bridges across these would only require a few nanogranules of $CaCO_3$, the 30-50nm sized basic building blocks in bivalve shells (Wolf et al., 2016).

The crossed-acicular ultrastructure (iOSL) is built less complex than the prismatic ultrastructure (oOSL) and consists of only two architectural orders: (i) cross-layered individual lamellae of a few microns in length are angled at approx. 80° to each other and have dipping angles of <20° towards the inner shell surface, and (ii) cross-layered bundles of co-oriented lamellae at a higher hierarchical order (Fig. S8). Similar bundle-like arrangements of crossed-acicular lamellae were observed

by Carter (1989) in the marine gastropod *Cuvierina*, but these show larger angles to each other and smaller dipping angles than those observed in this study. In orientation maps for the iOSL (Figs. 5A, S12), some pseudo-prisms (Pérez-Huerta et al., 2014) can be identified (outlined in yellow) that consist of co-oriented lamellas.

A common structural motif of aragonitic bivalve shells is the high amount of crystallographic twinning. In *K. rhytiphora*, we observed 46 % (oOSL) and 56 % (iOSL) twin boundaries. Similar to amounts reported for crossed-lamellar (26%) and nacreous (20-65%) ultrastructures (Chateigner et al., 2000; Agbaje et al., 2017b). Aragonite twinning in bivalve shells encompasses all length scales including the nano-scale (Kobayashi and Akai, 1994) and values obtained by EBSD are minimum values as they are a function of the spatial resolution.

The smallest mineralized unit of both ultrastructures in the shells are granules with sizes of tens of nanometres (Fig. 4E). These granules are similar in size to the nano-granular texture observed in nacroprismatic and crossed-lamellar shell samples and have been found to be a common motif for bivalve shells (Jacob et al., 2008; Wolf et al., 2016; Agbaje et al., 2017b). Previous studies showed that these granules are often less well-crystallized or even amorphous and are enveloped by thin organic sheaths (Jacob et al., 2008; Wolf et al., 2012). They are most often the vestiges of a non-classical crystallization pathway via amorphous calcium carbonate ACC (de Yoreo et al., 2015).

## 4.2 Mechanical properties

The mechanical properties of shells i.e. stiffness, impact resistance, and toughness outcompete aragonite single crystals by several magnitudes (Jackson et al., 1988; Wang et al., 2001; Katti et al., 2006) Through evolutionary fine-tuning bivalve shells optimize their mechanical properties via their hierarchical organization, crystallographic twinning, nano-granularity, and the intimate intergrowth of mineral and organic phases at the nanoscale and aim at minimizing anisotropy in certain directions of the shell (Weiner et al., 2000). An important parameter to describe the stiffness of a material in response to stress and strain is the Young's modulus (Hashin, 1962). Young's moduli for *K. rhytiphora* shells, calculated from the EBSD dataset and the elastic constants of aragonite single crystals (Liu et al., 2005) yield a maximum of 139 GPa for the iOSL (Fig. 7A), 132 GPa for the oOSL (Fig. 7B), resulting in 135 GPa for both shell layers together (Fig. 7C). These values are in the range of those reported for crossed-lamellar (Agbaje et al., 2017b) and nacreous shells (Fitzer et al., 2015). The mechanical anisotropy can be defined as 200*(max-min)/(max+min) with max and min being the maximum and minimum values in GPa. For both layers, the mechanical anisotropy reaches 30%. The stereographic projection of the Young's modulus (Fig. 7A-C) reveals a girdle-like maximum of elastically stiffer orientations for the shell that differs significantly from aragonite single crystals (Fig. 7D), but is similar to results for other bivalve shells (Agbaje et al., 2017b). In reference to the shell morphology, this non-random arrangement of crystallographic orientations results in a quasi-isotropic plane of maximum fracture resistance parallel to the local growth lines (GL, Fig. 7A-C) and perpendicular to the local growth direction (and thus curvature) of the shell. Hence, the strongest, most fracture-resistant direction in the shell is parallel to its surface, thus maximising the shell's protective function.

## 4.3 Growth features and growth in the wild

*K. rhytiphora* shells form ornamental ridges on their outer shell surface (Fig. 1), and it is an interesting question how these ridges relate to shell growth. In the case of specimen K2-04 (Fig. 4), the ridge feature spans the area between the two most recent growth checks (Fig. 4A), suggesting a one year growth period for the ridge feature, which is also supported by estimating the growth period using growth rates for this specimen (2.2 µm/day using LE1). However, ridge features are not always associated with growth checks (Fig. 4A, purple arrow). Ridges are evenly distributed and similar in width (Fig. 1C), resulting in a decrease in the number of ridges per year with ontogenetic age of the shell. Similar to our findings for *K. rhytiphora* the surface spines of the gastropod *Strombus gigas* were found to be produced at different periods of time across different individuals, suggesting a genetic rather than an environmental control (Radermacher et al., 2009).

Looking at the formation of the ridge features in more detail, specimen K2-04 shows that the beginning of a new ridge as a fine protruding tip (Figs. 4A, S16), is associated with the highest local growth rates (2.2 µm/day using LE1, Table 2, Fig. S14). Evaluating this observation across all shells shows that at the same point in time, those shells with higher growth rates (e.g. 1.86, Table 2, Fig. S14) started producing their next ridge feature (Fig. S14D, E) while those shells with lower growth rates (e.g. 0.37, 0.46, Table 2, Fig. S14) lag behind (Fig. S14B, F). Supporting the delicate protruding tip of a new ridge by modulating growth rates could be a protective mechanism for this growth feature.

A major difference in the growth patterns between both layers of ultrastructures is that while the growth front in the iOSL is homogeneous and runs straight (Fig. 4D), the growth front in the oOSL is undulated (Fig. 4B and outlined in Fig S13). The centres of first-order prisms in the oOSL protrude compared to their rims (Fig S13) and the constant thickness of the Sr-labelled shell demonstrates that growth rates, measured perpendicular to the growth front, are homogeneous across this area. This undulation is not observed in other prismatic ultrastructures (simple prismatic ultrastructure: Dauphin et al., 2018) and the underlying reasons for this are yet unknown.

Total growth for oOSL and iOSL in aquaculture are on average 28.4 µm (oOSL) and 24.2 µm (iOSL) with daily growth rates of 0.85 ± 0.11 µm for oOSL and 0.73 ±0.07 µm for iOSL (Table 2). Consistent with the curved geometry of the shell and as previously documented (Carré et al., 2005; Foster et al., 2009), the oOSL in *K. rhytiphora* grows 17 % faster than iOSL (Table 2, Fig. 6), and growth rates for this layer are less variable than for the oOSL, both within individual specimens and across the population (Tables 2, S4, S5). While first-order prisms extend between two growth lines and are likely annual, second-order prisms (3 to 6 µm long, Fig. S5) and crossed-acicular lamellae (1.8 µm, Fig. S7) grow at rates of days in our aquaculture experiment, while nanometre-sized third-order prisms (Fig. S6) form within hours. Note however, that while growth rates for the architectural units relative to each other are valid, absolute growth rates in the wild are likely higher compared to aquaculture.

In fact, some insight into shell growth in the wild can be gained from shell portions predating the aquaculture experiments and are described here for the shell section depicted in Fig. 1, which is representative for three specimens in which these observations were made: Alternating light and dark bands seen in the shell cross-section (Fig. 1C-E) represent winter

(light bands) and summer (dark bands) shell growth. This is verified from the final dark band at the tip of the shell that corresponds to growth in late November when the bivalves were sacrificed. Cyclic changes in the greyscale line profile across these bands (Fig. 1C, D) correlate with tidal cycles: light grey and dark grey portions fall together with full and new moon cycles, respectively (Fig. 1D) and indicate that this shell section was deposited over the period two years. Intervals between grey-shaded areas in Fig. 1D correlate well with neap tides. This growth pattern is in accordance with findings that shell growth is strongly influenced by tidal cycles (Evans, 1972; Schöne, 2008; Hallmann et al., 2009), whereby neap tides result in light coloured increments that are generally wider than the dark increments (Rhoads and Lutz, 1980; Schöne et al., 2002; Carré et al., 2005; Carré et al., 2006). Hence, *K. rhytiphora* shells in the wild show a well-defined fortnightly shell increment resolution. Micro-growth bands at the outermost shell tip (Fig. 1D, red box) can be correlated with tides at the sampling locality at Port Lincoln, South Australia from mid-August to mid-September 2016 (Fig. 1E) and indicate that these micro-growth bands formed over this period in 2016 and prior to aquaculture (started mid-September 2016). From this time onward, the line profile ceases to correlate with tides (Fig. 1D, E blue band) and shell increments formed during aquaculture are very dark, reflecting lower than normal growth rates.

Analysis of the Sr labelled bands in the shell at high magnification by Backscatter Electron Microscopy allows further insight into growth conditions in aquaculture: Sr label LE1, for example, (Fig. 4B, oOSL) consists of pairs of bright, narrow and darker, wide increments (Fig. S17A). Identical patterns can be seen in the micro-Raman maps (Figs. 3, S2) and confirm that these variations in greyscale observed in BSE are caused by variable Sr concentrations in the labelled shell portion. A similar pattern is observed in the iOSL (Fig. S17B). It is noteworthy that the number of increment pairs in the label matches the number of days in Sr-enriched conditions (Fig. S17A, B) although the bivalves were maintained at constant conditions (including amount and timing of feeding) with Sr concentrations being the only varied parameter. This indicates that the acclimatisation period of three weeks at the start of the aquaculture experiments was enough for the bivalves to adjust from their circatidal and circalunidian cycles in the wild to the circadian cycle in aquaculture.

## 4.4 Implications for growth dynamics and biomineralization in pulse Sr labelled shells

Rather than gradual transitions in greyscale, the changeover between labelled and unlabelled areas in the shells is characterized by a ca. 500 nm narrow greyscale transition in the oOSL (ca. 150 nm in the iOSL), which is roughly equivalent to shell growth over 5 hours at the growth rates for this shell and this particular local growth rate (K2-04: 2.22 µm/day (oOSL) and 0.72 µm/day, Table 2). The activation volume of the incident electron beam, which could falsify the width estimate of the transition in greyscale is ca. 250 nm (Goldstein et al., 2017); Fig.17A, B), thus does not affect our estimate here. These short-term Sr-concentration changes in the shell thus mirror the immediate change in experimental conditions reasonably well, where seawater was replaced completely both at the start and the end of each Sr-enriched incubation and show that there is no significant lag between change in seawater Sr concentration and Sr incorporation in the shell. This suggests that in the biomineralization of this bivalve species there is no role for a significant 'Sr-reservoir', which would otherwise retain Sr-

concentrations different to the respective batch of seawater and cause gradual changes in greyscale in the BSE images of the shells over a wider shell portion.

One such biomineralization reservoir in bivalves is thought to reside in the space between the mantle epithelium and the growth front of the shell, namely the extrapallial space. The fluid in this space contains high concentrations of $Ca^{2+}$-binding proteins, important agents in biomineralization (Cusack et al., 2008; Rousseau et al., 2009). Our findings that the change in Sr-concentration in the shell closely mirrors the batch-changes of seawater suggest however, that the extrapallial fluid cannot be very voluminous, if it exists at all (Addadi et al., 2006; Marin et al., 2012). These results also demonstrate that changes in Sr concentrations (and, by inference also changes in concentrations of other trace elements) are recorded in the shell without significant temporal delay, which underscores the high suitability of bivalve shells as high-resolution archives of environmental change (Schöne et al., 2005).

An important finding of this study is that all hierarchical architectural units in both shell layers are transected by the Sr label (Fig. 4B, D, summarised in Fig. 8). Thus, rather than forming one individual architectural unit after the other, the growth front in the shell progresses homogeneously, transecting not only all mineral units, but also their individual organic envelopes. Naturally, and consistent with other pulse labelling studies on marine calcifiers the macroscopic morphology of the growth front follows the outside morphology of the skeleton. Nevertheless, at the micron to submicron scale, the homogeneous growth front observed here highlights a fundamental difference to growth processes in other calcifying organisms, where growth fronts are extremely heterogeneous in morphology and in growth rate (Gorzelak et al., 2014; Domart-Coulon et al., 2014). Our observation also potentially challenges the prevailing model for the formation of nacre by successive filling of pre-existing empty organic envelopes (Bevelander and Nakahara, 1969; Levi-Kalisman et al., 2001). Instead, our results for *K. rhytiphora* call for a more dynamic shell growth mechanism that allows for simultaneous formation of organic sheaths and mineral components, perhaps along the lines of models of calcification via directional solidification as recently proposed by Schoeppler et al. (2018).

## 4.5 Strontium/calcium ratios in the shell

The Sr/Ca ratio in bivalve shells has been used as a proxy for sea surface temperature (e.g., Dodd, 1965; Swan, 1956; Zhao et al., 2017a). However, a plethora of studies argues that Sr/Ca ratios in bivalve shells are mainly influenced by growth rate (e.g., Takesue and van Geen, 2004) and metabolic rate (Bailey and Lear, 2006; Foster et al., 2009; Gillikin et al., 2005; Purton et al., 1999) rather than by temperature. Distribution coefficients $D_{Sr/Ca}$ calculated as $(Sr/Ca_{unlabelled\ shell})/(Sr/Ca_{mean\ ocean\ water})$ for both shell layers in this study are very similar for labelled and ambient aquaculture conditions (0.14 and 0.15, Table 3) and are only slightly higher than those in the wild before aquaculture (0.13 for oOSL and 0.12 for iOSL). These $D_{Sr/Ca}$ values are in a similar range as aquaculture-derived $D_{Sr/Ca}$ for shells of the freshwater bivalve *Corbicula fluminea* (0.19 - 0.29, Zhao et al., 2017a), however these values, both for *K. rhytiphora* and *C. fluminea* are significantly smaller than $D_{Sr/Ca}$ for equilibrium incorporation of Sr/Ca in synthetic aragonite of 1.19 at 20 °C (Gaetani and Cohen, 2006). This discrepancy once again highlights the complexities involved in the interpretation of the chemical signatures in biominerals and their correct application

to arrive at accurate reconstructions of past environments. While the exact reasons for the large difference between synthetic and biomineralised aragonite are yet unknown, multi-step fractionation mechanisms connected with the step-wise nonclassical crystallization pathway (Jacob et al., 2017), which is the confirmed formation pathway for many calcifying organisms (de Yoreo et al., 2015) could play a major role.

## 4.6 Effects of aquaculture and pulsed Sr-labelling on growth and composition of the shells

Reduced growth rates are a common observation for bivalves held in aquaculture and *K. rhytiphora* in this study is no exception. A major contributing factor to reduced growth rates in aquaculture for intertidal bivalves such as *K. rhytiphora*, is the very different environment with respect to tidal cycles and lower water depths (e.g., Pannella and MacClintock, 1968). The strong influence of tides on shell growth for intertidal bivalves is well known (Rhoads and Lutz, 1980; Schöne et al., 2002; Carré et al., 2005; Carré et al., 2006), hence an aquaculture protocol that takes increased water pressures into account would be expected to enhance growth rates in future experiments.

The Micro-Raman maps demonstrate the influence of the incorporation of high Sr concentrations on the aragonite crystal structure: In the Sr- labels the $\nu_1[CO_3]$ $\nu_1$ symmetric-stretching band-position is broadened by ca. 0.5 cm$^{-1}$ (Fig. 3) and down shifted by ca. 0.5 cm$^{-1}$ (Fig. S2, Table S3) compared to the areas formed at ambient conditions. This peak shift as well as the peak broadening results from changes in the interatomic distances in the aragonite crystal structure and slightly increase in structural disorder due to the incorporation of the larger Sr ion on nine-fold coordinated smaller Ca-sites (Alia et al., 1997). These effects on the Raman bands of the anionic complexes in minerals are typical when larger cations are substituted in the crystal lattice (Bischoff et al., 1985; O'Donnell et al., 2008; Ruschel et al., 2012).

Hence, while Raman spectra show that Sr-labelling has a measurable effect on the crystal structure of the aragonitic shell, this effect is minor, because (i) Sr substitution into the shell aragonite does not result in formation of a discrete SrCO$_3$ phase, which would have been detected as a band at 1073 cm$^{-1}$ (Alia et al., 1997), and (2) analysis of the EBSD data (Figs. 6, S10, S12) does not show systematic deviations between the labelled and ambient areas in the shell. Furthermore, daily local growth rates of Sr-labelled and unlabelled areas, do not show systematic trends (Fig. 5C). Hence, while shell growth rates are downscaled during aquaculture, the multi-scale architecture of the shell down to the atom-level show no significant deviation from natural shells, indicating that the shell growth processes in aquaculture under the conditions chosen in this study are comparable to those in the wild.

Hence, pulsed Sr-labelling experiments offer the potential to study calcification processes down to the sub-micron range without apparent alteration of the growth processes and offer excellent analytical detectability for a wide range of micro-beam techniques. Pulse Sr-labelling is thus superior to experiments with fluorescent markers that are limited to the spatial resolution of light microscopy and have been shown to impact vitality and biomineralization processes in some calcifiers (Russell and Urbaniak, 2004; Thébault et al., 2006; Allison et al., 2011; Gorzelak et al., 2014).

## 5. Conclusion

Pulsed Sr-labelling experiments and correlated, in situ NanoSIMS and Raman mapping together with WDS spot analysis and FEG-SEM BSE imaging resolve local growth rates at the nanometre scale and show compelling potential to shed light on submicron growth mechanism in bivalve shells:

- All hierarchical architectural units and intercalated organic sheaths are transected by the Sr label and demonstrated bivalve shell growth to progress homogeneously instead of forming one individual architectural unit after the other.
- Sharp transitions between labelled and unlabelled shell areas indicate that physiological transport processes for Sr have no significant lag and suggest that the extrapallial fluid cannot be very voluminous.
- Both architectures have similar $D_{Sr/Ca}$ for labelled and unlabelled shells that agree well with those of shell formed in the wild and are all significantly below $D_{Sr/Ca}$ for equilibrium incorporation of Sr/Ca in synthetic aragonite.

## Data availability

All data can be accessed by email request to the corresponding author.

## Author contribution

LMO and DEJ designed and coordinated the study. LMO conducted aquaculture experiments, sample preparation, analyses. OBA, CL, and PH, participated in data collection. NanoSIMS and EBSD data were collected by MRK and PT, respectively. HH participated in EBSD data processing. All authors contributed to the manuscript and gave final approval for publication.

## Conflict of interest

The authors declare that they have no conflict of interest.

## Acknowledgements

We acknowledge Michael W. Förster, Antje Sorowka, Steve Craven, and Jacob Bynes for help and advice on sample preparation. We thank the Macquarie University Faculty of Science and Engineering Microscope Facility (MQFoSE MF) for access to its instrumentation and support from its staff members Sue Lindsay and Chao Shen. Jane Williamson and Josh Aldridge are thanked for access to and assistance at the Macquarie Seawater Facility. Wayne O'Connor (Port Stephens Fisheries Centre, NSW Department of Primary Industries) is thanked for insightful discussions on husbandry protocols. The authors received financial support through an Australian Government International Postgraduate Research Scholarship (IPRS) awarded to LMO, a Macquarie University Research Excellence Scholarship (iMQRES) awarded to OBA, and DEJ is supported

through an ARC Discovery Grant (DP160102081). C.L. gratefully acknowledges funding through the ARC Centre of Excellence CCFS at Macquarie University, Sydney and financial support by the Austrian Science Fund (FWF), through project J3662-N19.The authors acknowledge Microscopy Australia, the Science and Industry Endowment Fund, and the State Government of Western Australian for contributing to the Ion Probe Facility at the University of Western Australia. We are grateful to the handling editor, H. Kitazato, and 2 anonymous reviewers for their constructive comments.

**Figures and figure captions**

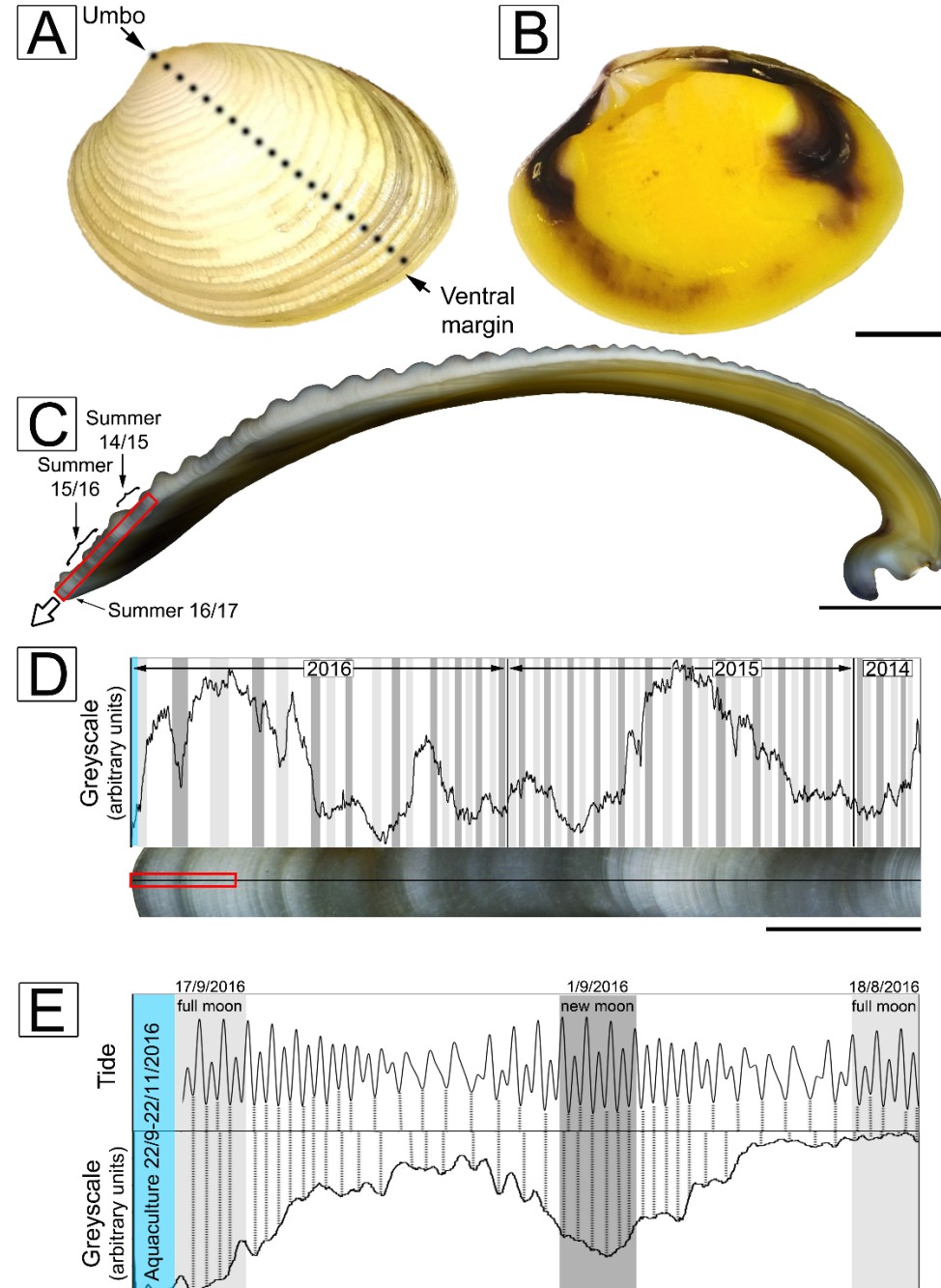

Figure 1: Outer (A) and inner shell surface (B) of an unlabelled *K. rhytiphora* shell. Dashed black line in (A) indicates where the shell was cut to produce the cross-section where a white arrow in (C) indicates the general growth direction of the shell. All cross-sections in this study are prepared as radial sections along the maximum growth axis unless otherwise specified. Dark bands (indicated by

arrows in C) result from growth during summer between lighter coloured winter periods and are magnified in D (red box in C) with a greyscale line profile. Darker greyscale intensities correlate with 48 out of 50 spring tides in two years from the collection site of the bivalves (full moon: light grey, new moon: dark grey) suggesting a fortnightly growth resolution in this shell area. Greyscale line profiles (E) of the area marked by the red box in D shows the most recent dark shell growth increment formed in the wild (mid-August to mid-September 2016). In this shell part, tides correlate with most shell increments (black dashed lines), while this correlation is lost after start of aquaculture (blue area). Blue area in D, E marks the aquaculture period with lower than normal growth rates. Scale bars are 10 mm (A-B), 5 mm (C), 1 mm (D), 0.1 mm (E).

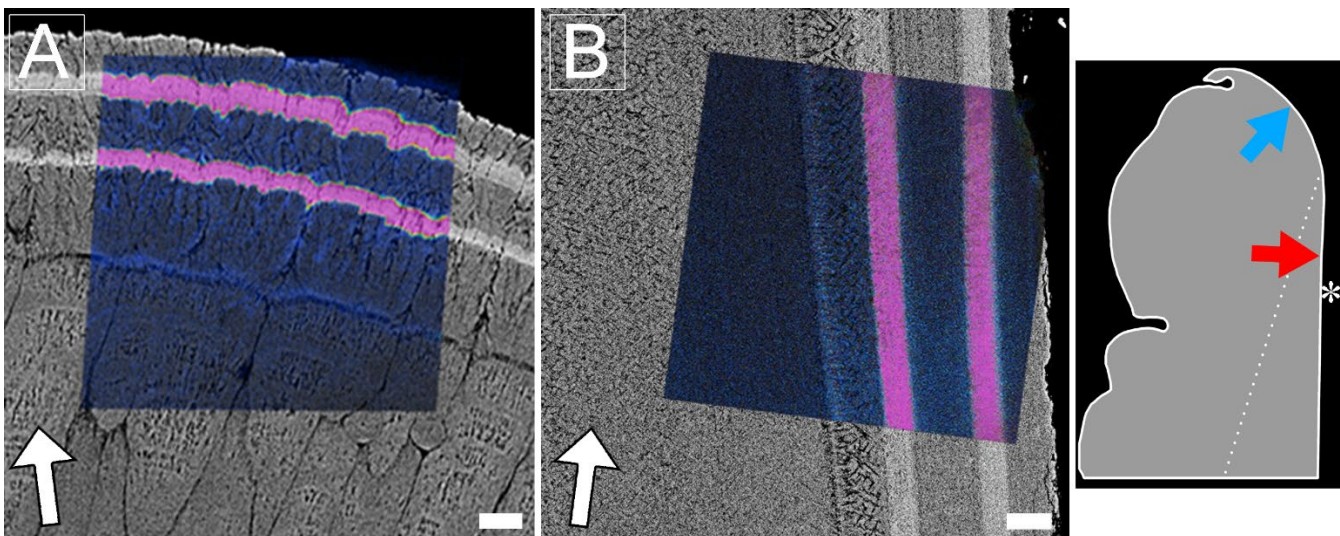

Figure 2: FEG-SEM BSE images showing polished cross-sections of the oOSL (A) and iOSL (B) of a Sr-labelled *K. rhytiphora* shell (specimen ID: K2-06) overlain with NanoSIMS $^{88}Sr/^{40}Ca$ maps. Shell layers grown in ambient seawater $^{88}Sr/^{40}Ca$ ratios are depicted in blue, while shell formed during Sr-enriched incubations are shown in pink. White arrows point towards the general growth direction of the shell, while the Sr-labelled shell layers from the underlying BSE image visualise the local growth directions for each ultrastructure. A schematic of the shell tip shows the exact location of the NanoSIMS maps with a blue and red arrow pointing towards the locations of the prismatic oOSL (a) and crossed-acicular iOSL (b) sampling location, respectively. Asterisk marks the inner shell surface. Scale bars are 10 μm.

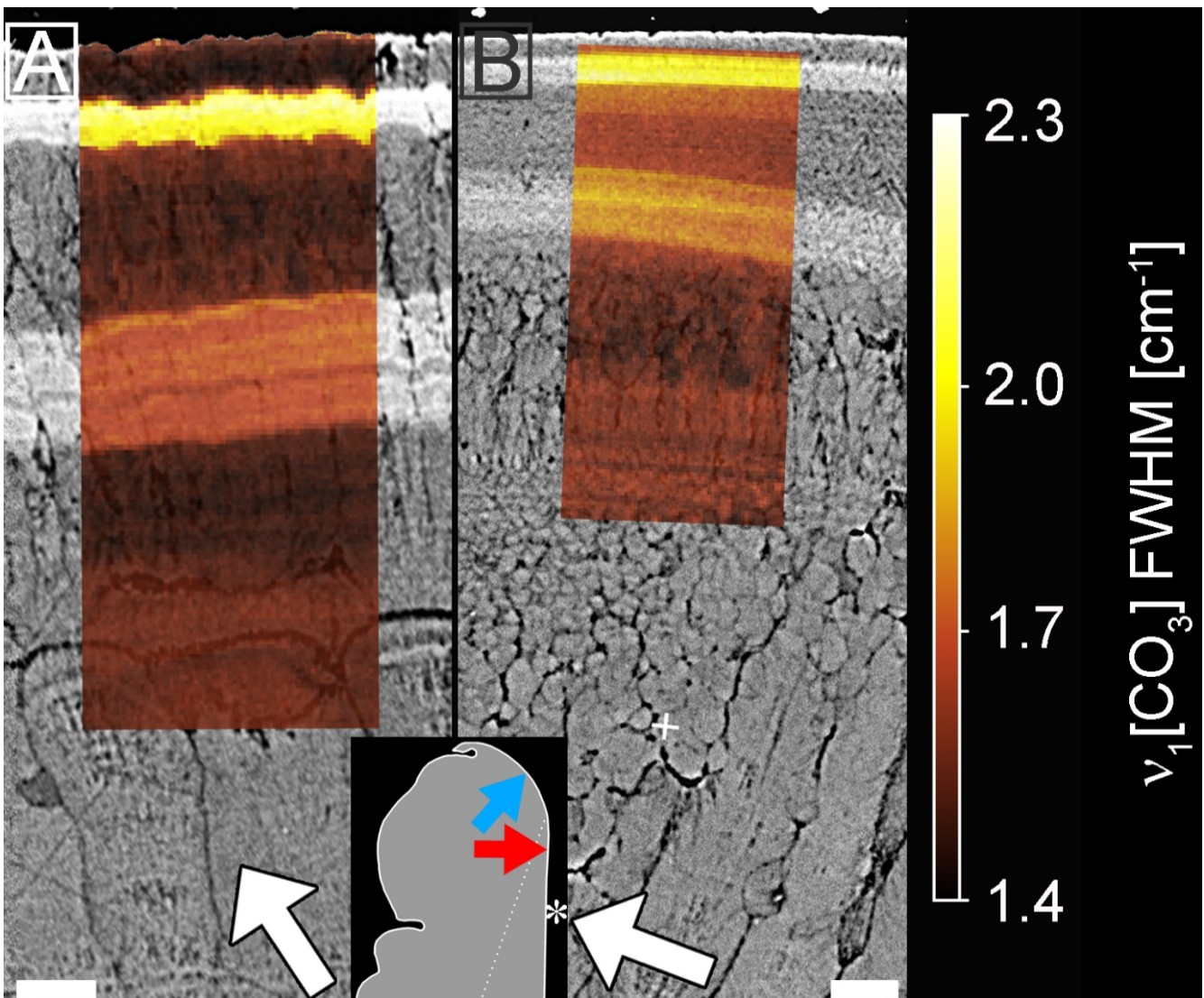

**Figure 3: Micro-Raman maps (sample K2-04) showing the effect of Sr concentrations on the FWHM of peak ν1 at 1084.8 cm⁻¹ in the oOSL (A) and iOSL (B). Raman maps are overlain on BSE images. White arrows point towards the general growth direction of the shell, while the Sr-labelled shell layers from the underlying BSE image visualise the local growth directions for each ultrastructure. For Micro-Raman maps of peak shifts see Fig. S2. All values are bandwidth corrected after Váczi (2014). A schematic of the shell tip shows the exact location of the Raman maps with a blue and red arrow pointing towards the locations of the prismatic oOSL and crossed-acicular iOSL sampling location, respectively. Asterisk marks the inner shell surface. Scale bars are 10 µm.**

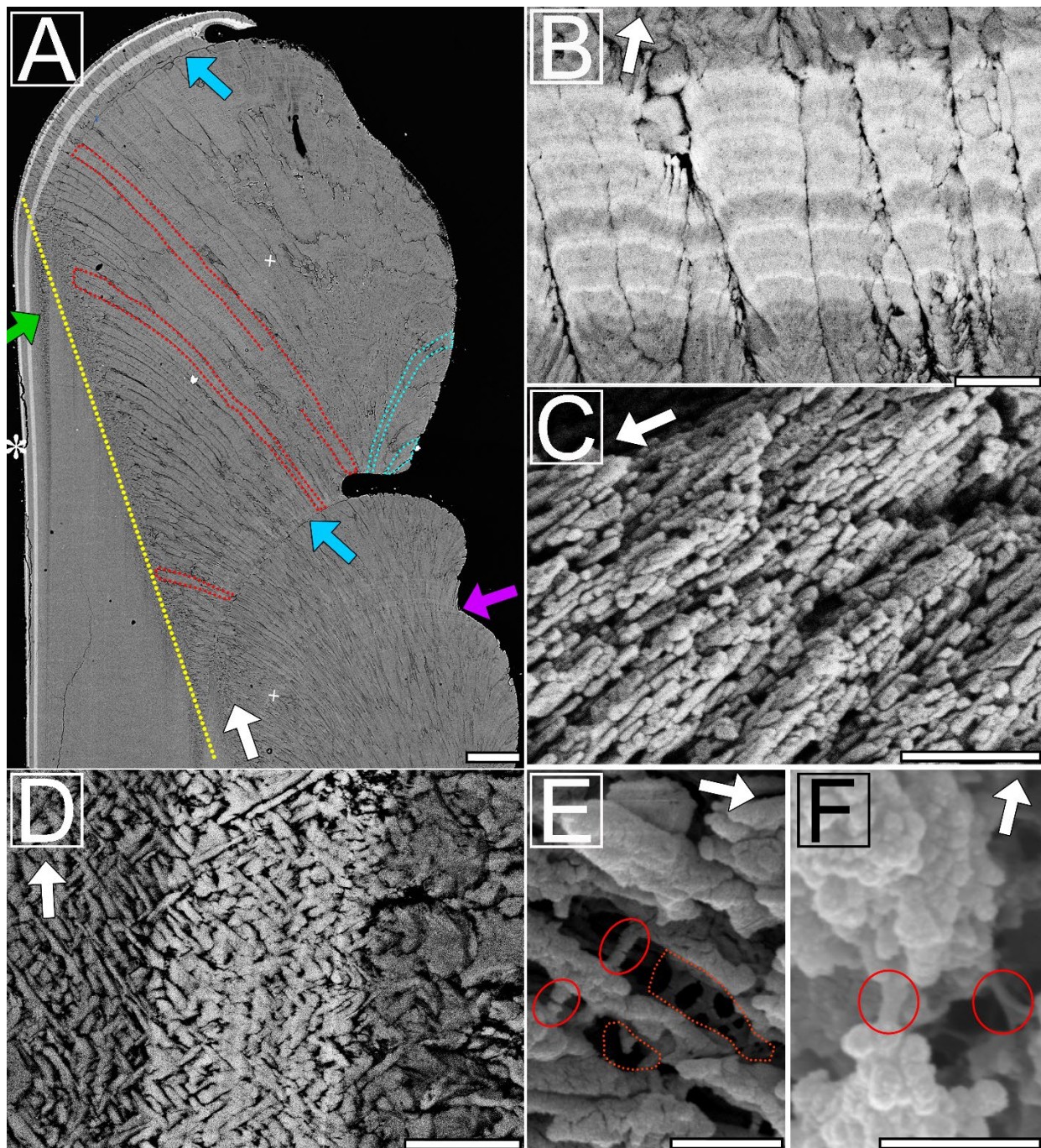

**Figure 4: Electron images showing a cross-sections along the maximum growth axis of Sr-labelled *K. rhytiphora* shells: (A) BSE image shows the ventral margin of the shell. First-order prisms in the prismatic oOSL bend inwards (red outlined) reach lengths of up to 700 μm with widths of 17 μm. Outward bending prisms (blue outlined) form the ridged surface ornamentation of the shell.**

Growth checks (blue arrows) are observed to occur directly at the end of ridge feature, while not all ridge features are concluded by growth checks (purple arrow). The yellow dashed line marks the boundary between iOSL and oOSL. Both Sr labels show bright greyscales and follow the growth front of the shell. In the iOSL, the growth check continues as a prismatic layer (green arrow). Strontium-labels within the oOSL (B) show first-order prisms to consist of radially arranged second-order prisms, which in turn consist of third-order prisms with their long axis parallel to each other, as seen in a broken piece of shell (C, SE-image). The iOSL has a crossed-acicular ultrastructure (D, BSE image) that is composed of needle-like lamellae intersecting at an angle of ca. 82°. Etched specimens (E, F: SE images) reveal the nano-granular texture of the mineral phase as well as organic compounds with fibre (red circles) and sheet-like structures (dashed red lines) in the prismatic (E) and crossed-acicular (F) layers. White arrows point towards the general growth direction of the shell, while the Sr-labelled shell layers in BSE images visualises the local growth directions for each ultrastructure. For more details see Fig. S3-S8. Scale bars: 100 μm (A), 5 μm (B and D), and 500 nm (C, E and F).

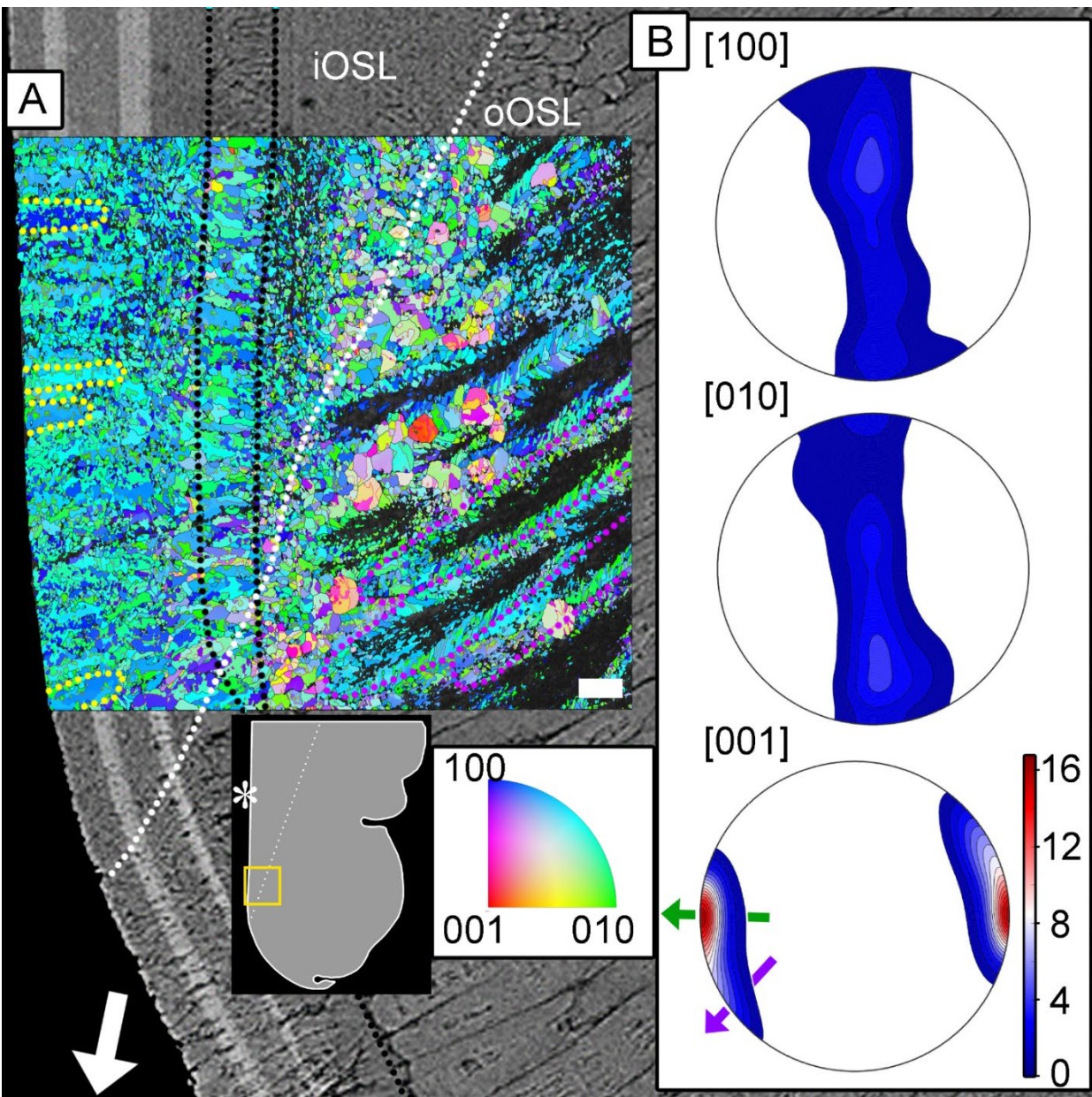

**Figure 5: Orientation map for aragonite (A) of a pulsed Sr-labelled shell (specimen ID: K2-11) overlain on the BSE image of the same area. The dotted white line indicates the boundary between the crossed-acicular iOSL and prismatic oOSL shell layer portions. The organic growth check in the oOSL that continues as a prismatic layer in the iOSL is highlighted with black dotted lines. Blue, green, and red colours depict the crystallographic a- [100], b- [010], and c-axes [001] of aragonite, respectively. Twinned grain boundaries are presented in red. The map is color-coded to show the crystallographic orientation normal to the image plane. Predominantly green and blue colours in the map indicate that the a-[100] and b-[010] axes are randomly aligned mainly normal to the image plane. First-order prisms in the oOSL (some outlined in purple) have unindexed cores, and feathery arranged second-**

order prisms are visible at their rims. Individual lamellae of the iOSL form co-oriented stacks up to 17 µm in size (circled in yellow). Pole figures (B) (lower hemisphere, equal area projection) show a strong clustering of the [001] axes for both layers. The local growth direction of the iOSL (green arrow), perpendicular to the light grey Sr-labelled layers in the underlying BSE image, differs by about 90° from the general growth direction (white arrow in (A)). The local growth direction of the oOSL (purple arrow) has a smaller angle with the general shell growth direction. The crystallographic a- and b-axes are randomly distributed in a plane normal to the local growth direction (i.e. parallel to the growth lines of the iOSL). Maximum density values of pole figures are colour-coded according to scale with the [001] axes achieving 16.8 times uniform. A schematic of the shell tip shows the location of the orientation map and the BSE image. Asterisk marks the inner shell surface. Scale bar is 10 µm.

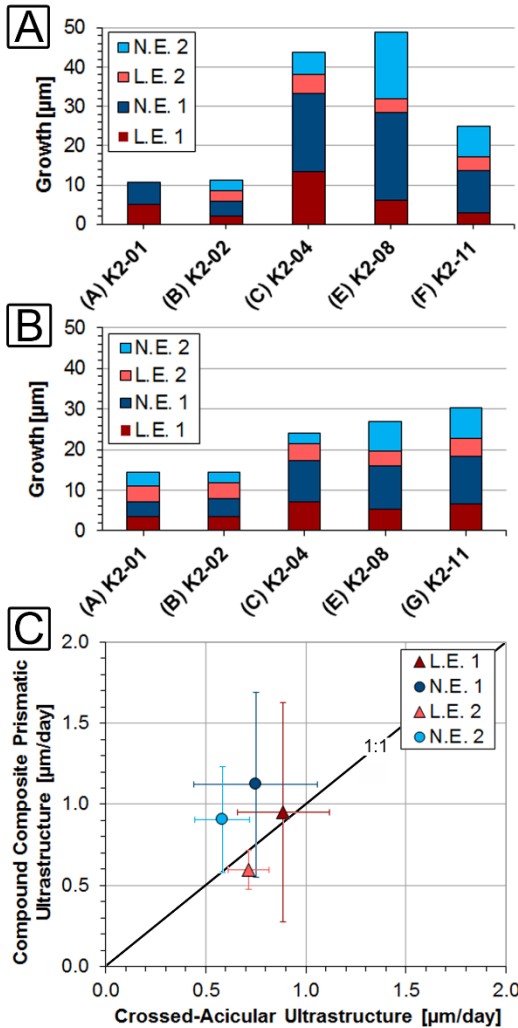

Figure: 6: Average growth of the compound composite prismatic (oOSL) layer (A) and crossed-acicular layer (iOSL) (B) (for values see Table 2). Distances were measured in triplicate at 5 different locations (Fig. S14, S15) along the axis of maximum growth using the software ImageJ. Local growth rates shown in (C) agree well within the first standard deviation between labelling and ambient conditions.

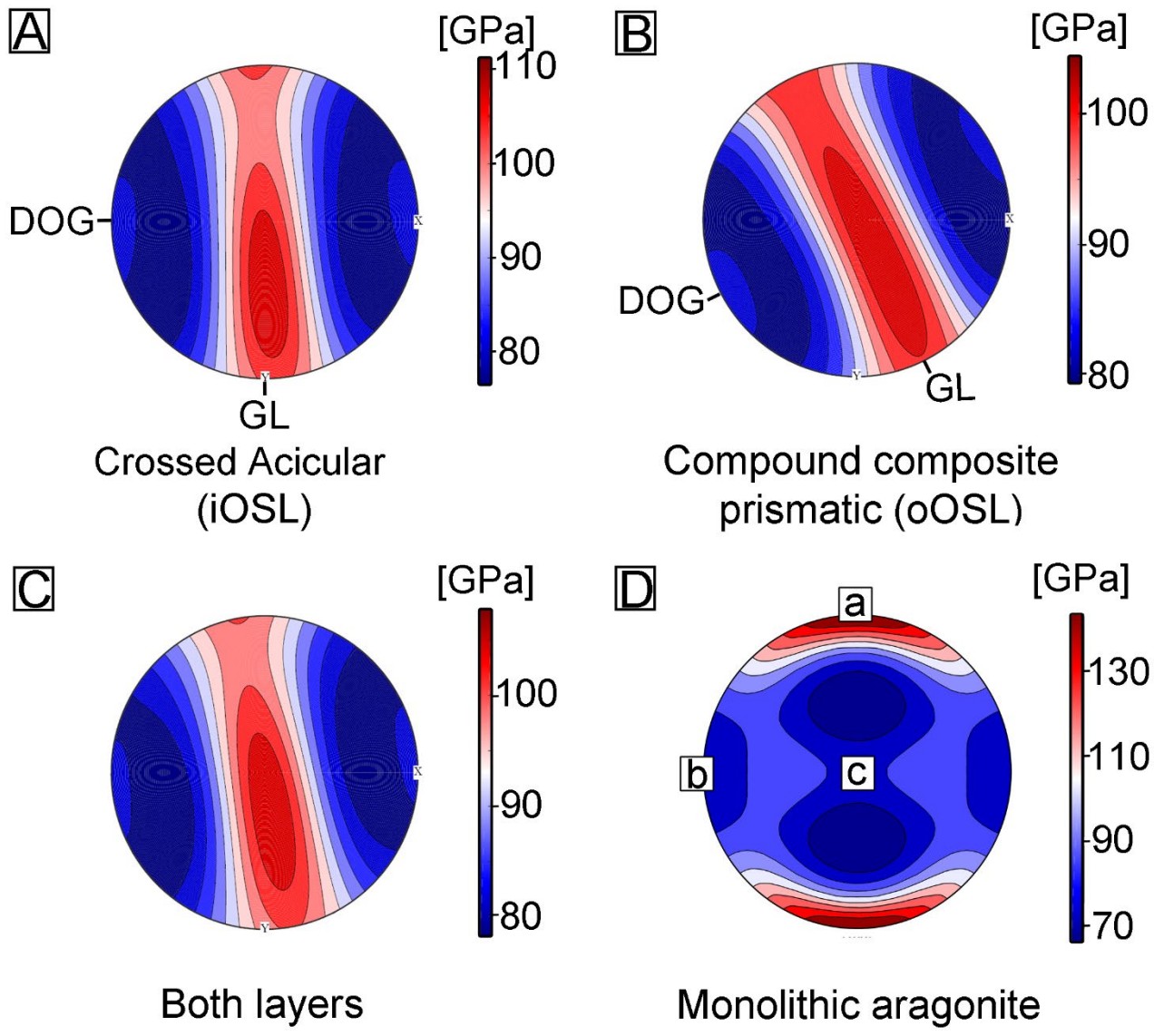

**Figure 7: Young's moduli (upper hemisphere and equal area projection), for the compound composite prismatic (oOSL, A) and crossed-acicular ultrastructure (iOSL, B) as well as for both layers together (C). Calculations were made with the Hill averaging scheme (colour scale on the right). (D) The Young's modulus for a aragonite single crystal is calculated with the Voigt–Reuss–Hill averaging scheme and is based on the elastic constants published in De Villiers (1971). We used the aragonite single crystal elastic properties of Liu et al. (2005) and the EBSD data collected from *K. rhytiphora* from this study as inputs (GL – local growth line, DOG – local direction of growth. Note the reference frame for (D) is given by the aragonite crystallographic axes.**

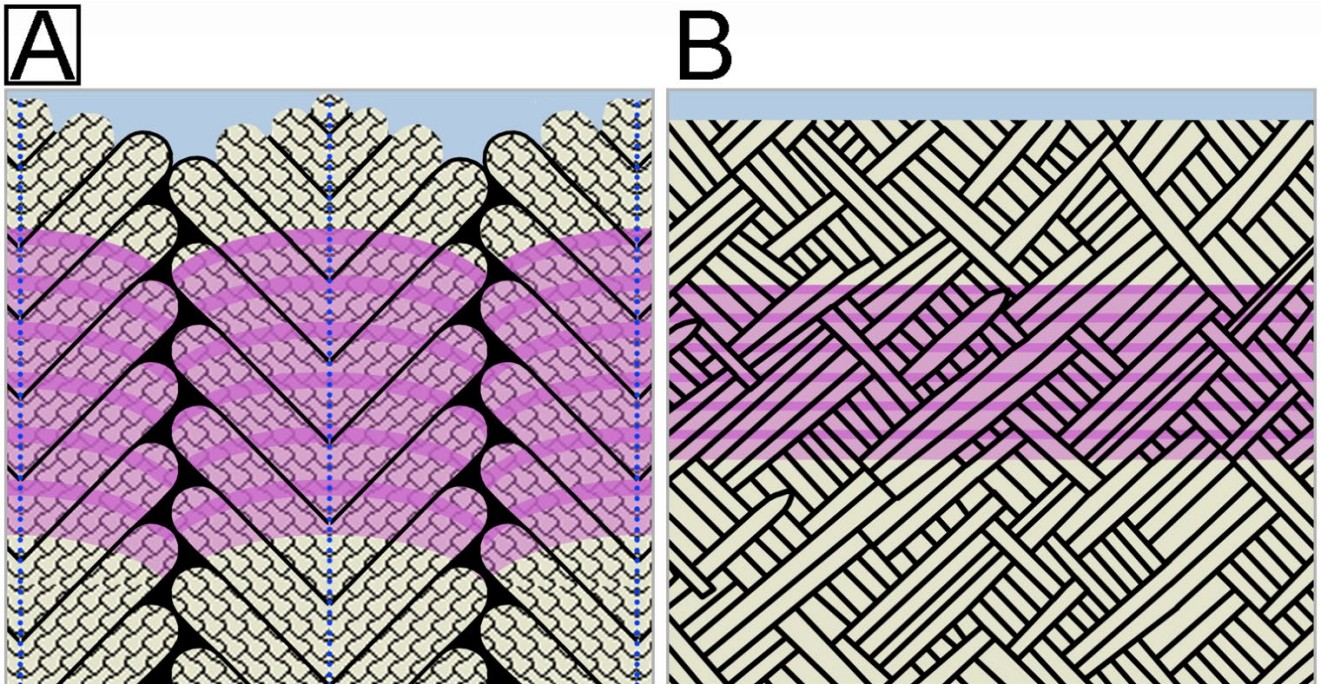

Figure 8: Schematic (not to scale) of the compound composite prismatic (oOSL, A) and crossed-acicular architecture (iOSL, B; modified after Bandel, 1977) transected by Sr-labels (purple) summarizing the observations in this study. Unlabelled aragonitic architectural units (beige) are outlined by organic sheaths (black). First-order prisms in (A) have thick organic sheaths, curved growth fronts, and consist of second-order prisms that are arranged at ca. 60° to the central axis of each first-order prism (A, blue dashed lines). Third-order prisms (A, tiled pattern) are oriented parallel to each other within second-order prisms. The shading in the pink Sr labels illustrates the internal BSE greyscale variations in the Sr labels reflecting variable Sr-concentrations within the Sr-label in the shells. The Sr-label is generally wider in the oOSL compared to the iOSL and transects all architectural units in both ultrastructures.

# Tables

**Table 1:** Geochemical composition of *K. rhytiphora* obtained from wavelength-dispersive X-ray spectrometry (WDS) electron probe micro analyser (EPMA) provided as $\mu g \cdot g^{-1}$ averages (Avg.) and standard deviations (Stdev.) as well as molar element/Ca ratios for shell compositions grown under different conditions in the wild ("pre-aqua"), in aquaculture during pulsed Sr-labelling ("LE 1" and "LE 2"), and non-labelling ("pre-NE 1" and "NE 1") periods.

| | | | Na | Mg | S | Cl | Ca | Sr | Na/Ca | Mg/Ca | Sr/Ca |
|---|---|---|---|---|---|---|---|---|---|---|---|
| **Compound composite prismatic** | Pre-Aqua | Avg. | 5,300 | 180 | 400 | 400 | 389,000 | 1,000 | 23.95 | 0.77 | 1.19 |
| | (n=5) | Stdev | 450 | 120 | 160 | 100 | 1,400 | 300 | 2.00 | 0.51 | 0.30 |
| | Pre-LE 1 | Avg. | 4,200 | 240 | 560 | 400 | 391,000 | 1,100 | 18.84 | 1.02 | 1.29 |
| | (n=3) | Stdev | 300 | 60 | 120 | 100 | 500 | 90 | 1.32 | 0.25 | 0.10 |
| | LE 1 | Avg. | 4,200 | 240 | 480 | 300 | 378,000 | 20,000 | 19.17 | 1.05 | 24.16 |
| | (n=3) | Stdev | 300 | 60 | 80 | 100 | 600 | 600 | 1.37 | 0.26 | 0.72 |
| | NE 1 | Avg. | 3,000 | 180 | 680 | 300 | 391,000 | 1,350 | 13.56 | 0.76 | 1.58 |
| | (n=3) | Stdev | 70 | 180 | 200 | 200 | 3,100 | 90 | 0.33 | 0.76 | 0.10 |
| | LE 2 | Avg. | 4,800 | 240 | 480 | 200 | 385,000 | 19,000 | 21.86 | 1.03 | 23.04 |
| | (n=3) | Stdev | 300 | 60 | 120 | 100 | 1,100 | 200 | 1.35 | 0.26 | 0.20 |
| **Crossed-Acicular\*** | Pre-Aqua | Avg. | 5,600 | bdl | 200 | 200 | 386,000 | 900 | 25.12 | bdl | 1.10 |
| | (n=5) | Stdev | 670 | - | 240 | 100 | 400 | 300 | 3.01 | - | 0.40 |
| | Pre-LE 1 | Avg. | 5,700 | bdl | 200 | 200 | 384,000 | 900 | 25.91 | bdl | 1.01 |
| | (n=3) | Stdev | 670 | - | 120 | 200 | 500 | 300 | 3.03 | - | 0.40 |
| | LE 1 | Avg. | 5,600 | bdl | 520 | 300 | 380,000 | >12,000 | 25.52 | bdl | >14.55 |
| | (n=3) | Stdev | 670 | - | 280 | 100 | 400 | 300 | 3.06 | - | 0.41 |
| | NE 1 | Avg. | 5,300 | 180 | 800 | 200 | 390,000 | 1,300 | 23.89 | 0.77 | 1.49 |
| | (n=3) | Stdev | 670 | 300 | 280 | 200 | 400 | 300 | 2.99 | 1.28 | 0.40 |
| | Limits of Detection: | | 400 | 100 | 100 | 100 | 300 | 200 | - | - | - |

Mn, Ba, P, K, and Fe, were analysed and were always below detection limits (200 $\mu g \cdot g^{-1}$ for Mn, Ba, Fe and 100 $\mu g \cdot g^{-1}$ for P, K). *LE2 and NE 2 in the crossed-acicular ultrastructure (iOSL) were too close to the edge and are excluded, LE1 is a minimum value as the analysed area slightly exceeds label width. See Table S2 for data in wt.% ($g \cdot g^{-1}$) oxide.

**Table 2**: Average daily local growth rates from pulsed Sr-labelling experiments. Rates in bold in NE2 were formed within 6 days (K2-01 to K2-04), all other rates in this column are within 12 days (K2-06 to K2-11). Average daily local growth rates for the entire experimental period are 0.85 (oOSL) and 0.73 μm (iOSL) For full details lists of all measurements see Tables S3 and S4.

| Sample ID: | Shell layer | LE 1 [μm/d] | NE 1 [μm/d] | LE 2 [μm/d] | NE 2 [μm/d] | Total growth period [μm/30d], [μm/36d] | Daily growth period [μm/d] |
|---|---|---|---|---|---|---|---|
| K2-01* | oOSL | 0.85 ± 0.10 | 0.48 ± 0.05 | n.a. | n.a. | **10.8 ± 1.1** | 0.66 ± 0.08 |
| | iOSL | 0.58 ± 0.03 | 0.31 ± 0.03 | 0.67 ± 0.03 | *0.55 ± 0.03* | **14.5 ± 0.9** | 0.53 ± 0.03 |
| K2-02 | oOSL | 0.37 ± 0.07 | 0.31 ± 0.04 | 0.43 ± 0.08 | *0.47 ± 0.05* | **11.3 ± 1.6** | 0.39 ± 0.08 |
| | iOSL | 0.60 ± 0.05 | 0.36 ± 0.03 | 0.65 ± 0.03 | *0.43 ± 0.05* | **14.4 ± 1.1** | 0.45 ± 0.05 |
| K2-04 | oOSL | 2.22 ± 0.15 | 1.67 ± 0.05 | 0.80 ± 0.12 | *0.97 ± 0.13* | **43.9 ± 2.9** | 1.41 ± 0.15 |
| | iOSL | 1.17 ± 0.32 | 0.85 ± 0.08 | 0.72 ± 0.07 | *0.43 ± 0.12* | **24.0 ± 3.9** | 0.79 ± 0.17 |
| K2-06 | oOSL | 0.63 ± 0.20 | 0.88 ± 0.14 | 0.58 ± 0.12 | 1.03 ± 0.13 | 30.1 ± 1.3 | 0.78 ± 0.12 |
| | iOSL | 1.00 ± 0.10 | 1.13 ± 0.05 | 0.92 ± 0.03 | 0.84 ± 0.03 | 35.1 ± 1.8 | 0.97 ± 0.05 |
| K2-08 | oOSL | 1.07 ± 0.30 | 1.86 ± 0.13 | 0.58 ± 0.15 | 1.41 ± 0.04 | 49.2 ± 1.2 | 1.23 ± 0.13 |
| | iOSL | 0.87 ± 0.08 | 0.89 ± 0.03 | 0.62 ± 0.05 | 0.61 ± 0.03 | 26.9 ± 1.4 | 0.75 ± 0.08 |
| K2-11 | oOSL | 0.46 ± 0.12 | 0.90 ± 0.11 | 0.60 ± 0.15 | 0.64 ± 0.11 | 24.9 ± 4.2 | 0.65 ± 0.11 |
| | iOSL | 1.12 ± 0.05 | 0.97 ± 0.17 | 0.73 ± 0.03 | 0.63 ± 0.03 | 30.3 ± 0.9 | 0.86 ± 0.02 |
| **Av. oOSL** | | **0.93 ± 0.15** | **1.02 ± 0.09** | **0.60 ± 0.12** | **0.76 ± 0.08** | **28.4 ± 2.1** | **0.85 ± 0.11** |
| **Av. iOSL** | | **0.88 ± 0.10** | **0.75 ± 0.04** | **0.72 ± 0.05** | **0.47 ± 0.03** | **24.2 ± 1.7** | **0.73 ± 0.07** |

*This individual did not show prismatic growth after NE1, while the crossed-acicular ultrastructure kept growing.

**Table 3**: Distribution coefficients of Ca and Sr between shell and seawater for both ultrastructures as well as for pulse Sr-labelled and unlabelled conditions. Concentrations for Ca and Sr in shell are from Table 1, while seawater values are mean ocean water.

| Distribution coefficients: | Compound composite prismatic (oOSL) ultrastructure | Crossed-acicular (iOSL) ultrastructure |
|---|---|---|
| $D_{Sr}$ labelled (Sr/Ca labelled shell)/(Sr/Ca seawater) | 0.15 | >0.09* |
| $D_{Sr}$ unlabelled (Sr/Ca unlabelled shell/(Sr/Ca seawater) | 0.14 | 0.14 |
| $D_{Sr}$ natural environment (Sr/Ca natural shell/(Sr/Ca seawater) | 0.13 | 0.12 |

*Minimum value as the analysed area slightly exceeds label width.

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
