# Peer review of "S1. Supplementary Figures"

_Biogeosciences, 2018_

## Referee Comment (RC1) · Anonymous Referee #1 · 26 Nov 2018

Otter and colleagues exposed specimens of a veneroid bivalve from Australia to episodically strongly elevated Sr levels (18 times above normal marine levels) in order to make the shell growth visible. They studied the effect of high Sr levels in the water on shell ultrastructure, crystallographic orientation, shell chemistry and growth rate. Except for the shell chemistry, all above mentioned shell properties remained unchanged. Sr/Ca values in the shell increased proportionately to that in the water, i.e., ca. 18 times, which still is way below expected thermodynamic equilibrium, a result supporting previous studies. Findings were interpreted to indicate an "intracellular, diffusion driven, selective transport" of ions across the mantle epithelium and subsequent shell formation processes via amorphous calcium carbonate.

[Figure]

The experiment and analyses were superbly executed and I really enjoyed reading the results. A broad variety of different machines (EBSD, nanoSIMS, $\mu$Raman spectroscopy, EPMA, TGA, optical microscopy and FEG-SEM) were employed to study physical and chemical properties of the shells. Yet, the study contains a number of flaws that need to be addressed in a significantly revised version of the ms.

(1) Authors need to specify the overarching goals of their study more clearly and formulate specific hypotheses. For example, I do not think that the main goal was just "to visualize growth" with Sr labeling as stated in the first (= most important) sentence of the Abstract. The title lists at least two other topics. In contrast to the great data presented in this manuscript, the Abstract and Introduction are very weak, poorly structured and organized, and the overarching (and far-reaching) purpose of the study remains elusive. The text is full of juxtapositions, i.e., sentences and paragraphs need better transition. In the Abstract, actual numbers of key data must be given, i.e., the 18 times enrichment in the shell (at least in the outer portion thereof; see below) following exposure to 144 $\mu$g/g Sr instead of 8$\mu$g/g (translate these data into molar Sr/Ca ratios, please). In the Introduction, authors should first place their study into broader context and identify the motivation for this investigation (which is not that existing in-situ staining methods affect the physiology of bivalves! See below). They need to describe open research questions and how they were addressed here. At the end of the Introduction and later in the Conclusions section, authors need to describe the implications of their finding, e.g., that bivalves likely serve as faithful recorders of the ocean chemistry etc. (which essentially emerges from the observation that Sr/Cashell changes proportionately to Sr/Cashell if the Srwater level is increased, or, as the authors expressed it – an interesting point of view by the way – irrespective of the Sr level of the water, Cashell/Cawater and Srshell/Srwater remained the same).

(2) Authors erroneously speak of outer and inner shell layer, but, in fact, they have only studied the outer shell layer, which in almost all bivalves is divided into two ultrastructurally different portions, i.e., the outer and inner portion of the outer shell layer (in the

following, oOSL and iOSL). The inner shell layer (ISL) is located way back (below what is depicted in Fig. 4C) and (in a cross-sectioned shell) starts where the myostracum intersects with the inner shell surface (= aka pallial line) and ends somewhere at the hinge portion. In Figure 1B, the inner shell layer is formed approx. inside the brown areas, whereas the brown section and portions outside thereof largely belong to the iOSL; the oOSL is likely not seen in this image. The pallial line delimits the ISL from the iOSL. I recommend to look at Fig. 2A in Schöne (2013).

(3) Surprisingly, a number of relevant recent papers dealing with very similar issues remain uncited. For example:

(3a) In-situ labeling: Mouchi et al (2013) labeled oysters with manganese to study growth rates, and Mouchi et al. (2016) used immunogold to obtain insights into biomineralization processes of Crassostrea gigas. Riascos et al (2007) tested three different stains in abalone and the surf clam, i.e., calcein, alizarin and strontium chloride.

(3b) Zhao et al (2017a) recently demonstrated that Sr/Ca in the outer shell layer of Corbicula fluminea increases proportionately to Sr/Ca in the ambient water and is not affected by growth rate effects. A very similar finding as reported here.

(3c) An alternative mechanism of how the bivalve controls the trace and minor element levels in the shell – brought forward by Shirai et al. (2014) and based on Stephenson et al. (2008) – was also ignored: Organic macromolecules near the shell formation front exert control on which and how many ions are incorporated into the carbonate phase of the shells. If the overall production of biomass and thus growth rate decreases (e.g., during times of low food availability), less of such organic substances are produced and the level of trace impurities in the shell carbonate automatically increases. This in turn, affect the morphology of biominerals and likely explains the more primitive ultrastructure at growth annual and even daily growth lines (biochecks) (Füllenbach et al. 2017), i.e., irregular simple/spherulitic prismatic ultrastructure (Schöne 2013). Data in Table 1 also indicate that different microstructures in your study contain different Sr

levels, likely for the very reason described above. However, you did not discuss this or the fact that the relative change in the iOSL is only ca. 14 times, not 18.

(4) The alternative mechanism of element incorporation mentioned in 2c does not require any control on uptake of elements. Although the chemical composition of the extrapallial fluids or gels (outer EPF forming the OSL, inner EPF the ISL) of marine bivalves have rarely been measured, the few available studies (e.g., Wada & Fujinuki 1976) unequivocally show that they have nearly the same ionic strength and chemical composition as the ambient seawater (Crenshaw 1972, Lorens & Bender 1980). This is no surprise, because bivalves are osmoconformers, like all other marine organisms. Imagine which energetic efforts were otherwise required if the bivalves had to constantly pump these ions out of the body fluids. Some elements such as strontium, magnesium and sodium reach the body fluids as ions from the ambient water through the gills and the gut (Wilbur & Saleuddin 1983) and across the mantle epithelium (passive diffusion). I have prepared a table for you summarizing data from Wada & Fujinuki (1976) (Table 1).

Table 1 see extra file

Despite this, shells are strongly depleted in many trace and minor elements. For example, if measured with a spatial resolution of ca. $50\mu$m Sr/Ca in aragonitic OSL of Arctica islandica ranges between ca. 1-3 mmol/mol and Mg/Ca remains below 0.8 mmol/mol (e.g., Schöne et al. 2011). Even when measured by much higher spatial resolution (nanoSIMS) which might be advantageous given the strong chemical heterogeneity of the shell at the $\mu$m-scale, Sr/Ca in aragonite of Cerastoderma edule does reach values expected for equilibrium fractionation (Füllenbach et al., 2017). In calcitic shells of various species, Mg/Ca ranges between ca. 4-28 mmol/mol (see summary in Vihtakari et al. 2016). These findings lend support to the hypothesis that unwanted elements are actively excluded from the shell by specialized organic macromolecules directly at the site of shell formation (Schöne 2013; Shirai et al. 2014). How this mechanism fits to the ACC-mediated shell formation processes needs to be discussed.

Since the chemistry of body fluids of bivalves resembles that of seawater, there is no need for any active transmembrane element transport. Zhao et al. (2017b) recently demonstrated very clearly that Sr, Mg and Ba levels in shells of Corbicula fluminea were not transported by active transport mechanisms and did not use the same pathways as Ca. These authors have poisoned Ca2+ATPase and blocked Ca2+ channels. According to the finding by Zhao and colleagues, a passive diffusion pathway across the mantle epithelium is much more likely and would perfectly fit to the incorporation control by organic macromolecules at the shell formation front. I strongly feel that these alternative explanations must be presented and discussed.

(5) Another argument against ATP-mediated uptake mechanism is unchanged growth rate of the bivalve. If the hypothesis by Otter and colleagues holds true according to which an "intracellular, diffusion driven, selective transport" of ions is responsible for the observed low Sr shell concentrations, then it is surprising that shell growth rate remained unchanged. A selective transport consumes energy = ATP), and the energy demand for such a transport process increases if the Sr level in the water rises. If more energy is devoted to the control of Sr incorporation into the shell, less energy is available for shell formation resulting in lower growth rate.

(6) There is a confusing usage of the term "uptake" (e.g., P2L8). 'Uptake' refers to way elements take from the environment to body fluids. This can either occur through mantle epithelia (in ionic form, potentially by one of the pathways listed in your paper) or during digestion of food. Is this really what you mean here on page 2 or rather the 'incorporation' of elements into the shell at the site of shell formation? From the context, I assume you meant the latter: "Recent studies showed that the uptake of some trace elements, such as strontium, are strongly influenced by crystal growth rates, shell curvature and ontogeny in addition to physiological effects".

(7) A number of observation were only presented, but not discussed and combined with other aspects of the study, e.g., different amounts of organics in different ultra-structures.

(8) Interpretation of the timing of shell growth, meaning of microgrowth increments (= daily), major biochecks (= annual) and greyscale changes (= fortnights) is purely speculative and not supported by the data presented. This would require mark-and-recovery experiments. Though not unlikely that the regular change in greyscale results from fortnightly changes, you need to cite at least relevant papers dealing in detail with such tide-controlled growth patterns (Evans 1972, Ohno 1989, Schöne 2008, Hallmann et al. 2009). B the way, you did not say where the bivalves lived: in the intertidal zone?

You also noticed that you observed 6 lines in portions formed in tanks during 6 (solar) days suggesting that at least these growth patterns are circadian. However, you have no evidence that the same applies to shell portions formed in nature. Given that the specimens lived in the intertidal zone (please provide details on tidal regime: diurnal or semidiurnal, tidal range etc.), it is reasonable to assume that they have formed circalunidian growth patterns (lunar days). Perhaps, acclimatization to circadian lab conditions were sufficient to reset biological clock resulting tin switch from lunar to solar daily. However, all this needs some discussion (in the Discussion section, not results as currently presented).

(9) Since you are aiming to publish your paper in a journal that is often read by people of the proxy and paleoclimate communities, you need to translate oxide values into element concentrations (as well as molar element/Ca data), and all element/Ca data into molar ratios (required for easier, direct comparison with published data). Likewise, instead of reporting Ca/Sr ratios, please turn this around and give Sr/Ca data.

(10) I do not think your results allow any conclusions on whether higher Sr levels in water have or have not affected shell growth rate. If growth conditions remained invariant (aside from changing Sr levels), shells should have grown much more homogeneously. But in fact, there is a significant slowdown from LE1 over NE1, LE2 to NE2 suggesting that growth conditions deteriorated through time (Table 2).

Other issues:
- Please check orthography in entire ms. I am not familiar with the Australian English, and whether this represents a mix of American English (e.g., analyze, labeling, meter) and British English (analyse, labelling, metre).

- Consistent use of hyphenation is required in entire ms: crossed-lamellar, cross-acicular, 3 mm-thick, high-resolution, crossed-lamellar, crossed-acicular, organic-rich etc. need a hyphen

- Headings: Consistently capitalize heading or use sentence case.

- No colon at the end of headings! E.g., P8L21: "The inner crossed[-]acicular [shell] layer:", P9L1, etc.

- P1L16, "aragonite crystals": As you noticed in the following sentence, "the smallest mineral units are nanogranules" which are enveloped by proteinaceous materials. I suggest to employ the term "mesocrystals", because the definition of an abiogenic aragonite crystal does not include nanocomposites consisting of aragonite and organic material.

- P1L19, replace "shells" by 'shell portions' or 'ultrastructures'. There are no bivalves consisting entirely of nacre. I assume you intended to say that different ultrastructures contain different amounts of organics.

- P1L19/20: I do not understand this sentence. Growth rates = growth patterns? Outer structure = outer shell layer. Prisms can be correlated to growth rates? Do you mean that each 3rd order prism forms in one day? Moreover, you did not mention anywhere in the text sub-daily growth patterns.

- P1L20, "outer structure": You used the term "structure" in two different ways: as a synonym for "ultrastructure" and "shell layer" (e.g., P6L32). Be consistent. Do not use "structure", but one of the other terms above. Check and change throughout ms.

- P1L23, "physiological processes during calcification have no lag": Rephrase, this is hard to understand. Shells do not just consist of $CaCO_3$, but also organics which need

to be fabricated, and the building blocks for these substances derive from ingested food. Digestion of food and fabrication of organic molecules that end up in the shell need time. There is hence a lag between ingestion of food and shell production. Or what do you mean with "physiological processes . . . have no lag".

- P1L23, "calcification" is the wrong term here (and used improperly in many other studies). Calcification rate includes density and is not synonymous to growth rate! Calcification rate = amount CaCO3 precipitated per time interval per area. Replace all instances with 'shell growth rate'

- P1L25, "Sr-conditions": no hyphen; 'Sr level' or 'Sr concentration' sounds better

- P1L26, "Sr-enrichment": no hyphen

- P1L26, "Sr-enrichment factors for labelled and ambient conditions": This remains insufficiently explained and is oddly phrased. Do you mean artificially elevated Sr levels vs. normal marine Sr levels? Give actual numbers! What do you mean with "identical enrichment factors": Sr levels in shell increase proportionately to that in the water (i.e., 18 times)? As far as I can tell from Table 1, this does not apply to both shell layers (and ultrastructures).

- P1L31, "aragonite or calcite": replace "or" by 'and/or'. Note there are species with different CaCO3 polymorphs in the outer and inner shell layers. Further note that some species also come with vaterite, ref

- Introduction: better transition between paragraphs needed

- P2L3: delete "recent and fossil", superfluous

- P2L4+5: None of these papers used trace elements of shells as environmental proxies. Replace by suitable citations:

(a) temperature: Klein et al. (1996a), Wanamaker et al (2008), Schöne et al. (2011), Zhao et al. (2017a)

(b) salinity: Klein et al. (1996b)

(c) pH: Zhao et al. (2017c)

- P2L5: "uptake" refers to element uptake from the environment either through mantle epithelia (in ionic form, potentially by one of the pathways listed in your paper) or during digestion of food. Is this really what you mean here or rather the 'incorporation' of elements into the shell at the site of shell formation? (see main comments above)

- P2L10: "Urey et al., 1951" is neither a "recent" study nor a study that looked at trace elements. One reference that must be added here is Shirai et al. (2014) which discussed another potential mechanism that controls Sr incorporation into the shell (see main comments above).

- P2L12: substitute "shell" with 'trace and minor elements in shells'

- P2L14: substitute "but" with 'and'

- P2L14: Firstly, always say 'ultrastructure', not "structure", because at other places you use "structure" as a synonym for shell layer. Secondly, this statement needs a reference.

- P2L15-16: Delete sentence starting with "Apart...". Then start next sentence with "Apart from those,"

- P2L17: replace "which are found" by 'which occur'

- P2L21: The homogeneous ultrastructure forms an own category and is not a subgroup of the crossed-acicular category (compare Marin et al. 2012)

- P2L21: "venerid" must not be italicized

- P2L22: "Shimamoto, 1986" is outdated (?), check most recent revision of ultrastructures by Carter JG et al. (2012)

- P2L24: "Pulsed strontium labelling ... understanding of other marine calcifiers": You

have used exposure to higher Sr levels not only to study the effects of this trace element on the ultrastructure and as a time gauge, but primarily to identify potential mechanisms of element incorporation into shells. And the latter has been already done with bivalves by Zhao et al (2017a).

- P2L28: delete hyphen after "micro"

- P2L33: replace "between umbo and ventral margin" by 'parallel to the main growth axis' or 'parallel to the umbo-ventral margin axis'

- P3L5-6: "Growth lines. . ." show/refer to figure

- P3L7: crossed-acicular ultrastructure is not a subcategory of the homogeneous ultrastructure. The latter forms an own category. Refer to more recent studies (Marin et al. 2012, Carter et al. 2012).

- P3L16: Two main clauses combined by conjunction require comma; check and correct throughout ms: ', and'

- P3L17: Specimens: Much more information needed here: sediment type, tidal height, intertidal zone(?), how many specimens collected/prepared/used for which analytical technique, when collected. Table would be best. Part of this information is relevant for the temporal alignment of the shell growth patterns.

- P3L17: replace "live-collected" by 'collected alive'

- P3L20: use '$\times$' as mathematical operator (consistently throughout ms)

- P3L23: "which is a reliable sign for the absence of handling stress"... says who? This claim is unsupported. And looking at the decline in daily shell growth (Table 2) during the experimental phase, the specimens do not seem to have liked the new environment. So, handling stress cannot be precluded.

- P3L24-26: Has the element composition of the food been measured as well? How do you know that all Sr and Ca comes from the water? Has always the same amount

of food being offered? When were they fed, during simulated day or nighttime?

- P3L29-30: An "event" is a very short-term incident. This sentence should be rephrased, e.g., "exposure to background conditions, i.e., normal marine Sr levels".

- P4L7: P400-P2000

- P4L12: thickness of gold-coating?

- P4L21: 20,000×

- P5L9: replace "was used" by "were used"

- P5L21: $\mu$m2 (superscript)

- P5L27 "The inner and outer layer of a K. rhytiphora shell were separated with a DREMEL tool and mechanically cleaned." Be more specific here: Have you obtained powdered material or fractions of the two portions of the outer shell layer? How have you managed exactly to separate them?

- P6L3: Actually wrong. You have only studied the outer shell layer, which consists of two portions with different ultrastructure, an outer and inner portion, respectively (oOSL, iOSL)!

- P6L8: Rephrase (and italicize genus and species names): 'The outer shell layer of studied K. rhytiphora specimens is ... near the ventral margin'

- P6L10: "in agreement with previous studies..." This phrasing means that the other species studied by Carré and Soldati and colleagues lived in Australian waters. Rephrase.

- P6L11: "growth periods": delete "periods"

- P6L13: "troughs" Odd phrasing. Something like this is better: 'Cyclic changes in greyscale near the ventral margin correlate strongly with tidal cycles, i.e., light grey and dark grey portion fall together with full and new moon cycles, respectively.' The main

problem is that you do not provide any evidence for the timing of shell growth! Where is the evidence that the dark and light portions really have formed during new and full moon? This is an interpretation at most, and as such belong to the Discussion section (where you need to refer to previous studies of intertidal bivalves which found narrower increments and thicker growth lines formed during spring tides, and these portions then appear darker than shell portions formed during neap tides when viewed at lower magnification and under reflected light. More suitable Refs: Evans 1972, Schöne 2008, Hallmann et al. 2009)

- P6L16-25 also needs to be moved to Discussion. Only keep descriptive part here. You have no evidence that these grey bands formed on a circalunidian basis, but you can certainly interpret them as such based on previous work.

- Timing of shell growth: You later noticed that you observed 6 lines in portions formed in tanks during 6 days suggesting that at least these growth patterns are circadian. However, you have no evidence that the same applies to shell portions formed in nature. Given that the specimens lived in the intertidal (please provide details on tidal regime: diurnal or semidiurnal, tidal range etc.), it is reasonable to assume that they have formed circalunidian growth patterns. Here, please stick to descriptions, not interpretation.

- P6L25: 'in two other specimens', not "on two other specimens"

- Section 3.2: Title is more suitable for Discussion. – This section should be expanded as it is an essential component of the ms and forms the basis for your hypothesis on element incorporation. Describe Table 1 in much more detail. Report molar ratios as well. Compute and tell reader by how much the Sr levels increased in the shell when exposed to 18 times higher Sr levels in water. This will then show that the Sr levels in oOSL increased by 18 to 20 times, whereas the iOSL only by ca. 14 times. This needs to be discussed later.

- P6L27: "Sr incorporation": no hyphen - P6L28: "Sr concentration": no hyphen

- P6L32: replace "structure" by "shell layer

- P7L2: "…were identical within uncertainty": i.e., they have remained invariant, stayed the same? I suggest you rephrase this to avoid confusion.

- Title Section 3.3: Section heading should inform about content of section, not which method has been used.

- P7L31: "This species develops annual growth checks" On what evidence is this statement based? How did you analyze when the shell portions formed? Likely correct, but pure speculation... or is there previous work on this species?

- Section 3.4.3: Interesting information, but what is the purpose of having this measured and reported?

- P9L10: crystallographically

- P9 "Calcification Rates" includes density, not synonymous to growth rate! Calcification rate = amount CaCO3 precipitated per time interval per area; this is not what you mean.

- P9L28-29: "Due to the geometry of first order prisms without- and inward bending in cross-sections,…" No sentence

- P10L2: Provide image showing where you determined increment widths, or even better trace two growth lines to show that growth in oOSL is faster than in iOSL due to shell geometry.

- P10L4-5: quite complicated phrasing: absolute growth rates vary among specimens

- P10L5: grew, on average, 5.6... same for the other "on average": separate by comma and place before number

- P10L12: "Also, rates tend to decrease effectively with increasing distance to the ventral margins (Fig. 4A).": Unclear what you mean and purpose of mentioning this. You need to trace fortnights in Figure 4A to support your statement.

- "bivalve species": you listed genera not species rephrase: ... structure of other bivalves, e.g., Pinna..., the aragonitic...

- P10L23, P11L5, P11L23: "In K. rhytiphora the first order prisms" comma after species name

- P11L10: Equally-sized (adverbial usage)

- P11L30: Since you did not capitalize "aragonite", you should also use lower case here (except for the acronym/abbreviated form). Besides that, you used lower case in the Abstract.

- P12L16: replace "shell" by 'shell portions'

- P12L30: replace "the outside of their shells" by 'outer shell surface (Fig. 1), and'

- P13L1: "growth time": Firstly, you have no evidence that these growth checks formed annually. Secondly, no bivalve grows 365 days. Note also that such ornamentation patterns do not agree with growth patterns in other species, and likely this is a coincident and only true for shell portions near the ventral margin in the studied specimens. Rephrase.

- P13L12-13: Perfect! This is your time gauge. It verifies the circa daily nature of these growth features and could further be used to support your hypothesis of fortnightly growth bundles appearing as greyscale changes.

- P13L14-15: replace "higher" with 'faster', "short" with 'narrow', "longer" with 'broader'

- P13L15: "day": An interesting question that you need to discuss is that these are probably circadian (24h) periods entrained by the 12/12 light/dark cycle experimental conditions. The adjustment interval was probably long enough that the natural, tide-entrained shell formation cyclicality (resulting in circalunidian, 24.8h, periods) vanished. Under natural conditions though, you would need to have circatidal (12.4h) and circalunidian increments, because otherwise your interpretation of the other 48 or 50

dark cycles representing fortnight periods would not hold true.

- P13L22: "We suggest a diel physiologically controlled variation of calcification" Not sure exactly what you mean. Circadian clock controlling growth/calcification rate? This has been reported previously elsewhere.

- P13P29"physiological processes involving Sr incorporation", rephrase: 'physiological processes controlling Sr incorporation'

- P13L29 "have no lag"? Well, this depends on the temporal scale you are looking at. Where is the evidence that there was no gradual increase in shell Sr levels during the course of minutes or so? Diffusion of Sr through the mantle epithelium takes at least some time.

- P13L30-31: I do not think that the implications provided are supporting an ACC-mediated growth of shell in bivalves.

- P13L33-34: "A fundamental observation of this study is that the calcification front runs evenly across all structural units and architectural orders of the shell independently of the current growth rate. This" But this is known and no a new finding of this study!

- P14L1: "show the labels to cut across the different architectural building blocks": could also occur if extrapallial space is gel-filled or epithelial cells are in direct contact with shell

- P14L2 "where the label would rather follow a zig-zag trend between fully labelled and unlabelled units" Impossible to understand what you intend to say here. Rephrase please. Do you mean that the growth front is uncoupled from the ultrastructures? This is known as well: In freshwater bivalves the large prisms continue to grow over many years and daily growth lines cross them perpendicularly (studies by Dunca, Mutvei etc.).

- P14L2-4: "This is clearly visible from the sharply defined change between labelled and unlabelled shell areas (Fig. 4B and D), as well as from the cyclic variations in

short-term growth rates (discussed above). Our" Likewise hard to understand

- P14L10 "active selective transport consuming Ca2+-ATPase enzymes": transport consumes energy which is provided by ATP, and the enzyme that accomplishes the transportation is the Ca ATPase. Rephrase.

- P14L14-15 "We observed virtually identical enrichment factors for Ca and Sr (CaShell/CaSeawater and SrShell/SrSeawater) in labelled and ambient conditions (Table 3).": Interesting point of view! But this does not mean anything else than Sr/Ca shell increases proportionately to that of Sr/Ca seawater, and this has already been shown by Zhao et al. (2017), which you did not cite.

- P14L15: "Sr-ion transport is independent from... " if so, the energy demand of the bivalve increases in order to keep the Sr out of the shell. Do you see a decrease in growth rate during Sr enrichment as opposed to 'normal' Sr levels in water?

- P14L16-17: "Sr ion would be at the expense of a Ca ion": Not really clear what you mean; Since this is the essence of your paper, you need to describe this more clearly and convincingly. Why exactly can transport mechanism 1 not be true?

- P14L17: Replace "Sr-enrichment" by 'shell Sr concentrations'

- P14L18: Ca/Sr: please also or only report Sr/Ca

- P14L19-20: Replace "Hence, the strong enrichment of Ca from seawater to shell" by 'strong enrichment of Ca in shell'

- P14L25: "Ca to be transported as ACC-nanogranules to the calcification front (Loste et al., 2004; Addadi et al., 2006; Jacob et al., 2011; Zhang and Xu, 2013)." Check if all cited studied were using bivalves (not gastropods or other taxa), and which ultrastructures were analzed, report this here.

- Section 4.5: Here you discuss more (and different stuff) than what the heading implies.

- P15L12: italicize genus and species names

- P15L14-15: "a systematic change in growth increments during Sr-enriched periods": Do you mean 'growth increment widths'? You need to highlight here again that food levels and other extrinsic factors that could potentially have affected growth rate remained unchanged during the experiment, and you would have expected invariant increment widths if Sr had no effect on growth rate... see comment further above on relationship between growth rate and Sr exclusion from shell

- P15L18: Replace "calcification" by 'growth rate'

- P15L23-24: "Reduced growth rates in aquaculture conditions cannot be explained by ontogenetic trends alone but result from missing tidal cycles." Sorry, but this is pure speculation and likely wrong. Much more likely is that you did not provide proper food and the animals did not really 'like' the tank conditions.

- P15L30: 'nanometer'?

- More comments in pdf with annotated figures and tables.

References

Carter et al 2012. Treatise on Invertebrate Paleontology. Part N. Revised Vol 1. Chapter 31. Glossary of the Bivalvia. Treatise Online 48, 209p.

Crenshaw MA 1972. Inorganic composition of molluscan extrapallial fluid. Biol Bull 143, 506-512.

Evans JW (1972) Tidal growth increments in the cockle Clinocardium nuttalli. Science 176:416–417.

Füllenbach CS, Schöne BR, Shirai K, Takahata N, Ishida A and Sano Y 2017. Minute co-variations of Sr/Ca ratios and microstructures in the aragonitic shell of Cerastoderma edule (Bivalvia) – Are geochemical variations at the ultra-scale masking potential environmental signals? Geochimica et Cosmochimica Acta 205, 256-271.

Hallmann N, Burchell M, Schöne BR, Irvine GV and Maxwell D 2009. High-resolution sclerochronological analysis of the bivalve mollusk Saxidomus gigantea from Alaska and British Columbia: techniques for revealing environmental archives and archaeological seasonality. Journal of Archaeological Science 36, 2353-2364.

Klein RT et al 1996a. Bivalve skeletons record sea-surface temperature and $\delta$18O via Mg/Ca and 18O/16O ratios. Geology 24, 415-418.

Klein RT et al 1996b. Sr/Ca and 13C/12C ratios in skeletal calcite of Mytilus trossulus: Covariation with metabolic rate, salinity, and carbon isotopic composition of seawater. Geochimica et Cosmochimica Acta 60, 4207-4221.

Lorens RB and Bender ML 1980. The impact of solution chemistry on Mytilus edulis calcite and aragonite. Geochim Cosmochim Acta 44, 1265-1278.

Marin F et al 2012. The formation and mineralization of mollusk shell. Frontiers in Bioscience S4, 1099-1125.

Mouchi V et al 2013. Chemical labelling of oyster shells used for time-calibrated high-resolution Mg/Ca ratios: A tool for estimation of past seasonal temperature variations. Palaeogeography, Palaeoclimatology, Palaeoecology 373, 66-74.

Mouchi V et al 2016. Chalky versus foliated: a discriminant immunogold labelling of shell microstructures in the edible oyster Crassostrea gigas. Mar Biol 163:256.

Ohno T (1989) Palaeotidal characteristics determined by microgrowth patterns in bivalves. Palaeontology 32:237–263.

Riascos J et al 2007. Suitability of three stains to mark shells of Concholepas concholepas (Gastropoda) and Mesodesma donacium (Bivalvia). Journal of Shellfish Research, 26, 43-49.

Schöne BR, 2008. The curse of physiology – Challenges and opportunities in the interpretation of geochemical data from mollusk shells. Geo-Marine Letters 28, 269-

285.

Schöne BR 2013. Arctica islandica (Bivalvia): A unique paleoenvironmental archive of the northern North Atlantic Ocean. Global and Planetary Change 111, 199-225.

Schöne BR, Zhang Z, Radermacher P, Thébault J, Jacob D, Nunn EV & Maurer A-F 2011. Sr/Ca and Mg/Ca ratios of ontogenetically old, long-lived bivalve shells (Arctica islandica) and their function as paleotemperature proxies. Palaeogeography, Palaeoclimatology, Palaeoecology 302, 52-64.

Shirai K, Schöne BR, Miyaji T, Radermacher P, Krause RA Jr and Tanabe K 2014. Assessment of the mechanism of elemental incorporation into bivalve shells (Arctica islandica) based on elemental distribution at the ultrastructural scale. Geochimica et Cosmochimica Acta 126, 307-320.

Stephenson A. E., DeYoreo J. J., Wu L., Wu K. J., Hoyer J. and Dove P. M. (2008) Peptides enhance magnesium signature in calcite: insights into origins of vital effects. Science 322, 724–727.

Vihtakari M et al 2016. A key to the past? Element ratios as environmental proxies in two Arctic bivalves. Palaeogeogr Palaeoclimatol Palaeoecol 465, 316-332.

Wada K and Fujinuki T 1976. Biomineralization in bivalve molluscs with emphasis on the chemical composition of the extrapallial fluid. In: N Watabe and KM Wilbur (Eds), The Mechanisms of Mineralization in the Invertebrates and Plants. Univ South Carolina Press, 175-190. Wanamaker et al 2008. Experimentally determined Mg/Ca and Sr/Ca ratios in juvenile bivalve calcite for Mytilus edulis: implications for paleotemperature reconstructions. Geo-Mar Lett 28: 359–368.

Wilbur KM and Saleuddin ASM 1983. Shell formation. In: ASM Saleuddin and KM Wilbur (Eds), The Mollusca. Vol. 4. Physiology Part 1. Academic Press, New York, pp. 235-287.

Zhao L, Schöne BR and Mertz-Kraus R 2017a. Controls on strontium and barium

incorporation into freshwater bivalve shells (Corbicula fluminea). Palaeogeography, Palaeoclimatology, Palaeoecology 465, 386-394.

Zhao L, Schöne BR and Mertz-Kraus R, 2017b. Delineating the role of calcium in shell formation and elemental composition of Corbicula fluminea (Bivalvia). Hydrobiologia 790, 259-272.

Zhao L, Schöne BR, Mertz-Kraus R and Yang F 2017c. Insights from sodium into the impacts of elevated pCO2 and temperature on bivalve shell formation. Journal of Marine Experimental Biology and Ecology 486, 148-154.

Please also note the supplement to this comment:
https://www.biogeosciences-discuss.net/bg-2018-469/bg-2018-469-RC1-supplement.pdf

Table 1. Average element-to-Ca ratios in the inner extrapallial fluid of marine bivalves in comparison to seawater. Calculated from chemical data reported in Wada & Fujinuki (1976).

|  | Seawater | EPS during growth | EPS during resting |
|---|---|---|---|
| Na/Ca (mol/mol) | 44.3 | 44.1 | 42.4 |
| Li/Ca (mmol/mol) | 2.1 | 2.6 | 2.7 |
| Mg/Ca (mol/mol) | 5.0 | 5.1 | 4.9 |
| Sr/Ca (mmol/mol) | 8.3 | 9.4 | 8.0 |
| Mn/Ca (mmol/mol) | 30.2 | 291.1 | 223.9 |

**Fig. 1.**

**Supplement:**

**Figures and Figure Captions**

[Figure]

Figure 1: **Outside (A) and inside (B) view** of an unlabelled *K. rhytiphora* shell. Dashed black line in (A) indicates where the shell was sectioned. All cross-sections in this study are prepared as radial sections along the maximum growth axis unless otherwise specified. (C). Dark bands (arrows in C) **are** summer growth periods between lighter coloured winter periods and are magnified in D (red box in C) with a greyscale line profile. Distinct **troughs** in greyscale intensity correlate with 48 out of 50 spring tides in two

18

years from the collection site of the bivalves (full moon: light grey, new moon: dark grey) giving fortnightly growth resolution in this shell area. Greyscale line profiles (E) of the area marked by the red box in D shows the most recent shell growth increment in the wild (mid-August to mid-September 2016). In this part tides correlate with most shell increments (black dashed lines), while this correlation is lost after start of aquaculture (blue area). Blue area in D, E marks the aquaculture period with lower than

5    normal growth rates. Scale bars are 10 mm (A-B), 5 mm (C), 1 mm (D), 0.1 mm (E).

[Figure]

[Figure]

10    **Figure 2: FEG-SEM BSE images showing polished cross-sections of the outer (A) and inner (B) layers of a Sr-labelled *K. rhytiphora* shell (specimen ID: K2-06) overlain with NanoSIMS $^{88}$Sr/$^{40}$Ca maps. Ambient seawater $^{88}$Sr/$^{40}$Ca ratios are depicted in blue, while shell formed during Sr-enriched incubations are shown in pink.** Scale bars are 10 μm.

[Figure]

**Figure 3: Micro-Raman maps (sample K2-04) showing the effect of Sr concentrations on the FWHM of peak $v_1$ at 1084.8 cm$^{-1}$ in the outer (A) and inner layer (B). Raman maps are overlain on BSE image. Bright grey-scale areas in the BSE images show elevated Sr-contents correlating with peak broadening (FWHM increase, see scale in cm$^{-1}$) in the labelled shell areas. For Micro-Raman maps of peak shifts see Fig. S2. All values are bandwidth corrected after Váczi (2014). Scale bars are 10 μm.**

[Figure]

**Figure 4: Backscattered images showing cross-sections along the maximum growth axis of Sr-labelled *K. rhytiphora* shells: (A) shows the ventral margin of the shell. First-order prisms in the prismatic outer layer bend inwards (red outlined) reach lengths of up to 700 μm with widths of 17 μm. Outward bending prisms (blue outlined) form the ridged surface ornamentation of the shell. Organic-rich growth checks are observed to occur directly at the end of a ridge feature (blue arrows), while not all ridge features are concluded by growth checks (purple arrow). The yellow dashed line marks boundary between inner and outer shell layers. Both Sr-labels show bright grayscales and follow the growth front of the shell. In the inner layer, the growth check continues as a prismatic**

layer (green arrow). Strontium-labels within the outer layer (B) show first-order prisms to consist of radially arranged second-order prisms, which in turn consist of third-order prisms with their long axis parallel to each other, as seen in a broken piece of shell (C, Fig. S11). The inner, crossed acicular layer (D, BSE image) is composed of needle-like lamellae intersecting at an angle of ca. 82°. Etched specimens (E, F: SE images) reveal the nano-granular texture of the mineral phase as well as organic compounds with fibre (red circles) and sheet-like structures (dashed red lines) in the prismatic (E) and crossed acicular (F) layers. White arrows mark the general growth direction for each ultrastructure. For more details see Fig. S5-S8. Scale bars: 100 µm (A), 5 µm (B and D), and 500 nm (C, E and F).

[Figure]

**Figure 5: Orientation map for aragonite (A) of a pulsed Sr-labelled K. rhytiphora shell (specimen ID: K2-11). Blue, green, and red represent the crystallographic a- [100], b- [010], and c-axes [001] of aragonite, respectively. The map is color-coded to show which crystallographic axis is aligned parallel to the growth direction of the shell-layers. The dotted white line indicates the boundary between inner and outer shell layers. The organic growth check in the outer structure that transitions into a thin prismatic layer in the inner layer is highlighted with light blue dotted lines. First-order prisms in the outer structure (some green outlined) have unindexed cores, while feathery arranged second-order prisms are visible at their rims. Individual lamellae of the inner layer form co-oriented stacks up to 17 µm in size (circled yellow). Pole figures (B) (lower hemisphere, equal area projection) show a strong clustering of [001] axes for both shell layers and coincides with the growth direction of the inner layer (DOG Cr.Ac.), but is at an angle relative to the growth direction of the outer layer (DOG Pr. In B). Crystallographic a- and b-axes are randomly distributed in the plane normal to the growth direction (i.e. containing the growth lines of the inner layer). Maximum density values of pole figures are color-coded according to scale bar with the [001] axes achieving 16.79 times uniform. Scale bar is 10 µm.**

[Figure]

**Figure: 6: Average growth of the outer compound composite prismatic (A) and inner crossed acicular layers (B) (Table 2). Distances were measured thrice at 5 different locations (Fig. S13, S14) along the axis of maximum growth using ImageJ. Growth rates agree well between labelling and ambient conditions and are within 1ɓ (C).**

[Figure]

**Figure 7: Young moduli (upper hemisphere and equal area projection), for the compound composite prismatic (A) and crossed acicular structure (B) as well as complete shell (C). Calculations were made with the Hill averaging scheme and used the aragonite single crystal elastic properties of Pavese et al. (1992) and the EBSD data collected for this study as inputs (Mainprice et al., 2011). For every section, the minimum force needed to induce a fracture coexists with the local direction of growth. The shell presents a plane of greater resistance (~20-25% increase) normal to the local direction of growth and is oriented parallel to the growth lines.**

**Tables:**

**Table 1:** Geochemical composition of *K. rhytiphora* obtained from wavelength-dispersive X-ray spectrometry (WDS) electron probe micro analyser (EPMA) provided as wt.% ($g \cdot g^{-1}$) averages (Avg.) and standard deviations (Stdev.) for shell compositions grown under different conditions in the wild ("pre-aqua"), in aquaculture during labelling ("LE 1" and "LE 2"), and non-labelling ("pre-NE 1" and "NE 1") experiments.

| | | | $Na_2O$ | MgO | $SO_3$ | Cl | CaO | SrO |
|---|---|---|---|---|---|---|---|---|
| **Compound composite prismatic** | **Pre-Aqua** | **Avg.** | **0.72** | **0.03** | **0.10** | **0.04** | **54.38** | **0.12** |
| | (n=5) | Stdev | 0.06 | 0.02 | 0.04 | 0.01 | 0.20 | 0.03 |
| | **Pre-LE 1** | **Avg.** | **0.57** | **0.04** | **0.14** | **0.04** | **54.73** | **0.13** |
| | (n=3) | Stdev | 0.04 | 0.01 | 0.03 | 0.01 | 0.07 | 0.01 |
| | **LE 1** | **Avg.** | **0.56** | **0.04** | **0.12** | **0.03** | **52.86** | **2.36** |
| | (n=3) | Stdev | 0.04 | 0.01 | 0.02 | 0.01 | 0.09 | 0.07 |
| | **NE 1** | **Avg.** | **0.41** | **0.03** | **0.17** | **0.03** | **54.72** | **0.31** |
| | (n=3) | Stdev | 0.01 | 0.03 | 0.05 | 0.02 | 0.43 | 0.01 |
| | **LE 2** | **Avg.** | **0.65** | **0.04** | **0.12** | **0.02** | **53.80** | **2.29** |
| | (n=3) | Stdev | 0.04 | 0.01 | 0.03 | 0.01 | 0.15 | 0.02 |
| **Crossed Acicular\*** | **Pre-Aqua** | **Avg.** | **0.75** | **bdl** | **0.05** | **0.02** | **54.01** | **0.11** |
| | (n=5) | Stdev | 0.09 | - | 0.06 | 0.01 | 0.06 | 0.04 |
| | **Pre-LE 1** | **Avg.** | **0.77** | **bdl** | **0.05** | **0.02** | **53.76** | **0.10** |
| | (n=3) | Stdev | 0.09 | - | 0.03 | 0.02 | 0.07 | 0.04 |
| | **LE 1** | **Avg.** | **0.75** | **bdl** | **0.13** | **0.03** | **53.18** | **1.43** |
| | (n=3) | Stdev | 0.09 | - | 0.07 | 0.01 | 0.06 | 0.04 |
| | **NE 1** | **Avg.** | **0.72** | **0.03** | **0.20** | **0.02** | **54.52** | **0.15** |
| | (n=3) | Stdev | 0.09 | 0.05 | 0.07 | 0.02 | 0.06 | 0.04 |
| | **Limits of Detection:** | | **0.05** | **0.02** | **0.04** | **0.01** | **0.04** | **0.02** |

MnO (<0.025), BaO (<0.018), $P_2O_5$ (<0.028), $K_2O$ (<0.017), and FeO (<0.020), were analyzed and always below detection limits (provided in brackets as wt.%). *LE2 and NE 2 in the crossed acicular ultrastructure were too close to the edge to be measured with confidence and are excluded.

**Table 2**: Average growth rates from pulsed Sr-labelling experiments. Full lists of all measurements in Tables S3, S4. Rates in italics in ambient conditions NE2 were deposited within 6 days, all other rates within 12 days. Daily growth rates over the experimental period are 0.85 and 0.73 μm for the outer and inner layer, respectively, resulting in a ~17 % higher growth rate for the outer layer.

| Sample ID: | Structure: | LE 1 [μm/6d] | NE 1 [μm/12d] | LE 2 [μm/6d] | NE 2 [μm/6d] [μm/12d] | Total growth experimental period [μm/30d] or [μm/36d] | Daily growth experimental period [μm/d] |
|---|---|---|---|---|---|---|---|
| K2-01* | Outer | 5.1 ± 0.6 | 5.7 ± 0.6 | n.a. | n.a. | *10.8 ± 1.1* | 0.66 ± 0.08 |
|  | Inner | 3.5 ± 0.2 | 3.7 ± 0.3 | 4.0 ± 0.2 | *3.3 ± 0.2* | *14.5 ± 0.9* | 0.53 ± 0.03 |
| K2-02 | Outer | 2.2 ± 0.4 | 3.7 ± 0.5 | 2.6 ± 0.5 | *2.8 ± 0.3* | *11.3 ± 1.6* | 0.39 ± 0.08 |
|  | Inner | 3.6 ± 0.3 | 4.3 ± 0.3 | 3.9 ± 0.2 | *2.6 ± 0.3* | *14.4 ± 1.1* | 0.45 ± 0.05 |
| K2-04 | Outer | 13.3 ± 0.9 | 20.0 ± 0.6 | 4.8 ± 0.7 | *5.8 ± 0.8* | *43.9 ± 2.9* | 1.41 ± 0.15 |
|  | Inner | 7.0 ± 1.9 | 10.2 ± 1.0 | 4.3 ± 0.4 | *2.6 ± 0.7* | *24.0 ± 3.9* | 0.79 ± 0.17 |
| K2-06 | Outer | 3.8 ± 1.2 | 10.5 ± 1.7 | 3.5 ± 0.7 | 12.4 ± 1.6 | 30.1 ± 1.3 | 0.78 ± 0.12 |
|  | Inner | 6.0 ± 0.6 | 13.6 ± 0.6 | 5.5 ± 0.2 | 10.1 ± 0.4 | 35.1 ± 1.8 | 0.97 ± 0.05 |
| K2-08 | Outer | 6.4 ± 1.8 | 22.3 ± 1.6 | 3.5 ± 0.9 | 16.9 ± 0.5 | 49.2 ± 1.2 | 1.23 ± 0.13 |
|  | Inner | 5.2 ± 0.5 | 10.7 ± 0.3 | 3.7 ± 0.3 | 7.3 ± 0.3 | 26.9 ± 1.4 | 0.75 ± 0.08 |
| K2-11 | Outer | 2.8 ± 0.7 | 10.8 ± 1.3 | 3.6 ± 0.9 | 7.7 ± 1.3 | 24.9 ± 4.2 | 0.65 ± 0.11 |
|  | Inner | 6.7 ± 0.3 | 11.6 ± 0.2 | 4.4 ± 0.2 | 7.6 ± 0.3 | 30.3 ± 0.9 | 0.86 ± 0.02 |
|  | **Av. Outer** | **5.6 ± 0.9** | **12.2 ± 1.1** | **3.6 ± 0.7** | **9.1 ± 0.9** | **28.4 ± 2.1** | **0.85 ± 0.11** |
|  | **Av. Inner** | **5.3 ± 0.6** | **9.0 ± 0.5** | **4.3 ± 0.3** | **5.6 ± 0.4** | **24.2 ± 1.7** | **0.73 ± 0.07** |

*This individual did not show prismatic growth after NE1, while the crossed acicular structure kept growing.

---

## Referee Comment (RC2) · Anonymous Referee #2 · 25 Jan 2019

Otter and colleagues examined the aragonitic shell of Veneridae Katelysia rhytiphora, which has compound composite prismatic ultrastructure in the outer layer and cross acicular ultrastructure in the inner layer, for studying shell architecture, growth dynamics and biomineralization processes. For this purpose, the authors employed comprehensive approach using various techniques including FE-SEM, EPMA, NanoSIMS, EBSD, Micro-Raman spectroscopy, and Thermal Gravimetric Analysis, on the specimens aqua-cultured in tanks treated by pulsed labeling experiments using Sr-enriched seawater. They also simulated Young's stiffness based on the EBSD crystallographic data. This MS provides excellent comprehensive data set and important insights related with biomineralization of Veneridae clam in which the shells are composed of

representative two ultrastructures in bivalve mollusk. I felt that the MS is very descriptive, less deep insights, but I can imagine that it is an adequate way to present such a big dataset. Such comprehensive demonstration of the details of the sehll wll be very valuable for wide range of communities such as biomineralization, paleoceanography, and paleontology, so topics treated in this MS will largely appeal to a broad readership of Biogeosciences. However, I felt that there are some parts that should be improved before acceptance for publication, thus my decision is "Moderate Revision".

Description and interpretation of the data which related to crystallography and biomineralization seems to be OK, however, discussion about elemental transportation was based on very weak evidence thus problematic. Especially, the evidences the authors based on is (1) fluctuation of gray contrast observed at the growth portion during the Sr-enriched labelling experiments obtained by BSE image, even though BSE contrast is unreliable method for quantifying Sr concentration, and (2) similar enrichment factor (Shell/Seawater ratios) in labelled and non labelled conditions in both ultrastructural layers aquired by EPMA analysis, while the way for presentation of this enrichment factor is not adequate for discussing the element transport. Because most of the discussion regarding biomineralization is good quality, and because the length of the MS is already enough, so I recommend to simply delete the contents related to element transportation.

I would like to also suggest to add a new schematic drawing for summarizing the biomineralization and shell formation mechanisms obtained by this study. SEM and EBSD pictures are of course very nice, but they are sometimes too complicated for readers. A simplified drawing will be very helpful for readers to grasp the main conclusion of this MS.

Major comments: Title: The authors not only examined the pulsed Sr-labelled portion of the shell, but also examined the shell comprehensively, so I recommend changing the title.

[Figure]

P1, L24, L26-27, As mentioned above and below, the discussion of the element transportation is based on too weak evidence, so I recommend to deleting this part.

P3, L31, More detailed information of labeled seawater circulation is necessary. Did the authors use a single batch of seawater, or prepare labeled seawater every time for changing the water? How robust was the stability of the Sr concentration? The seawater renewing was performed constantly or done at once? Because the authors did not provide seawater composition, the Sr fluctuation, if exist, is suspicious. Changes in Sr/Ca ratio in seawater can easily produce Sr/Ca fluctuation in the shell. This is very important and critical for the discussion for the elemental transport mechanism.

P12, L1-17, I would suggest adding simulation data of Young's stiffness for two test cases, (1) Single aragonitic crystal, and (2) The same crystal arrangement, but have a random orientation of the crystals. Is it possible? The comparison between (1) and (2) will provide the contribution of complex 3D construction of multi-order unit of crystal arrangement, and that of between (2) and the results presented in the MS will provide a contribution of control of crystal orientation by bivalve, is this right? I am not familiar with the stiffness simulation, so I am not completely sure that this suggestion is pointing or not.

P13, L13, the "bright grey areas" must not be caused "by variation in Sr concentration". It is OK to say that the contrast between labeled and non-labeled part is caused by the Sr concentration changes, because this is validated by Sr/Ca analysis by NanoSIMS and EPMA. However, the variation within the labeled portion was not be assured. Can you see this fluctuation also in Sr/Ca map? The contrast of BSE image is not only induced by Sr concentration but also by density (mass number) and topography. As the authors discussed, organic concentration can even change the contrast of BSE. If the authors want to discuss Sr concentration variation, they should be based on Sr analysis, not on BSE image. According to this, the evidence for the discussion at P13, L19-23 relies on very weak evidence. Additionally, the authors did not provide Sr and Ca composition of seawater, so it is difficult to exclude the possibility that this variation

is attributed to the changes in seawater composition.

P14, L8-29, "4.4 Revisiting the Concept of Ion Transport Pathways". I recommend omitting this section because this section seems to be based on very weak evidence as mentioned above comments. In addition to the unreliability of BSE as Sr indicator, similar "enrichment factors for Ca and Sr (Ca-shell/Ca-seawater and Sr-shell/Sr-seawater" is not an appropriate parameter for discussing the elemental fractionation. This should be discussed by distribution coefficient (Sr/Ca-shell)/(Sr/Ca-seawater). Judging from the data in Table3, the data does not seem to satisfy enough robustness for discussing this topic. The authors also ignore fractionation between EPF (if exist) and carbonate. This can also produce low Sr/Ca ratio in the shell, without changing the EPF composition. No evidence was also presented for justifying the ACC formation obtained in this study. So, overall this section is not supported by the original data, thus should be omitted.

P16, L1-6, Conclusion. The second conclusion is OK, but the first and third conclusions were not supported by the data presented in this MS, because of the reasons as mentioned above.

Minor comments: P2, L5-10, Organic macromolecules itself can also control trace element incorporation. See, Stephenson A. E., DeYoreo J. J., Wu L., Wu K. J., Hoyer J. and Dove P. M. (2008) Peptides enhance magnesium signature in calcite: insights into origins of vital effects. Science 322, 724– 727 Wang D. B., Wallace A. F., De Yoreo J. J. and Dove P. M. (2009) Carboxylated molecules regulate magnesium content of amorphous calcium carbonates during calcification. Proc. Natl. Acad. Sci. U.S.A. 106, 21511–21516.

P4, L14, Magnification is not necessary, because it will be ultimately depends on print or screen size.

P4, L26, What is "Phenom XL"? P5, L27, "DREMEL tool" is not adequate. Maybe you should provide information of producer company, or use general name?

P13, L14, insert space between 6 and increment.

P15, L29, Why don't you add "EBSD"?

---

## Author Comment (AC1) · 13 Mar 2019

Answer to anonymous referee #1

We thank the referee for their constructive comments which will provide a helpful basis for the revision of our ms in due course. Below, we address the main points raised by referee #1.

As to the concepts we aim to present in our ms, many of the comments have shown us that, rather than being in disagreement with the views of referee #1, we did not arrive at articulating some of them clearly enough. Our answers below strive to clarify our

[Figure]

concepts better and resolve some of the perceived disagreements.

Referee comment: Otter and colleagues exposed specimens of a veneroid bivalve from Australia to episodically strongly elevated Sr levels (18 times above normal marine levels) in order to make the shell growth visible. They studied the effect of high Sr levels in the water on shell ultrastructure, crystallographic orientation, shell chemistry and growth rate. Except for the shell chemistry, all above mentioned shell properties remained unchanged. Sr/Ca values in the shell increased proportionately to that in the water, i.e. ca. 18 times, which still is way below expected thermodynamic equilibrium, a result supporting previous studies. Findings were interpreted to indicate an "intracellular, diffusion driven, selective transport" of ions across the mantle epithelium and subsequent shell formation processes via amorphous calcium carbonate. The experiment and analyses were superbly executed and I really enjoyed reading the results. A broad variety of different machines (EBSD, nanoSIMS, $\mu$Raman spectroscopy, EPMA, TGA, optical microscopy and FEG-SEM) were employed to study physical and chemical properties of the shells. Yet, the study contains a number of flaws that need to be addressed in a significantly revised version of the ms.

Referee comment: (1) Authors need to specify the overarching goals of their study more clearly and formulate specific hypotheses. For example, I do not think that the main goal was just "to visualize growth" with Sr labeling as stated in the first (= most important) sentence of the Abstract. The title lists at least two other topics. In contrast to the great data presented in this manuscript, the Abstract and Introduction are very weak, poorly structured and organized, and the overarching (and far-reaching) purpose of the study remains elusive. The text is full of juxtapositions, i.e., sentences and paragraphs need better transition. In the Abstract, actual numbers of key data must be given, i.e., the 18 times enrichment in the shell (at least in the outer portion thereof; see below) following exposure to 144 $\mu$g/g Sr instead of 8$\mu$g/g (translate these data into molar Sr/Ca ratios, please). In the Introduction, authors should first place their study into broader context and identify the motivation for this investigation (which

is not that existing in-situ staining methods affect the physiology of bivalves! See below). They need to describe open research questions and how they were addressed here. At the end of the Introduction and later in the Conclusions section, authors need to describe the implications of their finding, e.g., that bivalves likely serve as faithful recorders of the ocean chemistry etc. (which essentially emerges from the observation that Sr/Cashell changes proportionately to Sr/Cashell if the Srwater level is increased, or, as the authors expressed it – an interesting point of view by the way – irrespective of the Sr level of the water, Cashell/Cawater and Srshell/Srwater remained the same).

Answer: As outlined in our introductory paragraph above, we agree with the referee that both abstract and introduction could provide a better focus on the topics and research questions the ms touches on. Some of the misunderstanding below could have been avoided and will be clarified when these sections are rewritten upon revision of the ms. Contrary to what the referee may think, it is indeed the overarching goal of this ms to characterize shell architecture and growth at the submicron scale via visualization using Sr-pulse labelling. It is, on the other hand, only natural that this approach enables study and discussion of related aspects, which are therefore also mentioned in the abstract.

Referee comment: (2) Authors erroneously speak of outer and inner shell layer, but, in fact, they have only studied the outer shell layer, which in almost all bivalves is divided into two ultrastructurally different portions, i.e., the outer and inner portion of the outer shell layer (in the following, oOSL and iOSL). The inner shell layer (ISL) is located way back (below what is depicted in Fig. 4C) and (in a cross-sectioned shell) starts where the myostracum intersects with the inner shell surface (= aka pallial line) and ends somewhere at the hinge portion. In Figure 1B, the inner shell layer is formed approx. inside the brown areas, whereas the brown section and portions outside thereof largely belong to the iOSL; the oOSL is likely not seen in this image. The pallial line delimits the ISL from the iOSL. I recommend to look at Fig. 2A in Schöne (2013).

Answer: We are grateful to the referee for pointing out the intricacies of shell nomenclature. We did not mean to use zoological terminology in our ms, but merely applied appropriate terminology to distinguish the structurally inner parts of the shell from the outer parts in the studied area of the shell at the shell tip. While we realize that this may have been misleading, it is not the aim of our study to describe the ultrastructure in its entirety across the shell. Our study targets the area at and along the ventral margin to the outside of the pallial line. We will clarify this upon revision and adopt the appropriate nomenclature in agreement with the morphological elements of a bivalve shell.

Referee comment: (3) Surprisingly, a number of relevant recent papers dealing with very similar issues remain uncited. For example: (3a) In-situ labeling: Mouchi et al (2013) labeled oysters with manganese to study growth rates, and Mouchi et al. (2016) used immunogold to obtain insights into biomineralization processes of Crassostrea gigas. Riascos et al (2007) tested three different stains in abalone and the surf clam, i.e., calcein, alizarin and strontium chloride. (3b) Zhao et al (2017a) recently demonstrated that Sr/Ca in the outer shell layer of Corbicula fluminea increases proportionately to Sr/Ca in the ambient water and is not affected by growth rate effects. A very similar finding as reported here.

Answer: 'Labelling' methods have been around for decades and provide us with a powerful tool for many different purposes. It is therefore important to refer to the specific purpose rather than to generalize. We would argue that the papers referred to by the referee above do not at all deal with "very similar issues" and we will clarify this in the revised ms.

The main aim in our study is to use Sr pulse-labelling as a marker to study the structure of the shell at the nano-micro scale. This variety of labelling, termed 'pulse-labelling', is an accepted method often used for corals, employing either elemental or enriched isotope spikes (e.g. Brahmi et al., 2012, Domart-Coulon et al., 2014). Pulse-labelling highlights growth features at the micro-nano-scale, which 'general' labelling is not able to and, thus, the further has an entirely different focus of study than the latter.

Instead, the labelling studies carried out by Mouchi et al (2013), Riascos et al (2007) as well as of Zhao et al (2017) aimed at growth rate determination at a much lower spatial scale and, thus, were not carried out for the same purpose as our study. Similarly, immunogold labelling (Mouchi et al., 2017) is a routine method in protein chemistry used to label functional groups in specific organic molecules present in the shell. Unlike in our study, it is carried out 'ex situ' and not on living bivalves.

Our speculations on the effect of growth rates on Sr/Ca ratios are a secondary result that warranted discussion, but this topic is in no way the focus of this study. It is interesting to see, that our study apparently reproduced the observations of Zhao et al. (2017) on the lack of a growth rate effect on Sr/Ca and we will make sure to reference their work in the revised version. However, these authors used a very different and taxonomically unrelated bivalve species which, unlike the one we studied, lives in freshwater environments, and has a very different shell architecture. Therefore, this outcome, if correct, is not intuitive.

References for this answer:

Brahmi, C., Domart-Coulon, I., Rougée, L., Pyle, D. G., Stolarski, J., Mahoney, J. J. et al. (2012). Pulsed 86 Sr-labeling and NanoSIMS imaging to study coral biomineralization at ultra-structural length scales. Coral Reefs, 31(3), 741-752.

Domart-Coulon, I., Stolarski, J., Brahmi, C., Gutner-Hoch, E., Janiszewska, K., Shemesh, A., & Meibom, A. (2014). Simultaneous extension of both basic microstructural components in scleractinian coral skeleton during night and daytime, visualized by in situ 86Sr pulse labeling. Journal of Structural Biology, 185(1), 79-88.

Referee comment: (3c) An alternative mechanism of how the bivalve controls the trace and minor element levels in the shell – brought forward by Shirai et al. (2014) and based on Stephenson et al. (2008) – was also ignored: Organic macromolecules near the shell formation front exert control on which and how many ions are incorporated into the carbonate phase of the shells. If the overall production of biomass and thus

growth rate decreases (e.g., during times of low food availability), less of such organic substances are produced and the level of trace impurities in the shell carbonate automatically increases. This in turn, affect the morphology of biominerals and likely explains the more primitive ultrastructure at growth annual and even daily growth lines (biochecks) (FuÌĹlenbach et al. 2017), i.e., irregular simple/spherulitic prismatic ultrastructure (Schöne 2013). Data in Table 1 also indicate that different microstructures in your study contain different Sr levels, likely for the very reason described above. However, you did not discuss this or the fact that the relative change in the iOSL is only ca. 14 times, not 18.

Answer to 3C discussed by theme: (2) Growth lines, shell composition and architecture: In contrast to the referee's statement, reduced growth rates in bivalve shells do not scale to all moieties (mineral and organic) in the shell. Many bivalve species with nacroprismatic shell structure, for example, form an annual growth line that is organic-rich (and poorly mineralized; e.g. Soldati et al., 2008), suggesting that these species independently downregulate the mineralization of the shell from the production of the organic moiety in times of slow growth. These organic-rich shell areas do not contain vastly differing trace element budgets compared to the more mineralized parts of the shell, demonstrating that there is nothing 'automatical' about this process that could be generalized across species. Shells of bivalve species that form a mineralized growth line (e.g. Arctica islandica) contain much less overall organic moiety compared to nacroprismatic shells (Non-nacreous: ca. 1-1.5 wt% vs nacreous: 3-5 wt%, Agbaje et al., 2017a,b, 2019). It would be interesting to see any direct evidence for a downregulated production of organic components in those bivalve species that form mineralized growth lines, rather than correlative speculation as presented in Füllenbach et al. 2017.

(3) Potential control of the shell architecture by organic macromolecules: To date there is no direct evidence for how the complexities of the bivalve shell ultrastructure are connected (if at all) to the composition and amount of organic molecules present in the shell. Instead, it is well known that the composition and amount of the organic moiety in

shells varies significantly between species (Agbaje et al., 2017a,b, 2018, 2019, Currey et al 1976, Hare, 1965, Kamat et al 2000) and does so independently of the shell ultrastructure.

Compared to this direct evidence present in the literature, Füllenbach et al. (2017) base their model on how the ultrastructure of bivalve shells relates to the organic moiety on proxy analyses, namely S/Ca ratios in the shell analysed by EPMA. Direct characterization and analysis of the organic molecules in the shells is not presented in their study. Hence, this hypothesis brought forward by the referee is therefore highly speculative and suggestive at best.

(3) Potential control of trace element incorporation into the shell by organic molecules at the shell growth front This topic is also part of the next referee comment and will be addressed below. Following the referee's advice, we will be presenting and discussing these models developed by the Schöne group in more detail than currently done in the revised version of the ms.

References for this answer:

Agbaje, O. B., Thomas, D. E., McInerney, B. V., Molloy, M. P., & Jacob, D. E. (2017a). Organic macromolecules in shells of Arctica islandica: comparison with nacroprismatic bivalve shells. Marine Biology, 164(11), 208.

Agbaje, O. B. A., Wirth, R., Morales, L. F. G., Shirai, K., Kosnik, M., Watanabe, T., & Jacob, D. E. (2017b). Architecture of crossed-lamellar bivalve shells: the southern giant clam (Tridacna derasa, Röding, 1798). Royal Society Open Science, 4(9), 170622.

Agbaje, O.B.A., Ben Shir, I., Zax, D.B., Schmidt, A., Jacob, D.E. (2018) Biomacromolecules within bivalve shells: is chitin abundant? Acta Biomaterialia 80, 176-187; 10.1016/j.actbio.2018.09.009

Agbaje, O. B., Thomas, D. E., Dominguez, J. G., McInerney, B. V., Kosnik, M. A., & Jacob, D. E. (2019). Biomacromolecules in bivalve shells with crossed lamellar architecture. Journal of Materials Science, 54(6), 4952-4969.

Currey, J. D., & Kohn, A. J. (1976). Fracture in the crossed-lamellar structure of Conus shells. Journal of Materials Science, 11(9), 1615-1623.

Hare, P. E. (1965). Amino acid composition of some calcified proteins. Carnegie Inst. Washington Yearbk., 64, 223-232.

Kamat, S., Su, X., Ballarini, R., & Heuer, A. H. (2000). Structural basis for the fracture toughness of the shell of the conch Strombus gigas. Nature, 405(6790), 1036.

Referee comment: (4) The alternative mechanism of element incorporation mentioned in 2c does not require any control on uptake of elements. Although the chemical composition of the extrapallial fluids or gels (outer EPF forming the OSL, inner EPF the ISL) of marine bivalves have rarely been measured, the few available studies (e.g., Wada & Fujinuki 1976) unequivocally show that they have nearly the same ionic strength and chemical composition as the ambient seawater (Crenshaw 1972, Lorens & Bender 1980). This is no surprise, because bivalves are osmoconformers, like all other marine organisms. Imagine which energetic efforts were otherwise required if the bivalves had to constantly pump these ions out of the body fluids. Some elements such as strontium, magnesium and sodium reach the body fluids as ions from the ambient water through the gills and the gut (Wilbur & Saleuddin 1983) and across the mantle epithelium (passive diffusion). I have prepared a table for you summarizing data from Wada & Fujinuki (1976) (Table 1).

Despite this, shells are strongly depleted in many trace and minor elements. For example, if measured with a spatial resolution of ca. 50$\mu$m Sr/Ca in aragonitic OSL of Arctica islandica ranges between ca. 1-3 mmol/mol and Mg/Ca remains below 0.8 mmol/mol (e.g., Schöne et al. 2011). Even when measured by much higher spatial resolution (nanoSIMS) which might be advantageous given the strong chemical heterogeneity of the shell at the $\mu$m-scale, Sr/Ca in aragonite of Cerastoderma edule does reach values expected for equilibrium fractionation (FuÌLlenbach et al., 2017). In calcitic shells of

various species, Mg/Ca ranges between ca. 4-28 mmol/mol (see summary in Vihtakari et al. 2016). These findings lend support to the hypothesis that unwanted elements are actively excluded from the shell by specialized organic macromolecules directly at the site of shell formation (Schöne 2013; Shirai et al. 2014). How this mechanism fits to the ACC-mediated shell formation processes needs to be discussed. Since the chemistry of body fluids of bivalves resembles that of seawater, there is no need for any active transmembrane element transport. Zhao et al. (2017b) recently demonstrated very clearly that Sr, Mg and Ba levels in shells of Corbicula fluminea were not transported by active transport mechanisms and did not use the same pathways as Ca. These authors have poisened Ca2+ATPase and blocked Ca2+ channels. According to the finding by Zhao and colleagues, a passive diffusion pathway across the mantle epithelium is much more likely and would perfectly fit to the incorporation control by organic macromolecules at the shell formation front. I strongly feel that these alternative explanations must be presented and discussed.

Answer: This section in the submitted version of the ms is very speculative, and this was also pointed out by referee #2. We will follow the advice of referee #2 to reduce this section to remain closer to our robust and detailed results. Nevertheless, we welcome this opportunity to reply to the referee's comments above and to correct a number of flaws, inaccuracies and misconceptions articulated by the referee:

(1) Contrary to the referee's statement, not all marine organisms are osmoconformers.

(2) It is generally not helpful in this discussion to use poorly defined terms. This is even more relevant in the fundamentally interdisciplinary field of biomineralization where communication across discipline boundaries relies much on the correct usage of terminology. In this line of thought, terms such as "unwanted elements" which presumably refers to concepts of 'chemical fractionation' and 'incompatibility' rather than to an organism expressing its free will, and the term 'ACC-mediated' for a mechanism that produces metastable ACC as a transient precursor, but by no means as an active player that could actively 'mediate' any given process, are not furthering mutual

understanding nor scientific progress.

(3) We note that the biomineralization concepts articulated by the referee as well as in Füllenbach et al (2017) rely mainly on literature from the 1950 to 1980s. While many of the pioneering works in the field we are building on today have indeed been produced in this period of time, the field of biomineralization is very fast moving with rapid progress today being made mainly across chemistry, material sciences and physics. This large body of relevant literature is not captured in the referee's comments. One of the concepts, for instance, that experienced major revision is that of the extrapallial fluid (EPF), whose existence as a fluid with a defined composition is questioned today, to say the least. A valuable summary into the questionable nature and existence of the EPF is given in Marin et al. (2012), who state: "(. . .) its sampling is tricky. On different occasions, having done ourselves these experiments with a small syringe and a tiny needle on different model organisms, we were never fully convinced that the fluid that we were sampling was the right one! Furthermore, (. . .) it is likely that the composition of this fluid is not homogeneous, but varies from the central shell zone to the shell edge. Furthermore, it seems that the composition of this fluid also varies according to seasons." Following this reasoning we would challenge the referee's line of thought and suggest that the reason for why the table shows the composition of the EPF to be so similar to seawater is that its major component is indeed seawater, because the extrapallial space most likely is not fully sealed towards the outside.

(4) Lastly, after carefully studying Zhao et al. (2017) we find that the reasoning presented there is mostly correlative and highly speculative, while direct evidence is rarely provided to underpin their interpretation. Furthermore, the study focusses on a freshwater bivalve species with different shell architecture from the species we studied and uses different analytical methods at much lower spatial resolution. As in the life sciences generalization at this level and across species is difficult, we would be interested to learn the reasons for how the results of this study would be relevant for our work and why the apparent agreement between a subset of our results with those of Zhao et al.

(2017) would be more than a coincidence. Naturally, we are happy to reference Zhao et al. (2017) upon revision and would be more than happy to discuss the study in depth if it was relevant.

References for this answer:

Marin, F., Le Roy, N., & Marie, B. (2012). The formation and mineralization of mollusk shell. Front Biosci, 4(1099), 125.

Referee comment: (5) Another argument against ATP-mediated uptake mechanism is unchanged growth rate of the bivalve. If the hypothesis by Otter and colleagues holds true according to which an "intracellular, diffusion driven, selective transport" of ions is responsible for the observed low Sr shell concentrations, then it is surprising that shell growth rate remained unchanged. A selective transport consumes energy = ATP), and the energy demand for such a transport process increases if the Sr level in the water rises. If more energy is devoted to the control of Sr incorporation into the shell, less energy is available for shell formation resulting in lower growth rate.

Answer: Metabolic processes regulating shell growth are complex and not yet fully understood. It is an interesting and intuitive suggestion by the referee that ATP driven transport results in lower growth rates. However, without direct evidence, there is no way to test this hypothesis. This highlights just how speculative this section of the manuscript is and supports us in the decision to follow the advice given by referee #2 to cut this section significantly upon revision.

Referee comment: (6) There is a confusing usage of the term "uptake" (e.g., P2L8). 'Uptake' refers to way elements take from the environment to body fluids. This can either occur through mantle epithelia (in ionic form, potentially by one of the pathways listed in your paper) or during digestion of food. Is this really what you mean here on page 2 or rather the 'incorporation' of elements into the shell at the site of shell formation? From the context, I assume you meant the latter: "Recent studies showed that the uptake of some trace elements, such as strontium, are strongly influenced by

crystal growth rates, shell curvature and ontogeny in addition to physiological effects".

Answer: We agree with the referee that, to differentiate between 'uptake' from the water and 'incorporation' into the shell, it is more accurate to use 'incorporation' when referring to shell formation and will replace it as suggested in the revised version of the manuscript.

Referee comment: (7) A number of observation were only presented, but not discussed and combined with other aspects of the study, e.g., different amounts of organics in different ultrastructures.

Answer: We believe our discussion of the organic contents in different shell architectures is sound and fully based on the evidence provided in the ms.

Referee comment: (8) Interpretation of the timing of shell growth, meaning of microgrowth increments (= daily), major biochecks (= annual) and greyscale changes (= fortnights) is purely speculative and not supported by the data presented. This would require mark-and recovery experiments. Though not unlikely that the regular change in greyscale results from fortnightly changes, you need to cite at least relevant papers dealing in detail with such tide-controlled growth patterns (Evans 1972, Ohno 1989, Schöne 2008, Hallmann et al. 2009). B the way, you did not say where the bivalves lived: in the intertidal zone? You also noticed that you observed 6 lines in portions formed in tanks during 6 (solar) days suggesting that at least these growth patterns are circadian. However, you have no evidence that the same applies to shell portions formed in nature. Given that the specimens lived in the intertidal zone (please provide details on tidal regime: diurnal or semidiurnal, tidal range etc.), it is reasonable to assume that they have formed circalunidian growth patterns (lunar days). Perhaps, acclimatization to circadian lab conditions were sufficient to reset biological clock resulting tin switch from lunar to solar daily. However, all this needs some discussion (in the Discussion section, not results as currently presented).

Answer: In contrast to what the referee understands, the greyscale patterns in the

shells the referee refers to here (Fig. 1D, E) were not formed during aquaculture, but are growth features of the shell formed in the wild before shells were transferred to the aquarium. Our interpretation of these as time gauges for shell growth is therefore valid. Detail on the tidal regime in which these shells are found in nature will be provided in the revised version. As already stated in the discussion section of the manuscript these bivalves live in the intertidal zone. We will, however, add this information also in section "2.1 Aquaculture and labelling experiments". Discussion of these parameters will be moved to the Discussion section and relevant literature, as suggested by the referee, will be included.

Referee comment: (9) Since you are aiming to publish your paper in a journal that is often read by people of the proxy and paleoclimate communities, you need to translate oxide values into element concentrations (as well as molar element/Ca data), and all element/Ca data into molar ratios (required for easier, direct comparison with published data). Likewise, instead of reporting Ca/Sr ratios, please turn this around and give Sr/Ca data.

Answer: Molar element/Calcium ratios will be added to Table 1.

Referee comment: (10) I do not think your results allow any conclusions on whether higher Sr levels in water have or have not affected shell growth rate. If growth conditions remained invariant (aside from changing Sr levels), shells should have grown much more homogeneously. But in fact, there is a significant slowdown from LE1 over NE1, LE2 to NE2 suggesting that growth conditions deteriorated through time (Table 2).

Answer: We meant to articulate here that, while there is clearly a number of factors affecting shell growth in aquaculture, incorporation of Sr into the shell aragonite does not significantly affect growth rates in our experiment. This is evident from Figure 6C, which compares Sr-labelled and unlabelled growth increments. This figure shows clearly that all data lie within the standard deviation of the average and differences are insignificant. This will be clarified and re-written in the revised ms.

Minor comments:

Referee comment: Please check orthography in entire ms. I am not familiar with the Australian English, and whether this represents a mix of American English (e.g., analyze, labelling, meter) and British English (analyse, labelling, metre).

Answer: In the revised version of the ms we will ensure to edit all the text to British English as outlined in the journal's author guidelines.

Referee comment: Consistent use of hyphenation is required in entire ms: crossed-lamellar, crossacicular, 3 mm-thick, high-resolution, crossed-lamellar, crossed-acicular, organic-rich etc. need a hyphen

Answer: Agreed.

Referee comment: Headings: Consistently capitalize heading or use sentence case.

Answer: Agreed.

Referee comment: No colon at the end of headings! E.g., P8L21: "The inner crossed[-]acicular [shell] layer:", P9L1, etc.

Answer: Agreed.

Referee comment: P1L16, "aragonite crystals": As you noticed in the following sentence, "the smallest mineral units are nanogranules" which are enveloped by proteinaceous materials. I suggest to employ the term "mesocrystals", because the definition of an abiogenic aragonite crystal does not include nanocomposites consisting of aragonite and organic material.

Answer: Unfortunately, the referee's definition of the term 'mesocrystal' is not correct. Correctly, the term 'mesocrystal' refers to hybrid inorganic-organic nano-blocks that are aggregated to a crystal which exhibits the X-ray properties of a single crystal at the mesoscale (Cölfen and Mann, 2003). Or, as most recently defined by Bergström et al. (2015): "a nanostructured material with a defined long-range order on the atomic

scale, which can be inferred from the existence of an essentially sharp wide-angle diffraction pattern (with sharp Bragg peaks) together with clear evidence that the material consists of individual nanoparticle building units". Whether, or not some, or even all nanogranules are mesocrystals cannot be established here and is beyond the scope of the ms.

References for this answer:

BergstroÌLm, L., Sturm, E. V., Salazar-Alvarez, G., & CoÌLlfen, H. (2015). Mesocrystals in biominerals and colloidal arrays. Accounts of chemical research, 48(5), 1391-1402

Cölfen, H., & Mann, S. (2003). Higher‐order organization by mesoscale self-‐assembly and transformation of hybrid nanostructures. Angewandte Chemie International Edition, 42(21), 2350-2365.

Referee comment: P1L19, replace "shells" by 'shell portions' or 'ultrastructures'. There are no bivalves consisting entirely of nacre. I assume you intended to say that different ultrastructures contain different amounts of organics.

Answer: Agreed.

Referee comment: P1L19/20: I do not understand this sentence. Growth rates = growth patterns? Outer structure = outer shell layer. Prisms can be correlated to growth rates? Do you mean that each 3rd order prism forms in one day? Moreover, you did not mention anywhere in the text sub-daily growth patterns.

Answer: The timing of formation or the 3rd order prisms is mentioned in the text at P12L29: "while nanometre-sized third-order prisms form within hours (Fig. S6)." We will make sure to clarify and rewrite this part upon revision.

Referee comment: P1L20, "outer structure": You used the term "structure" in two different ways: as a synonym for "ultrastructure" and "shell layer" (e.g., P6L32). Be consistent. Do not use "structure", but one of the other terms above. Check and change throughout ms.

[Figure]

Answer: Agreed.

Referee comment: P1L23, "physiological processes during calcification have no lag": Rephrase, this is hard to understand. Shells do not just consist of CaCO3, but also organics which need to be fabricated, and the building blocks for these substances derive from ingested food. Digestion of food and fabrication of organic molecules that end up in the shell need time. There is hence a lag between ingestion of food and shell production. Or what do you mean with "physiological processes . . . have no lag".

Answer: We'll rewrite this and articulate this more nuanced to reflect that there is most likely 'some' lag, reflecting 'some time' as pointed out by the referee, but this lag is not significant enough to be quantified with our methods (see also comment to P13L29 below).

Referee comment: P1L23, "calcification" is the wrong term here (and used improperly in many other studies). Calcification rate includes density and is not synonymous to growth rate! Calcification rate = amount CaCO3 precipitated per time interval per area. Replace all instances with 'shell growth rate'.

Answer: We are following in our study the terminology as defined in Gillikin et al. (2005): "Considering that we discuss our results in the context of calcification processes, the distinction between growth rate and calcification rate should be made. In this study, the term growth rate is defined as the dorso-ventral linear extension of the shell per unit time (or growth increment per time)." We will clarify our usage of these terms upon revision.

Reference for this answer:

Gillikin, D. P., Lorrain, A., Navez, J., Taylor, J. W., André, L., Keppens, E., ... & Dehairs, F. (2005). Strong biological controls on Sr/Ca ratios in aragonitic marine bivalve shells. Geochemistry, Geophysics, Geosystems, 6(5).

Referee comment: P1L25, "Sr-conditions": no hyphen; 'Sr level' or 'Sr concentration'

sounds better

Answer: Agreed

Referee comment: P1L26, "Sr-enrichment": no hyphen

Answer: Agreed

Referee comment: P1L26, "Sr-enrichment factors for labelled and ambient conditions": This remains insufficiently explained and is oddly phrased. Do you mean artificially elevated Sr levels vs. normal marine Sr levels? Give actual numbers! What do you mean with "identical enrichment factors": Sr levels in shell increase proportionately to that in the water (i.e., 18 times)? As far as I can tell from Table 1, this does not apply to both shell layers (and ultrastructures).

Answer: We will follow the referee's advice and give numbers in the revised version.

Referee comment: P1L31, "aragonite or calcite": replace "or" by 'and/or'. Note there are species with different CaCO3 polymorphs in the outer and inner shell layers. Further note that some species also come with vaterite, ref

Answer: Agreed.

Referee comment: P2L3: delete "recent and fossil", superfluous

Answer: Agreed

Referee comment: P2L4+5: None of these papers used trace elements of shells as environmental proxies. Replace by suitable citations: (a) temperature: Klein et al. (1996a), Wanamaker et al (2008), Schöne et al. (2011), Zhao et al. (2017a). (b) salinity: Klein et al. (1996b). (c) pH: Zhao et al. (2017c)

Answer: Agreed.

Referee comment: P2L12: substitute "shell" with 'trace and minor elements in shells'

Answer: Agreed.

Referee comment: P2L14: substitute "but" with 'and'

Answer: will be replaced with "however"

Referee comment: P2L14: Firstly, always say 'ultrastructure', not "structure", because at other places you use "structure" as a synonym for shell layer. Secondly, this statement needs a reference.

Answer: References are already given in the text (line 15).

Referee comment: P2L15-16: Delete sentence starting with "Apart. . .". Then start next sentence with "Apart from those,"

Answer: No change

Referee comment: P2L17: replace "which are found" by 'which occur'

Answer: Agreed.

Referee comment: P2L21: The homogeneous ultrastructure forms an own category and is not a subgroup of the crossed-acicular category (compare Marin et al. 2012)

Answer: We understand that there are different schools of thought. In the current version of our ms we have followed Shimamoto et al. 1986: "(. . .) homogeneous structure in the present study is used in broader sense including crossed acicular and/or fine complex crossed lamellar structure of Carter (1980) (. . .)". Indeed, Marin et al 2012 state that "[crossed structures] represent a diversified group comprising the crossed-lamellar, complex crossed-lamellar, crossed acicular microstructures, found in most of the heterodont bivalves and in several gastropods". We will acknowledge both schools of thought upon revision of the manuscript.

Referee comment: P2L21: "venerid" must not be italicized

Answer: Agreed.

Referee comment: P2L22: "Shimamoto, 1986" is outdated (?), check most recent revision of ultrastructures by Carter JG et al. (2012)

Answer: We checked Carter et al. (2012) and found that Shimamoto 1986 is not outdated in this aspect. We will include a reference to Carter et al. (2012) in the revised version.

Referee comment: P2L33: replace "between umbo and ventral margin" by 'parallel to the main growth axis' or 'parallel to the umbo-ventral margin axis'

Answer: Agreed

Referee comment: P3L5-6: "Growth lines. . ." show/refer to figure

Answer: Agreed.

Referee comment: P3L16: Two main clauses combined by conjunction require comma; check and correct throughout ms: ', and'

Answer: Will be corrected

Referee comment: P3L17: Specimens: Much more information needed here: sediment type, tidal height, intertidal zone(?), how many specimens collected/prepared/used for which analytical technique, when collected. Table would be best. Part of this information is relevant for the temporal alignment of the shell growth patterns.

Answer: Referee has later (below) accepted our reasoning for the time gauge. No change.

Referee comment: P3L17: replace "live-collected" by 'collected alive'

Answer: Agreed.

Referee comment: P3L20: use 'x' as mathematical operator (consistently throughout ms)

Answer: Agreed

Referee comment: P3L24-26: Has the element composition of the food been measured as well? How do you know that all Sr and Ca comes from the water? Has always the same amount of food being offered? When were they fed, during simulated day or nighttime?

Answer: We did not measure the Ca and Sr content of the diet. Food was added in the morning and we observed that the water had cleared up by the end of the day (after ca. 6 hours). This indicates an extended time of filter feeding (food uptake). If significant Sr and Ca were derived from the diet we would expect concentration differences (visible in maps) in unlabelled areas between shell portions formed during day and night. This was not observed (Fig. X).

Referee comment: P3L29-30: An "event" is a very short-term incident. This sentence should be rephrased, e.g., "exposure to background conditions, i.e., normal marine Sr levels".

Answer: Sentence will be rephrased.

Referee comment: P4L7: P400-P2000

Answer: Agreed.

Referee comment: P4L12: thickness of gold-coating?

Answer: 15 nm - we will add this in the revised version of the manuscript.

Referee comment: P4L21: 20,000x

Answer: Agreed.

Referee comment: P5L9: replace "was used" by "were used"

Answer: Agreed.

Referee comment: P5L21: $\mu$m2 (superscript)

Answer: Agreed.

Referee comment: P5L27 "The inner and outer layer of a K. rhytiphora shell were separated with a DREMEL tool and mechanically cleaned." Be more specific here: Have you obtained powdered material or fractions of the two portions of the outer shell layer? How have you managed exactly to separate them?

Answer: We will clarify this in the revised version of the ms

Referee comment: P6L3: Actually wrong. You have only studied the outer shell layer, which consists of two portions with different ultrastructure, an outer and inner portion, respectively (oOSL, iOSL)!

Answer: We will clarify and correct this in the revised version of the ms

Referee comment: P6L8: Rephrase (and italicize genus and species names): 'The outer shell layer of studied K. rhytiphora specimens is ... near the ventral margin'

Answer: Agreed.

Referee comment: P6L10: "in agreement with previous studies. . ." This phrasing means that the other species studied by Carré and Soldati and colleagues lived in Australian waters. Rephrase.

Answer: We will edit this to clarify that these are literature examples from the Southern hemisphere but not from Australia.

Referee comment: P6L11: "growth periods": delete "periods"

Answer: Agreed.

Referee comment: P6L13: "troughs" Odd phrasing. Something like this is better: 'Cyclic changes in greyscale near the ventral margin correlate strongly with tidal cycles, i.e., light grey and dark grey portion fall together with full and new moon cycles, respectively.' The main problem is that you do not provide any evidence for the timing of shell growth! Where is the evidence that the dark and light portions really have formed during new and full moon? This is an interpretation at most, and as such belong to

the Discussion section (where you need to refer to previous studies of intertidal bivalves which found narrower increments and thicker growth lines formed during spring tides, and these portions then appear darker than shell portions formed during neap tides when viewed at lower magnification and under reflected light. More suitable Refs: Evans 1972, Schöne 2008, Hallmann et al. 2009)

Answer: Will be moved and rewritten upon revision

Referee comment: P6L16-25 also needs to be moved to Discussion. Only keep descriptive part here. You have no evidence that these grey bands formed on a circalunidian basis, but you can certainly interpret them as such based on previous work.

Answer: Agreed

Referee comment: Timing of shell growth: You later noticed that you observed 6 lines in portions formed in tanks during 6 days suggesting that at least these growth patterns are circadian. However, you have no evidence that the same applies to shell portions formed in nature. Given that the specimens lived in the intertidal (please provide details on tidal regime: diurnal or semidiurnal, tidal range etc.), it is reasonable to assume that they have formed circalunidian growth patterns. Here, please stick to descriptions, not interpretation.

Answer: Agreed – we will revise this part and add that the shell areas formed in the natural environment perhaps follow circalunidian growth patterns (a semi-lunar day eq. to 12h25mn).

Referee comment: P6L25: 'in two other specimens', not "on two other specimens"

Answer: Agreed.

Referee comment: Section 3.2: Title is more suitable for Discussion. – This section should be expanded as it is an essential component of the ms and forms the basis for your hypothesis on element incorporation. Describe Table 1 in much more detail. Report molar ratios as well. Compute and tell reader by how much the Sr levels increased

in the shell when exposed to 18 times higher Sr levels in water. This will then show that the Sr levels in oOSL increased by 18 to 20 times, whereas the iOSL only by ca. 14 times. This needs to be discussed later.

Answer: Thank you for the suggestions. This will be taken care of during general clarification and restructuring of the ms.

Referee comment: P7L2: ". . .were identical within uncertainty": i.e., they have remained invariant, stayed the same? I suggest you rephrase this to avoid confusion.

Answer: This is geochemically the correct terminology and means within analytical uncertainty.

Referee comment: Title Section 3.3: Section heading should inform about content of section, not which method has been used.

Answer: This section contains the Raman spectroscopy results and is, as such, correctly titled.

Referee comment: P7L31: "This species develops annual growth checks" On what evidence is this statement based? How did you analyze when the shell portions formed? Likely correct, but pure speculation... or is there previous work on this species?

Answer: Referee has below accepted our reasoning for the time gauge. No change necessary.

Referee comment: Section 3.4.3: Interesting information, but what is the purpose of having this measured and reported?

Answer: The amounts and composition of the organic moiety are not well known, particularly for non-nacreous shells. We are presenting new results here with the purpose of closing this knowledge gap. We note that the referee finds these results interesting.

Referee comment: P9L10: crystallographically

Answer: "crystallographic preferred orientation (CPO)" is the correct name of this type of data. No change.

Referee comment: P9 "Calcification Rates" includes density, not synonymous to growth rate! Calcification rate = amount CaCO3 precipitated per time interval per area; this is not what you mean.

Answer: see our answer above

Referee comment: P9L28-29: "Due to the geometry of first order prisms without- and inward bending in cross-sections,. . ." No sentence

Answer: This sentence will be edited upon revision.

Referee comment: P10L2: Provide image showing where you determined increment widths, or even better trace two growth lines to show that growth in oOSL is faster than in iOSL due to shell geometry.

Answer: These images are already in the supplement.

Referee comment: P10L4-5: quite complicated phrasing: absolute growth rates vary among specimens

Answer: Will be rephrased in the revised version.

Referee comment: P10L5: grew, on average, 5.6... same for the other "on average": separate by comma and place before number

Answer: Agreed.

Referee comment: P10L12: "Also, rates tend to decrease effectively with increasing distance to the ventral margins (Fig. 4A).": Unclear what you mean and purpose of mentioning this. You need to trace fortnights in Figure 4A to support your statement.

Answer: Agreed, will be clarified in the revised version.

Referee comment: "bivalve species": you listed genera not species rephrase: ... structure of other bivalves, e.g., Pinna..., the aragonitic...

Answer: Agreed.

Referee comment: P10L23, P11L5, P11L23: "In K. rhytiphora the first order prisms" comma after species name

Answer: Agreed

Referee comment: P11L10: Equally-sized (adverbial usage)

Answer: Agreed

Referee comment: P11L30: Since you did not capitalize "aragonite", you should also use lower case here (except for the acronym/abbreviated form). Besides that, you used lower case in the Abstract.

Answer: Agreed

Referee comment: P12L30: replace "the outside of their shells" by 'outer shell surface (Fig. 1), and'

Answer: Agreed.

Referee comment: P13L1: "growth time": Firstly, you have no evidence that these growth checks formed annually. Secondly, no bivalve grows 365 days. Note also that such ornamentation patterns do not agree with growth patterns in other species, and likely this is a coincident and only true for shell portions near the ventral margin in the studied specimens. Rephrase.

Answer: Referee comment below retracts this one – no change

Referee comment: P13L12-13: Perfect! This is your time gauge. It verifies the circa daily nature of these growth features and could further be used to support your hypothesis of fortnightly growth bundles appearing as greyscale changes.

Referee comment: P13L14-15: replace "higher" with 'faster', "short" with 'narrow',

"longer" with 'broader'

Answer: Agreed

Referee comment: P13L15: "day": An interesting question that you need to discuss is that these are probably circadian (24h) periods entrained by the 12/12 light/dark cycle experimental conditions. The adjustment interval was probably long enough that the natural, tide-entrained shell formation cyclicality (resulting in circalunidian, 24.8h, periods) vanished. Under natural conditions though, you would need to have circatidal (12.4h) and circalunidian increments, because otherwise your interpretation of the other 48 or 50 dark cycles representing fortnight periods would not hold true.

Answer: We agree and will add this to the discussion of our data in the revised version of the ms.

Referee comment: P13L22: "We suggest a diel physiologically controlled variation of calcification" Not sure exactly what you mean. Circadian clock controlling growth/calcification rate? This has been reported previously elsewhere.

Answer: Will be clarified upon revision

Referee comment: P13L29 "have no lag"? Well, this depends on the temporal scale you are looking at. Where is the evidence that there was no gradual increase in shell Sr levels during the course of minutes or so? Diffusion of Sr through the mantle epithelium takes at least some time.

Answer: See answer above

Referee comment: P13P29"physiological processes involving Sr incorporation", rephrase: 'physiological processes controlling Sr incorporation'

Answer: Agreed

Referee comment: P13L30-31: I do not think that the implications provided are supporting an ACC mediated growth of shell in bivalves.

[Figure]

Answer: Will be rephrased in the revised version

Referee comment: P13L33-34: "A fundamental observation of this study is that the calcification front runs evenly across all structural units and architectural orders of the shell independently of the current growth rate. This" But this is known and no a new finding of this study!

Answer: We disagree. This was not known for compound composite prismatic and crossed-acicular shells and is an entirely new finding. We note the lack of literature evidence for the referee's statement here.

Referee comment: P14L1: "show the labels to cut across the different architectural building blocks": could also occur if extrapallial space is gel-filled or epithelial cells are in direct contact with shell

Answer: Agreed.

Referee comment: P14L2 "where the label would rather follow a zig-zag trend between fully labelled and unlabelled units" Impossible to understand what you intend to say here. Rephrase please. Do you mean that the growth front is uncoupled from the ultrastructures? This is known as well: In freshwater bivalves the large prisms continue to grow over many years and daily growth lines cross them perpendicularly (studies by Dunca, Mutvei etc.).

Answer: This will be clarified. We disagree that this was already known. This was not known for compound composite prismatic and crossed-acicular shells.

Referee comment: P14L2-4: "This is clearly visible from the sharply defined change between labelled and unlabelled shell areas (Fig. 4B and D), as well as from the cyclic variations in short-term growth rates (discussed above). Our" Likewise hard to understand

Answer: Agreed and will be revised.

Referee comment: P14L10 "active selective transport consuming Ca2+-ATPase enzymes": transport consumes energy which is provided by ATP, and the enzyme that accomplishes the transportation is the Ca ATPase. Rephrase.

Answer: Agreed

Referee comment: P14L14-15 "We observed virtually identical enrichment factors for Ca and Sr (CaShell/CaSeawater and SrShell/SrSeawater) in labelled and ambient conditions (Table 3).": Interesting point of view! But this does not mean anything else than Sr/Ca shell increases proportionately to that of Sr/Ca seawater, and this has already been shown by Zhao et al. (2017), which you did not cite.

Answer: Will be clarified in the revised version.

Referee comment: P14L15: "Sr-ion transport is independent from. . . " if so, the energy demand of the bivalve increases in order to keep the Sr out of the shell. Do you see a decrease in growth rate during Sr enrichment as opposed to 'normal' Sr levels in water?

Answer: Will be clarified in the revised version.

Referee comment: P14L16-17: "Sr ion would be at the expense of a Ca ion": Not really clear what you mean; Since this is the essence of your paper, you need to describe this more clearly and convincingly. Why exactly can transport mechanism 1 not be true?

Answer: This section will undergo a major re-write and cut upon revision.

Referee comment: P14L17: Replace "Sr-enrichment" by 'shell Sr concentrations'

Answer: Agreed

Referee comment: P14L18: Ca/Sr: please also or only report Sr/Ca

Answer: Agreed

Referee comment: P14L19-20: Replace "Hence, the strong enrichment of Ca from

seawater to shell" by 'strong enrichment of Ca in shell'

Answer: Agreed

Referee comment: P14L25: "Ca to be transported as ACC-nanogranules to the calcification front (Loste et al., 2004; Addadi et al., 2006; Jacob et al., 2011; Zhang and Xu, 2013)." Check if all cited studied were using bivalves (not gastropods or other taxa), and which ultrastructures were analzed, report this here.

Answer: The point is here that ACC-nanogranules have been observed across ALL TAXA to be part of a common principle of biomineralization. We would be happy to refer the referee to review studies who do exactly what is suggested here, but which is beyond the scope of our study.

Referee comment: Section 4.5: Here you discuss more (and different stuff) than what the heading implies.

Answer: We disagree. All aspects and 'stuff' discussed here belong under this heading.

Referee comment: P15L12: italicize genus and species names

Answer: Agreed.

Referee comment: P15L14-15: "a systematic change in growth increments during Sr-enriched periods": Do you mean 'growth increment widths'? You need to highlight here again that food levels and other extrinsic factors that could potentially have affected growth rate remained unchanged during the experiment, and you would have expected invariant increment widths if Sr had no effect on growth rate... see comment further above on relationship between growth rate and Sr exclusion from shell

Answer: Will be clarified

Referee comment: P15L18: Replace "calcification" by 'growth rate'

Answer: We will define these terms better and use according to their definition.

Referee comment: P15L23-24: "Reduced growth rates in aquaculture conditions cannot be explained by ontogenetic trends alone but result from missing tidal cycles." Sorry, but this is pure speculation and likely wrong. Much more likely is that you did not provide proper food and the animals did not really 'like' the tank conditions.

Answer: Will be clarified and rewritten to be more convincing

Referee comment: P15L30: 'nanometer'?

Answer: This depends.

Referee comment: More comments in pdf with annotated figures and tables.

Reviewer Comments from the Supplementary Information:

Referee comment: [To Fig. 1] I recommend removal of boxes around letters A, B, C...

Answer: We thank the referee for this suggestion. No change.

Referee comment: [To Fig. 1] FYI: The inner shell layer is formed inside the brown areas, whereas the brown section and portions outside thereof largely belong to the iOSL; the oOSL is likely not seen in this image.

Answer: Already clarified above, shell layers will be renamed.

Referee comment: [To Fig. 1] The pallial line is the small indentation ca. 1.5 cm away from the ventral margin on the inner shell surface. The pallial line strikes out again at the cardinal tooth (hinge; likewise a small kink developed). The inner shell layer is formed inside the portion delimited by the pallial line 'strikeouts'.

Answer: This is correct.

Referee comment: [To Fig. 1] Outer (A) and inner shell surface (B)...

Answer: Agreed.

Referee comment: [To Fig. 1] Denote: Arrows cannot be summers!

Answer: Agreed.

Referee comment: [To Fig. 1] Low or high values? dark or light grey?

Answer: Agreed and will be clarified upon revision.

Referee comment: [To Fig. 1] Temporal alignment is purely speculative. Though I believe it is correct, you did not provide any convincing support this.

Answer: See referee comment above where they backtrack on this comment

Referee comment: [To Fig. 2] Captions should explain things. You fail to say what the pink and blue actually means: Sr enrichment in shell during immserion in Sr-enriched (give molar value) tank water and Sr shell levels during times of normal marine Sr conditions of 8.9 mmol/mol.

Answer: Agreed and will be clarified upon revision.

Referee comment: [To Fig. 2] Indicate DOG.

Answer: Agreed

Referee comment: [To Fig. 2] That are the inner and outer portions of the OUTER SHELL LAYER; B is not the inner shell layer!

Answer: Agreed

Referee comment: [To Fig. 2] This does not look polished, but etched.

Answer: The referee is incorrect.

Referee comment: [To Fig. 2] You need to provide a schematic of the shell to show where exactly the images in Figures 2-5 were taken (hinge or ventral margin, where in the ventral margin?). As outlined below you only sampled the inner and outer portions of the outer shell layer, but not actually the inner shell layer.

Answer: In our first caption we state "All cross-sections in this study are prepared as

radial sections along the maximum growth axis unless otherwise specified."

Referee comment: [To Fig. 3] Denote DOG

Answer: Agreed.

Referee comment: [To Fig. 4] That's incorrect! You are still in the outer shell layer here. These are just two different portions of the outer shell layer. The inner shell layer is way back (below in this image) and starts where the myostracum intersects with the inner shell surface (= aka pallial line) and then ends somewhere at the hinge portion. Shell material is added laminarily along the entire inner shell surface beyong the pallial line and results in thickening of ontogenetically younger shell portions (near the umbo and hinge). The inner portion of the outer shell layer actually does the as the outer portion: contributes to size increase of the entire shell, but in addtion also contributes to thickening. I recommend to look at the Figure in Schöne (2013) for morphological details of a veneroid.

Answer: Already addressed above.

Referee comment: [To Fig. 5] I am lost here. Is this image mirrored and upside down? Can you just present it in the same way as Figure 4? And please describe here again which portion on the two sides of the dotted line is the inner (left of line?) and outer portion of the outer shell layer (right of line).

Answer: Will be clarified upon revision.

Referee comment: [To Fig. 6] agree with what? each other? Rephrase. Within each shell layer, growth rates are unaffected by exposure to higher Sr content in water. How have you actually tested this mathematically? Student t-test? Normal distibution test done?

Answer: The information that the four different increments in 6C are within errors is based on their positions of the diagram. We will clarify this part of the caption upon revision.

Referee comment: Table 1: without considering Ca... 18-20x oOSL, 13-14x iOSL, Why different? Why does only Sr level in oOSL change proportinately to Sr increase in water?

Answer: Firstly it is a general observation that values are higher for oOSL compared to iOSL. It could be speculated that this is caused by the different overall length growth rates between the two shell layers combined with the curvature of the shell. We will discuss this in our revised version.

Referee comment: Table 2: I do not understand "or". Which of the data in this column are 12d and 6d? For direct comparison of data, provide daily growth rates for all data.

Answer: The referee may have missed the explanation already provided (coded by italicised and normal text). We will use a clearer coding upon revision.

---

## Author Comment (AC2) · 13 Mar 2019

Answer to anonymous referee #2

We thank the referee for their constructive comments, which will provide a helpful basis for the revision of our ms in due course.

We appreciate that the referee feels that the way of presenting our large dataset is adequate. The referee noted that the ms is more on the descriptive side, and this is indeed a main in our work as this shell architectural type has never been studied at the micro to nano scale before. We see the work presented in this ms as a baseline for

further work carried out on this type of shell architecture in the future, which necessarily needs to build on a detailed description.

In the following we are listing the referee's comments followed by our answer.

Referee comment: "Description and interpretation of the data which related to crystallography and biomineralization seems to be OK, however, discussion about elemental transportation was based on very weak evidence thus problematic. Especially, the evidences the authors based on is (1) fluctuation of gray contrast observed at the growth portion during the Sr-enriched labelling experiments obtained by BSE image, even though BSE contrast is unreliable method for quantifying Sr concentration, and (2) similar enrichment factor (Shell/Seawater ratios) in labelled and non-labelled conditions in both ultrastructural layers aquired by EPMA analysis, while the way for presentation of this enrichment factor is not adequate for discussing the element transport. Because most of the discussion regarding biomineralization is good quality, and because the length of the MS is already enough, so I recommend to simply delete the contents related to element transportation."

Answer: We agree that the discussion about elemental transportation is a more speculative part of the manuscript and after critically revisiting it, we have decided to shorten this part considerably upon revision. We also agree that BSE imaging per se is an unreliable method of quantification. This is exactly the reason why we went to great length to calibrate the grey scale of the BSE imaging by combining it with quantitative electron microprobe measurements using Wavelength Dispersive Spectrometry (WDS) with the same instrument and in the same session as carrying out the BSE imaging. Such calibration enables direct comparison of the grey scales in the BSE images with the quantitative data using WDS in the Electron Microprobe. Further down in the text, the referee agrees with us on this point.

Referee comment: "I would like to also suggest to add a new schematic drawing for summarizing the biomineralization and shell formation mechanisms obtained by this

study. SEM and EBSD pictures are of course very nice, but they are sometimes too complicated for readers. A simplified drawing will be very helpful for readers to grasp the main conclusion of this MS."

Answer: We will provide a schematic upon revision.

Referee comment: "The authors not only examined the pulsed Sr-labelled portion of the shell, but also examined the shell comprehensively, so I recommend changing the title."

Answer: We believe the current title already reflects the 'comprehensive examination of the shell' as stated by the referee. The purpose of the label is to provide markers in time for the study of the entire shell. We will however reorganise and add the term 'shell' to the title to specify the aim of the study. We suggest the new title to be: ''Architecture, Growth Dynamics, Biomineralization and Pulsed Sr-Labelling of Katelysia rhytiphora shells (Mollusca, Bivalvia)'

Referee comment: "P1, L24, L26-27, as mentioned above and below, the discussion of the element transportation is based on too weak evidence, so I recommend to deleting this part."

Answer: As outlined above we intend to shorten and focus this section in the ms and will rewrite the abstract to reflect this.

Referee comment: "P3, L31, More detailed information of labelled seawater circulation is necessary. Did the authors use a single batch of seawater, or prepare labelled seawater every time for changing the water? How robust was the stability of the Sr concentration? The seawater renewing was performed constantly or done at once? Because the authors did not provide seawater composition, the Sr fluctuation, if exist, is suspicious. Changes in Sr/Ca ratio in seawater can easily produce Sr/Ca fluctuation in the shell. This is very important and critical for the discussion for the elemental transport mechanism."

Answer: For labelling, seawater was enriched in Sr by adding 4.380 g Sr-hexachloride per 10 l of seawater and was freshly prepared each time the water had to be renewed. Renewing the water was done at the start and in the middle of each labelling period. As we used natural seawater from two large 2 seawater storage tanks of 10,000 ltr capacity each and a high precision balance, precise to the third digit, we consider the Sr concentration data robust.

Referee comment: "P12, L1-17, I would suggest adding simulation data of Young's stiffness for two test cases, (1) Single aragonitic crystal, and (2) The same crystal arrangement, but have a random orientation of the crystals. Is it possible? The comparison between (1) and (2) will provide the contribution of complex 3D construction of multi-order unit of crystal arrangement, and that of between (2) and the results presented in the MS will provide a contribution of control of crystal orientation by bivalve, is this right? I am not familiar with the stiffness simulation, so I am not completely sure that this suggestion is pointing or not."

Answer: Presenting the Young's modulus for a single crystal will require the use of a different reference frame, namely the crystal structure, which is distinct from that used here, which is using the shell orientation. We believe that this could lead to more confusion for the reader rather than contribute to a better understanding. We will include a reference to an earlier publication from our group that shows the Young's modulus for an aragonite single crystal as reference (Agbaje et al 2017). Depicting a randomly oriented fabric in a pole figure means that the aragonite crystallographic axes will be randomly oriented. Therefore, the elastic properties of the crystal would be averaged and the fabric would be isotropic. A pole figure depicting an isotropic orientation would show an even distribution across the entire pole figure and would therefore be very uninformative to the reader. We propose to add a sentence describing that a sample with random crystal orientation would lead to isotropic results.

References for this answer:

O.B.A. Agbaje, R. Wirth, L.F.G. Morales, K. Shirai, T. Watanabe, M. Kosnik, D.E. Jacob (2017). Architecture of crossed-lamellar bivalve shells: The Southern Giant Clam (Tirana dears, Roding, 1798). R. Soc. Open Sci. 4: 170622. http://dx.doi.org/10.1098/rsos.170622

Referee comment: "P13, L13, the "bright grey areas" must not be caused "by variation in Sr concentration". It is OK to say that the contrast between labelled and non-labelled part is caused by the Sr concentration changes, because this is validated by Sr/Ca analysis by NanoSIMS and EPMA. However, the variation within the labelled portion was not be assured. Can you see this fluctuation also in Sr/Ca map? The contrast of BSE image is not only induced by Sr concentration but also by density (mass number) and topography. As the authors discussed, organic concentration can even change the contrast of BSE. If the authors want to discuss Sr concentration variation, they should be based on Sr analysis, not on BSE image. According to this, the evidence for the discussion at P13, L19-23 relies on very weak evidence. Additionally, the authors did not provide Sr and Ca composition of seawater, so it is difficult to exclude the possibility that this variation is attributed to the changes in seawater composition."

Answer: The resolution of the Sr/Ca maps obtained by NanoSIMS unfortunately do not allow to observe any variation at this spatial scale. As argued above, the Sr and Ca composition of the water, particularly during the labelling periods, is constant within analytical uncertainty and can thus be excluded as a source of grey scale variability in BSE. Neither did we observe growth irregularities (e.g. organic components). Topography and edge effects would not result in such regular patterns of grey scales as observed here.

The fine grey banding, however, also shows up in the Raman maps (e.g. Figs.3, S2) and as Raman is not sensitive to electron density effects this would exclude these as a cause for the banding. Furthermore, deconvolution of the Raman signal is consisted with variation in $SrCO_3$ concentration as underlying cause for the grayscale banding observed in BSE, as increased Sr concentration in aragonite results in peak broadening

(Fig. 3) and peak shift of the main carbonate band to lower wavenumbers (Fig. S2) (Alia et al., 1997, O'Donnell et al., 2008, Ruschel et al., 2012). This is direct evidence for the correlation of lighter grey scales in BSE with higher Sr concentrations in the aragonitic shell. Upon revision of the manuscript we will clarify this connection between BSE and Raman analysis more.

References for this answer:

Alia, J. M., Mera, Y. D. de, Edwards, H. G.M., Martín, P. G., and Andres, S. L.: FT-Raman and infrared spectroscopic study of aragonite-strontianite (CaxSr1− xCO3) solid solution, Spectrochimica Acta Part A: Molecular and Biomolecular Spectroscopy, 53, 2347–2362, 1997

O'Donnell, M. D., Fredholm, Y., Rouffignac, A. de, and Hill, R. G.: Structural analysis of a series of strontium-substituted apatites, Acta Biomaterialia, 4, 1455–1464, 2008.

Ruschel, K., Nasdala, L., Kronz, A., Hanchar, J. M., Többens, D. M., Škoda, R., Finger, F., and Möller, A.: A Raman spectroscopic study on the structural disorder of monazite– (Ce), Mineralogy and Petrology, 105, 41–55, 2012.

Referee comment: "P14, L8-29, "4.4 Revisiting the Concept of Ion Transport Pathways". I recommend omitting this section because this section seems to be based on very weak evidence as mentioned above comments. In addition to the unreliability of BSE as Sr indicator, similar "enrichment factors for Ca and Sr (Ca-shell/Ca-seawater and Sr-shell/Sr-seawater" is not an appropriate parameter for discussing the elemental fractionation. This should be discussed by distribution coefficient (Sr/Ca-shell)/(Sr/Ca-seawater). Judging from the data in Table3, the data does not seem to satisfy enough robustness for discussing this topic. The authors also ignore fracnation between EPF (if exist) and carbonate. This can also produce low Sr/Ca ratio in the shell, without changing the EPF composition. No evidence was also presented for justifying the ACC formation obtained in this study. So, overall this section is not supported by the original data, thus should be omitted."

Answer: After reflecting on both reviewers' comments that this section is very speculative, we feel it is best to shorten this section considerably and to focus only on discussing the most likely ion transport pathway. In addition, we will use the distribution coefficients (Sr/Ca(shell)/Sr/Ca(seawater)) to outline our thoughts and change Table 3 accordingly. Including possible fractionation by any potential EPF is extremely speculative and is therefore not warranted. However, we will make sure that the revised version of the ms reflects any further potential fractionation by a (potential) EPF and tone down the assertive tone of this section.

Referee comment: "P16, L1-6, Conclusion. The second conclusion is OK, but the first and third conclusions were not supported by the data presented in this MS, because of the reasons as mentioned above."

Answer: We have listed four conclusions and believe the referee is referring to conclusion two and four as being too speculative. The revised ms will take this into account and this section will be re-written.

Referee comment: "Minor comments: P2, L5-10, Organic macromolecules itself can also control trace element incorporation. See, Stephenson A. E., DeYoreo J. J., Wu L., Wu K. J., Hoyer J. and Dove P. M. (2008) Peptides enhance magnesium signature in calcite: insights into origins of vital effects. Science 322, 724– 727 Wang D. B., Wallace A. F., De Yoreo J. J. and Dove P. M. (2009) Carboxylated molecules regulate magnesium content of amorphous calcium carbonates during calcification. Proc. Natl. Acad. Sci. U.S.A. 106, 21511–21516.

Answer: Thank you for pointing out the omission of these important works. This part of the discussion will be rewritten, also taking the comments referee #1 on board and we refer the readers to our answer to referee #1 for this issue.

Referee comment: "P4, L14, Magnification is not necessary, because it will be ultimately depends on print or screen size."

Answer: Thank you, mention of the magnification factor will be omitted in the revised version.

Referee comment: "P4, L26, What is "Phenom XL"? P5, L27, "DREMEL tool" is not adequate. Maybe you should provide information of producer company, or use general name?"

Answer: Phenom XL is the product name of the SEM used in this study. Similarly, a DREMEL tool is the official name of this tool. The term SEM is mentioned in the same sentence with Phenom XL. We suggest adding the term 'power tool' after 'DREMEL' to make this clearer.
* * *

---

## Author Response (AR1)

**Referee #1:**

We thank the referee for the constructive comments which will provided a helpful basis for the revision of our ms in due course. Below, we address the main points raised by referee #1.

As to the concepts we aim to present in our ms, many of the comments have shown us that, rather than being in disagreement with the views of referee #1, we did not arrive at articulating some of them clearly enough. Our answers below strive to clarify our concepts better and resolve some of the perceived disagreements.

Referee comment: Otter and colleagues exposed specimens of a veneroid bivalve from Australia to episodically strongly elevated Sr levels (18 times above normal marine levels) in order to make the shell growth visible. They studied the effect of high Sr levels in the water on shell ultrastructure, crystallographic orientation, shell chemistry and growth rate. Except for the shell chemistry, all above mentioned shell properties remained unchanged. Sr/Ca values in the shell increased proportionally to that in the water, i.e. ca. 18 times, which still is way below expected thermodynamic equilibrium, a result supporting previous studies. Findings were interpreted to indicate an "intracellular, diffusion driven, selective transport" of ions across the mantle epithelium and subsequent shell formation processes via amorphous calcium carbonate.

The experiment and analyses were superbly executed and I really enjoyed reading the results. A broad variety of different machines (EBSD, nanoSIMS, μRaman spectroscopy, EPMA, TGA, optical microscopy and FEG-SEM) were employed to study physical and chemical properties of the shells. Yet, the study contains a number of flaws that need to be addressed in a significantly revised version of the ms.

Referee comment: (1) Authors need to specify the overarching goals of their study more clearly and formulate specific hypotheses. For example, I do not think that the main goal was just "to visualize growth" with Sr labeling as stated in the first (= most important) sentence of the Abstract. The title lists at least two other topics. In contrast to the great data presented in this manuscript, the Abstract and Introduction are very weak, poorly structured and organized, and the overarching (and far-reaching) purpose of the study remains elusive. The text is full of juxtapositions, i.e., sentences and paragraphs need better transition. In the Abstract, actual numbers of key data must be given, i.e., the 18 times enrichment in the shell (at least in the outer portion thereof; see below) following exposure to 144 μg/g Sr instead of 8μg/g (translate these data into molar Sr/Ca ratios, please). In the Introduction, authors should first place their study into broader context and identify the motivation for this investigation (which is not that existing in-situ staining methods affect the physiology of bivalves! See below). They need to describe open research questions and how they were addressed here. At the end of the Introduction and later in the Conclusions section, authors need to describe the implications of their finding, e.g., that bivalves likely serve as faithful recorders of the ocean chemistry etc. (which essentially emerges from the observation that $Sr/Ca_{shell}$ changes proportionately to $Sr/Ca_{shell}$ if the $Sr_{water}$ level is increased, or, as the authors expressed it – an interesting point of view by the way – irrespective of the Sr level of the water, $Ca_{shell}/Ca_{water}$ and $Sr_{shell}/Sr_{water}$ remained the same).

Answer: As outlined in our introductory paragraph above, we agree with the referee that both abstract and introduction could have provided a better focus on the topics and research questions the ms touches on and we have edited these sections accordingly. Some of the misunderstandings below could have been avoided and have been clarified in the revised version of the ms. Contrary to what the referee may think, it is indeed the overarching goal of this ms to characterize shell architecture and growth at the submicron scale via visualization using Sr-pulse labelling. It is, on the other hand, only natural that this approach enables study and discussion of related aspects that are incorporated into this manuscript and that are therefore also mentioned in the abstract. We now provide numbers in the abstract and molar Sr/Ca ratios. We are aware that molar Sr/Ca ratios are frequently used in sclerochronology, however, this is not the case for biomineralization, geochemistry, and structural biology.
The increase of Sr in shell calculated by the reviewer are in fact incorrect, as these are based on oxide concentrations given in the manuscript and not on element concentrations. We provide now the correct values that result in lower values with 13x (oOSL) and 12x (iOSL) but are still in the same dimension as the 18x assumed by the referee. 12x for the iOSL results also from a minim value as stated in the footnote of table 1. For clarity we added ">" in front of the value within the table. These factors are now also presented in the results and discussion sections together with the new distribution factors, which demonstrate that the system is in disequilibrium compared to the equilibrum partitioning of Sr/Ca during inorganic precipitation of aragonite from seawater as shown by Gaetani & Cohen 2006.

References:

Gaetani, G. A., & Cohen, A. L. (2006). Element partitioning during precipitation of aragonite from seawater: a framework for understanding paleoproxies. Geochimica et Cosmochimica Acta, 70(18), 4617-4634.

Referee comment: (2) Authors erroneously speak of outer and inner shell layer, but, in fact, they have only studied the outer shell layer, which in almost all bivalves is divided into two ultrastructurally different portions, i.e., the outer and inner portion of the outer shell layer (in the following, oOSL and iOSL). The inner shell layer (ISL) is located way back (below what is depicted in Fig. 4C) and (in a cross-sectioned shell) starts where the myostracum intersects with the inner shell surface (= aka pallial line) and ends somewhere at the hinge portion. In Figure 1B, the inner shell layer is formed approx. inside the brown areas, whereas the brown section and portions outside thereof largely belong to the iOSL; the oOSL is likely not seen in this image. The pallial line delimits the ISL from the iOSL. I recommend to look at Fig. 2A in Schöne (2013).

Answer: We are grateful to the referee for pointing out the intricacies of shell nomenclature. We did not mean to use zoological terminology in our ms, but merely strived to apply appropriate terminology to distinguish the structurally inner parts of the shell from the outer parts in the studied area of the shell at the shell tip. While we realize that this may have been misleading, it is not the aim of our study to describe the ultrastructure in its entirety across the shell. Our study targets the area at and along the ventral margin to the outside of the pallial line. We have now clarified this in the new version of the ms and use the appropriate nomenclature in agreement with the morphological elements of a bivalve shell.

Referee comment: (3) Surprisingly, a number of relevant recent papers dealing with very similar issues remain uncited. For example:

(3a) In-situ labeling: Mouchi et al (2013) labeled oysters with manganese to study growth rates, and Mouchi et al. (2016) used immunogold to obtain insights into biomineralization processes of Crassostrea gigas. Riascos et al (2007) tested three different stains in abalone and the surf clam, i.e., calcein, alizarin and strontium chloride.

(3b) Zhao et al (2017a) recently demonstrated that Sr/Ca in the outer shell layer of Corbicula fluminea increases proportionately to Sr/Ca in the ambient water and is not affected by growth rate effects. A very similar finding as reported here.

Answer: 'Labelling' methods have been around for decades and provide us with a powerful tool for many different purposes. It is therefore important to refer to the specific purpose rather than to generalize. The papers referred to by the referee above do not at all deal with "very similar issues" and we will clarify this in the revised ms.

The main aim in our study is to use Sr pulse-labelling as a marker to study the structure of the shell at the nano-micro scale. This variety of labelling, termed 'pulse-labelling', is an accepted method often used for corals, employing either elemental or enriched isotope spikes (e.g. Brahmi et al., 2012, Domart-Coulon et al., 2014). Pulse-labelling highlights growth features at the micro-nano-scale, which 'general' labelling is not able to and, thus, the further has an entirely different focus of study than the latter.

Instead, the labelling studies carried out by Mouchi et al (2013), Riascos et al (2007) as well as of Zhao et al (2017) aimed at growth rate determination at a much lower spatial resolution and the labels were analysed by means of different instrumentation, thus, were not carried out for the same purpose as our study. Similarly, immunogold labelling (Mouchi et al., 2017) is a routine method in protein chemistry used to label functional groups in specific organic molecules present in the shell. Unlike our study, it is carried out 'ex situ' and not on living bivalves.

Our speculations on the effect of growth rates on Sr/Ca ratios are a secondary result that warranted discussion, but this topic is in no way the focus of this study. It is interesting to see that our study apparently reproduced the observations of Zhao et al. (2017) on the lack of a growth rate effect on Sr/Ca and we have now referenced this article in the revised version. However, these authors used a very different and taxonomically unrelated bivalve species which, unlike the one we studied, lives in freshwater environments, and has a very different shell architecture. Therefore, this outcome, if correct, is not intuitive.

*References for this answer:*

Brahmi, C., Domart-Coulon, I., Rougée, L., Pyle, D. G., Stolarski, J., Mahoney, J. J. et al. (2012). Pulsed 86 Sr-labeling and NanoSIMS imaging to study coral biomineralization at ultra-structural length scales. Coral Reefs, 31(3), 741-752.

Domart-Coulon, I., Stolarski, J., Brahmi, C., Gutner-Hoch, E., Janiszewska, K., Shemesh, A., & Meibom, A. (2014). Simultaneous extension of both basic microstructural components in scleractinian coral skeleton during night and daytime, visualized by in situ 86Sr pulse labeling. Journal of Structural Biology, 185(1), 79-88.

Referee comment: (3c) An alternative mechanism of how the bivalve controls the trace and minor element levels in the shell – brought forward by Shirai et al. (2014) and based on Stephenson et al. (2008) – was also ignored: Organic macromolecules near the shell formation front exert control on which and how many ions are incorporated into the carbonate phase of the shells. If the overall production of biomass and thus growth rate decreases (e.g., during times of low food availability), less of such organic substances are produced and the level of trace impurities in the shell carbonate automatically increases. This in turn, affect the morphology of biominerals and likely explains the more primitive ultrastructure at growth annual and even daily growth lines (biochecks) (Füllenbach et al. 2017), i.e., irregular simple/spherulitic prismatic ultrastructure (Schöne 2013). Data in Table 1 also indicate that different microstructures in your study contain different Sr levels, likely for the very reason described above. However, you did not discuss this or the fact that the relative change in the iOSL is only ca. 14 times, not 18.

Answer to 3C discussed by theme:

(2) Growth lines, shell composition and architecture:

In contrast to the referee's statement, reduced growth rates in bivalve shells do not scale to all moieties (mineral and organic) in the shell. Many bivalve species with nacroprismatic shell structure, for example, form an annual growth line that is organic-rich (and poorly mineralized; e.g. Soldati et al., 2008), suggesting that these species independently downregulate the mineralization of the shell from the production of the organic moiety in times of slow growth. These organic-rich shell areas do not contain vastly differing trace element budgets compared to the more mineralized parts of the shell, demonstrating that there is nothing 'automatical' about this process that could be generalized across species. Shells of bivalve species that form a mineralized growth line (e.g. *Arctica islandica*) contain much less overall organic moiety compared to nacroprismatic shells (Non-nacreous: ca. 1-1.5 wt% vs nacreous: 3-5 wt%, Agbaje et al., 2017a, b, 2019). It would be interesting to see any *direct evidence* for a downregulated production of organic components in those bivalve species that form mineralized growth lines, rather than correlative speculation as presented in Füllenbach et al. 2017.

(3) Potential control of the shell architecture by organic macromolecules:

To date, there is no direct evidence for how the complexities of the bivalve shell ultrastructure are connected (if at all) to the composition and amount of organic molecules present in the shell. Instead, it is well known that the composition and amount of the organic moiety in shells varies significantly between species (Agbaje et al., 2017a, b, 2018, 2019, Currey et al 1976, Hare, 1965, Kamat et al 2000) and does so independently of the shell ultrastructure.

Compared to this direct evidence present in the literature, Füllenbach et al. (2017) base their model on how the ultrastructure of bivalve shells relates to the organic moiety on proxy analyses, namely S/Ca ratios in the shell analysed by EPMA. Direct characterization and analysis of the organic molecules in the shells is not presented in their study. Hence, this hypothesis brought forward by the referee is therefore highly speculative and suggestive at best.

(3) Potential control of trace element incorporation into the shell by organic molecules at the shell growth front

This topic is also part of the next referee comment and will be addressed below.

Following the referee's advice, we have incorporated these models developed by Stephenson et al. (2008) and the Schöne group in the revised version of the ms.

*References for this answer:*

Agbaje, O. B., Thomas, D. E., Mclnerney, B. V., Molloy, M. P., & Jacob, D. E. (2017a). Organic macromolecules in shells of *Arctica islandica*: comparison with nacroprismatic bivalve shells. Marine Biology, 164(11), 208.

Agbaje, O. B. A., Wirth, R., Morales, L. F. G., Shirai, K., Kosnik, M., Watanabe, T., & Jacob, D. E. (2017b). Architecture of crossed-lamellar bivalve shells: the southern giant clam (*Tridacna derasa*, Röding, 1798). Royal Society Open Science, 4(9), 170622.

Agbaje, O.B.A., Ben Shir, I., Zax, D.B., Schmidt, A**.,** Jacob, D.E. (2018) Biomacromolecules within bivalve shells: is chitin abundant? *Acta Biomaterialia* 80, 176-187; 10.1016/j.actbio.2018.09.009

Agbaje, O. B., Thomas, D. E., Dominguez, J. G., McInerney, B. V., Kosnik, M. A., & Jacob, D. E. (2019). Biomacromolecules in bivalve shells with crossed lamellar architecture. Journal of Materials Science, 54(6), 4952-4969.

Currey, J. D., & Kohn, A. J. (1976). Fracture in the crossed-lamellar structure of *Conus* shells. Journal of Materials Science, 11(9), 1615-1623.

Hare, P. E. (1965). Amino acid composition of some calcified proteins. Carnegie Inst. Washington Yearbk., 64, 223-232.

Kamat, S., Su, X., Ballarini, R., & Heuer, A. H. (2000). Structural basis for the fracture toughness of the shell of the conch *Strombus gigas*. Nature, 405(6790), 1036.

Referee comment: (4) The alternative mechanism of element incorporation mentioned in 2c does not require any control on uptake of elements. Although the chemical composition of the extrapallial fluids or gels (outer EPF forming the OSL, inner EPF the ISL) of marine bivalves have rarely been measured, the few available studies (e.g., Wada & Fujinuki 1976) unequivocally show that they have nearly the same ionic strength and chemical composition as the ambient seawater (Crenshaw 1972, Lorens & Bender 1980). This is no surprise, because bivalves are osmoconformers, like all other marine organisms. Imagine which energetic efforts were otherwise required if the bivalves had to constantly pump these ions out of the body fluids. Some elements such as strontium, magnesium and sodium reach the body fluids as ions from the ambient water through the gills and the gut (Wilbur & Saleuddin 1983) and across the mantle epithelium (passive diffusion). I have prepared a table for you summarizing data from Wada & Fujinuki (1976) (Table 1).

Table 1. Average element-to-Ca ratios in the inner extrapallial fluid of marine bivalves in comparison to seawater. Calculated from chemical data reported in Wada & Fujinuki (1976).

|  | Seawater | EPS during growth | EPS during resting |
|---|---|---|---|
| Na/Ca (mol/mol) | 44.3 | 44.1 | 42.4 |
| Li/Ca (mmol/mol) | 2.1 | 2.6 | 2.7 |
| Mg/Ca (mol/mol) | 5.0 | 5.1 | 4.9 |
| Sr/Ca (mmol/mol) | 8.3 | 9.4 | 8.0 |
| Mn/Ca (mmol/mol) | 30.2 | 291.1 | 223.9 |

Despite this, shells are strongly depleted in many trace and minor elements. For example, if measured with a spatial resolution of ca. 50μm Sr/Ca in aragonitic OSL of *Arctica islandica* ranges between ca. 1-3 mmol/mol and Mg/Ca remains below 0.8 mmol/mol (e.g., Schöne et al. 2011). Even when measured by much higher spatial resolution (nanoSIMS) which might be advantageous given the strong chemical heterogeneity of the shell at the μm-scale, Sr/Ca in aragonite of *Cerastoderma edule* does reach values expected for equilibrium fractionation (Füllenbach et al., 2017). In calcitic shells of various species, Mg/Ca ranges between ca. 4-28 mmol/mol (see summary in Vihtakari et al. 2016). These findings lend support to the hypothesis that unwanted elements are actively excluded from the shell by specialized organic macromolecules directly at the site of shell formation (Schöne 2013; Shirai et al. 2014). How this mechanism fits to the ACC-mediated shell formation processes needs to be discussed. Since the chemistry of body fluids of bivalves resembles that of seawater, there is no need for any active transmembrane element transport. Zhao et al. (2017b) recently demonstrated very clearly that Sr, Mg and Ba levels in shells of *Corbicula fluminea* were not transported by active transport mechanisms and did not use the same pathways as Ca. These authors have poisoned Ca2+ATPase and blocked Ca2+ channels.

According to the finding by Zhao and colleagues, a passive diffusion pathway across the mantle epithelium is much more likely and would perfectly fit to the incorporation control by organic macromolecules at the shell formation front. I strongly feel that these alternative explanations must be presented and discussed.

Answer: This section in the submitted version of the ms is very speculative, and this was also pointed out by referee #2. We will follow the advice of referee #2 to omit this section to remain closer to our robust and detailed results. Nevertheless, we welcome this opportunity to reply to the referee's comments above and to correct a number of flaws, inaccuracies and misconceptions articulated by the referee:

Contrary to the referee's statement, not all marine organisms are osmoconformers.

It is generally not helpful in this discussion to use poorly defined terms. This is even more relevant in the fundamentally interdisciplinary field of biomineralization where communication across discipline boundaries relies much on the correct usage of terminology. In this line of thought, terms such as "unwanted elements" which presumably refers to concepts of 'chemical fractionation' and 'incompatibility' rather than to an organism expressing its free will, and the term 'ACC-mediated' for a mechanism that produces metastable ACC as a transient precursor, but by no means as an active player that could actively 'mediate' any given process, are not furthering mutual understanding nor scientific progress.

We note that the biomineralization concepts articulated by the referee as well as in Füllenbach et al (2017) rely mainly on literature from the 1950 to 1980s. While many of the pioneering works in the field we are building on today have indeed been produced in this period of time, the field of biomineralization is very fast moving with rapid progress today being made mainly across chemistry, material sciences and physics. This large body of relevant literature is not captured in the referee's comments. One of the concepts, for instance, that experienced major revision is that of the extrapallial fluid (EPF), whose existence as a fluid with a defined composition is questioned today, to say the least. A valuable summary into the questionable nature and existence of the EPF is given in Marin et al. (2012), who state: "(…) its sampling is tricky. On different occasions, having done ourselves these experiments with a small syringe and a tiny needle on different model organisms, we were never fully convinced that the fluid that we were sampling was the right one! Furthermore, (…) it is likely that the composition of this fluid is not homogeneous, but varies from the central shell zone to the shell edge. Furthermore, it seems that the composition of this fluid also varies according to seasons." Following this reasoning we would challenge the referee's line of thought and suggest that the reason for why the table shows the composition of the EPF to be so similar to seawater is that its major component is indeed seawater, because the extrapallial space most likely is not fully sealed towards the outside.

Lastly, after carefully studying Zhao et al. (2017) we find that the reasoning presented there is mostly correlative and highly speculative, while direct evidence is rarely provided to underpin their interpretation. Furthermore, the study focusses on a freshwater bivalve species with different shell architecture from the species we studied and uses different analytical methods at much lower spatial resolution. As generalization at this level and across species is difficult, we would be interested to learn the reasons for how the results of this study would be relevant for our work and why the apparent agreement between a subset of our results with those of Zhao et al. (2017) would be more than a coincidence. Although we have significantly shortened and rewritten this section we were happy to reference Zhao et al. (2017).

*References for this answer:*

Marin, F., Le Roy, N., & Marie, B. (2012). The formation and mineralization of mollusk shell. Front Biosci, 4(1099), 125.

Referee comment: (5) Another argument against ATP-mediated uptake mechanism is unchanged growth rate of the bivalve. If the hypothesis by Otter and colleagues holds true according to which an "intracellular, diffusion driven, selective transport" of ions is responsible for the observed low Sr shell concentrations, then it is surprising that shell growth rate remained unchanged. A selective transport consumes energy = ATP), and the energy demand for such a transport process increases if the Sr level in the water rises. If more energy is devoted to the control of Sr incorporation into the shell, less energy is available for shell formation resulting in lower growth rate.

Answer: Metabolic processes regulating shell growth are complex and not yet fully understood. It is an interesting and intuitive suggestion by the referee that ATP driven transport results in lower growth rates. However, without direct evidence, there is no way to test this hypothesis. This highlights just how speculative this section of the manuscript was and supports us in the decision to have followed the advice given by referee #2 to cut this section completely.

Referee comment: (6) There is a confusing usage of the term "uptake" (e.g., P2L8). 'Uptake' refers to way elements take from the environment to body fluids. This can either occur through mantle epithelia (in ionic form, potentially by one of the pathways listed in your paper) or during digestion of food. Is this really what you mean here on page 2 or rather the 'incorporation' of elements into the shell at the site of shell formation? From the context, I assume you meant the latter: "Recent studies showed that the uptake of some trace elements, such as strontium, are strongly influenced by crystal growth rates, shell curvature and ontogeny in addition to physiological effects".

Answer: We agree with the referee that, to differentiate between 'uptake' from the water and 'incorporation' into the shell, it is more accurate to use 'incorporation' when referring to shell formation and have replaced it as suggested.

Referee comment: (7) A number of observation were only presented, but not discussed and combined with other aspects of the study, e.g., different amounts of organics in different ultrastructures.

Answer: We believe our discussion of the organic contents in different shell architectures is sound and fully based on the evidence provided in the ms.

Referee comment: (8) Interpretation of the timing of shell growth, meaning of microgrowth increments (= daily), major biochecks (= annual) and greyscale changes (= fortnights) is purely speculative and not supported by the data presented. This would require mark-and recovery experiments. Though not unlikely that the regular change in greyscale results from fortnightly changes, you need to cite at least relevant papers dealing in detail with such tide-controlled growth patterns (Evans 1972, Ohno 1989, Schöne 2008, Hallmann et al. 2009). B the way, you did not say where the bivalves lived: in the intertidal zone? You also noticed that you observed 6 lines in portions formed in tanks during 6 (solar) days suggesting that at least these growth patterns are circadian. However, you have no evidence that the same applies to shell portions formed in nature. Given that the specimens lived in the intertidal zone (please provide details on tidal regime: diurnal or semidiurnal, tidal range etc.), it is reasonable to assume that they have formed circalunidian growth patterns (lunar days). Perhaps, acclimatization to circadian lab conditions were sufficient to reset biological clock resulting tin switch from lunar to solar daily. However, all this needs some discussion (in the Discussion section, not results as currently presented).

Answer: In contrast to what the referee understands, the greyscale patterns in the shells the referee refers to here (Fig. 1D, E) were not formed during aquaculture, but are growth features of the shell formed in the wild before shells were transferred to the aquarium. Our interpretation of these as time gauges for shell growth is therefore valid. We incorporated the tidal regime into the results as well as the materials and methods section. As already stated in the discussion section of the manuscript these bivalves live in the intertidal zone. Suggested relevant literature has been included and the part regarding greyscale line profiles of shell grown in the natural environment has been moved to the discussion as suggested.

Referee comment: (9) Since you are aiming to publish your paper in a journal that is often read by people of the proxy and paleoclimate communities, you need to translate oxide values into element concentrations (as well as molar element/Ca data), and all element/Ca data into molar ratios (required for easier, direct comparison with published data). Likewise, instead of reporting Ca/Sr ratios, please turn this around and give Sr/Ca data.

Answer: Although traditionally EPMA data is presented as wt.% oxide we have moved this table as it is to the supplement and replaced it in the ms with elemental concentrations provided as $\mu g \cdot g^{-1}$ (omitting oxides). In addition, we have added molar element/calcium ratios to this table.

Referee comment: (10) I do not think your results allow any conclusions on whether higher Sr levels in water have or have not affected shell growth rate. If growth conditions remained invariant (aside from changing Sr levels), shells should have grown much more homogeneously. But in fact, there is a significant slowdown from LE1 over NE1, LE2 to NE2 suggesting that growth conditions deteriorated through time (Table 2).

Answer: We meant to articulate here that, while there is clearly a number of factors affecting shell growth in aquaculture, incorporation of Sr into the shell aragonite does not significantly affect growth rates in our

experiment. This is evident from Figure 6C, which compares Sr-labelled and unlabelled growth increments. This figure shows clearly that all data lie within the standard deviation of the average and differences are insignificant, however we have changed Table 2 values to daily growth values (as suggested further down by this reviewer) and believe this makes our point more clear.

**Minor comments:**

Referee comment: Please check orthography in entire ms. I am not familiar with the Australian English, and whether this represents a mix of American English (e.g., analyze, labelling, meter) and British English (analyse, labelling, metre).

Answer: We edited all the text (manuscript and supplement) to British English as outlined in the journal's author guidelines.

Referee comment: Consistent use of hyphenation is required in entire ms: crossed-lamellar, crossacicular, 3 mm-thick, high-resolution, crossed-lamellar, crossed-acicular, organic-rich etc. need a hyphen

Answer: Agreed and edited accordingly.

Referee comment: Headings: Consistently capitalize heading or use sentence case.

Answer: Agreed to use only sentence case.

Referee comment: No colon at the end of headings! E.g., P8L21: "The inner crossed[-]acicular [shell] layer:", P9L1, etc.

Answer: Agreed and edited accordingly.

Referee comment: P1L16, "aragonite crystals": As you noticed in the following sentence, "the smallest mineral units are nanogranules" which are enveloped by proteinaceous materials. I suggest to employ the term "mesocrystals", because the definition of an abiogenic aragonite crystal does not include nanocomposites consisting of aragonite and organic material.

Answer: Unfortunately, the referee's definition of the term 'mesocrystal' is not correct. Correctly, the term 'mesocrystal' refers to hybrid inorganic-organic nano-blocks that are aggregated to a crystal which exhibits the X-ray properties of a single crystal at the mesoscale (Cölfen and Mann, 2003). Or, as most recently defined by Bergström et al. (2015): **"a nanostructured material with a defined long-range order on the atomic scale, which can be inferred from the existence of an essentially sharp wide-angle diffraction pattern (with sharp Bragg peaks) together with clear evidence that the material consists of individual nanoparticle building units"**. Whether, or not some, or even all nanogranules are mesocrystals cannot be established here and is beyond the scope of the ms.

*References for this answer:*

Bergström, L., Sturm, E. V., Salazar-Alvarez, G., & Cölfen, H. (2015). Mesocrystals in biominerals and colloidal arrays. Accounts of chemical research, 48(5), 1391-1402

Cölfen, H., & Mann, S. (2003). Higher-order organization by mesoscale self-assembly and transformation of hybrid nanostructures. Angewandte Chemie International Edition, 42(21), 2350-2365.

Referee comment: P1L19, replace "shells" by 'shell portions' or 'ultrastructures'. There are no bivalves consisting entirely of nacre. I assume you intended to say that different ultrastructures contain different amounts of organics.

Answer: Agreed and replaced with "ultrastructures".

Referee comment: P1L19/20: I do not understand this sentence. Growth rates = growth patterns? Outer structure = outer shell layer. Prisms can be correlated to growth rates? Do you mean that each 3rd order prism forms in one day? Moreover, you did not mention anywhere in the text sub-daily growth patterns.

Answer: The timing of formation or the 3$^{rd}$ order prisms is mentioned in the text at P12L29: "while nanometre-sized third-order prisms form within hours (Fig. S6)." We have edited this sentence for clarity.

Referee comment: P1L20, "outer structure": You used the term "structure" in two different ways: as a synonym for "ultrastructure" and "shell layer" (e.g., P6L32). Be consistent. Do not use "structure", but one of the other terms above. Check and change throughout ms.

Answer: we agree and have replaced "structure" with either "ultrastructure" or "shell layer" (depending on the context) throughout the ms and supplement.

Referee comment: P1L23, "physiological processes during calcification have no lag": Rephrase, this is hard to understand. Shells do not just consist of CaCO3, but also organics which need to be fabricated, and the building blocks for these substances derive from ingested food. Digestion of food and fabrication of organic molecules that end up in the shell need time. There is hence a lag between ingestion of food and shell production. Or what do you mean with "physiological processes . . . have no lag".

Answer: We have edited this for clarity and down-toned the timing as we agree with this reviewer that there is most likely 'some' lag, reflecting 'some time' but that this lag is not significant enough to be quantified with our methods (see also comment to P13L29 below).

Referee comment: P1L23, "calcification" is the wrong term here (and used improperly in many other studies). Calcification rate includes density and is not synonymous to growth rate! Calcification rate = amount CaCO3 precipitated per time interval per area. Replace all instances with 'shell growth rate'.

Answer: In our study we were following the terminology as defined in Gillikin et al. (2005): "Considering that we discuss our results in the context of calcification processes, the distinction between growth rate and calcification rate should be made. In this study, the term growth rate is defined as the dorso-ventral linear extension of the shell per unit time (or growth increment per time)". "Calcification rate" and "crystal growth rate" are often used synonymously in sclerochronology to refer to the narrow growth increments and architectural arrangements at higher magnification of shell layers that are angled to the macroscopic dorso-ventral linear shell extension. However, "crystal growth rate" is in fact incorrect as "crystal" does not cater to the mesocrystalline nature of the material that is initially formed as ACC. Therefore, we use "local growth rate" to describe growth rates at higher resolution within different layers of the shell that are measured parallel to e.g. the long axis of first order prisms and that can be at angles with the macroscopic dorso-ventral linear shell extension. We have edited the corresponding paragraph in the manuscript and have replaced "calcification rate" with "local growth rate" all throughout the ms and supplement.

*Reference for this answer:*

Gillikin, D. P., Lorrain, A., Navez, J., Taylor, J. W., André, L., Keppens, E., ... & Dehairs, F. (2005). Strong biological controls on Sr/Ca ratios in aragonitic marine bivalve shells. Geochemistry, Geophysics, Geosystems, 6(5).

Referee comment: P1L25, "Sr-conditions": no hyphen; 'Sr level' or 'Sr concentration' sounds better

Answer: Agreed and edited accordingly.

Referee comment: P1L26, "Sr-enrichment": no hyphen

Answer: "Sr-enrichment factors" requires a hyphen. According to the Oxford Dictionary, a hyphen is required to link two words that function together as an adjective and that are placed before the noun they're referring to, hence, no change here.

Referee comment: P1L26, "Sr-enrichment factors for labelled and ambient conditions": This remains insufficiently explained and is oddly phrased. Do you mean artificially elevated Sr levels vs. normal marine Sr levels? Give actual numbers! What do you mean with "identical enrichment factors": Sr levels in shell increase proportionately to that in the water (i.e., 18 times)? As far as I can tell from Table 1, this does not apply to both shell layers (and ultrastructures).

Answer: As we have replaced the 'enrichment factors' with 'distribution coefficient' (Sr/Ca-shell)/(Sr/Ca-seawater) as suggested by reviewer 2 this sentence had to be edited anyway to reflect the new data. Also, we have followed this referee's advice to give actual numbers.

Referee comment: P1L31, "aragonite or calcite": replace "or" by 'and/or'. Note there are species with different CaCO3 polymorphs in the outer and inner shell layers. Further note that some species also come with vaterite,

Answer: Agreed and edited as suggested.

Referee comment: P2L3: delete "recent and fossil", superfluous

Answer: Agreed and edited as suggested.

Referee comment: P2L4+5: None of these papers used trace elements of shells as environmental proxies. Replace by suitable citations: (a) temperature: Klein et al. (1996a), Wanamaker et al (2008), Schöne et al. (2011), Zhao et al. (2017a). (b) salinity: Klein et al. (1996b). (c) pH: Zhao et al. (2017c)

Answer: We replaced the references with the suggested ones.

Referee comment: P2L12: substitute "shell" with 'trace and minor elements in shells'

Answer: Agreed and edited as suggested.

Referee comment: P2L14: substitute "but" with 'and'

Answer: was replaced with "however".

Referee comment: P2L14: Firstly, always say 'ultrastructure', not "structure", because at other places you use "structure" as a synonym for shell layer. Secondly, this statement needs a reference.

Answer: References are already given in the text (line 15) we replaced "structure" with "ultrastructure" were appropriate throughout ms and supplement.

Referee comment: P2L15-16: Delete sentence starting with "Apart. . .". Then start next sentence with "Apart from those,"

Answer: No change

Referee comment: P2L17: replace "which are found" by 'which occur'

Answer: edited as suggested.

Referee comment: P2L21: The homogeneous ultrastructure forms an own category and is not a subgroup of the crossed-acicular category (compare Marin et al. 2012)

Answer: We understand that there are different schools of thought. In the current version of our ms we have followed Shimamoto et al. 1986: "(…) homogeneous structure in the present study is used in broader sense

including crossed acicular and/or fine complex crossed lamellar structure of Carter (1980) (…)". Indeed, Marin et al 2012 state that "[crossed structures] represent a diversified group comprising the crossed-lamellar, complex crossed-lamellar, crossed acicular microstructures, found in most of the heterodont bivalves and in several gastropods". We have acknowledged both schools of thought in the revised version of the manuscript.

Referee comment: P2L21: "venerid" must not be italicized

Answer: edited as suggested.

Referee comment: P2L22: "Shimamoto, 1986" is outdated (?), check most recent revision of ultrastructures by Carter JG et al. (2012)

Answer: We checked Carter et al. (2012) and found that Shimamoto 1986 is not outdated in this aspect.

Referee comment: P2L33: replace "between umbo and ventral margin" by 'parallel to the main growth axis' or 'parallel to the umbo-ventral margin axis'

Answer: edited as suggested.

Referee comment: P3L5-6: "Growth lines. . ." show/refer to figure

Answer: We have added a reference.

Referee comment: P3L16: Two main clauses combined by conjunction require comma; check and correct throughout ms: ', and'

Answer: edited as suggested.

Referee comment: P3L17: Specimens: Much more information needed here: sediment type, tidal height, intertidal zone(?), how many specimens collected/prepared/used for which analytical technique, when collected. Table would be best. Part of this information is relevant for the temporal alignment of the shell growth patterns.

Answer: Referee has later (below) accepted our reasoning for the time gauge. We added the requested details to the corresponding paragraph.

Referee comment: P3L17: replace "live-collected" by 'collected alive'

Answer: edited as suggested.

Referee comment: P3L20: use 'x' as mathematical operator (consistently throughout ms)

Answer: edited as suggested.

Referee comment: P3L24-26: Has the element composition of the food been measured as well? How do you know that all Sr and Ca comes from the water? Has always the same amount of food being offered? When were they fed, during simulated day or nighttime?

Answer: We did not measure the Ca and Sr content of the diet. Food was added in the morning and we observed that the water had cleared up by the end of the day (after ca. 6 hours). This indicates an extended time of filter feeding (food uptake). If significant Sr and Ca were derived from the diet we would expect concentration differences (visible in maps) in unlabelled areas between shell portions formed during day and night. This was not observed.

Referee comment: P3L29-30: An "event" is a very short-term incident. This sentence should be rephrased, e.g., "exposure to background conditions, i.e., normal marine Sr levels".

Answer: rephrased as suggested. "Event" was chosen as it indeed refers to very short periods in accordance to what the reviewer says.

Referee comment: P4L7: P400-P2000

Answer: edited as suggested.

Referee comment: P4L12: thickness of gold-coating?

Answer: we added the thickness of the coating (15 nm) in the ms.

Referee comment: P4L21: 20,000x

Answer: edited as suggested.

Referee comment: P5L9: replace "was used" by "were used"

Answer: edited as suggested.

Referee comment:  P5L21: µm2 (superscript)

Answer: edited as suggested.

Referee comment: P5L27 "The inner and outer layer of a *K. rhytiphora* shell were separated with a DREMEL tool and mechanically cleaned." Be more specific here: Have you obtained powdered material or fractions of the two portions of the outer shell layer? How have you managed exactly to separate them?

Answer: We have clarified this in the revised version of the ms

Referee comment: P6L3: Actually wrong. You have only studied the outer shell layer, which consists of two portions with different ultrastructure, an outer and inner portion, respectively (oOSL, iOSL)!

Answer: We have replaced "inner layer" and "outer layer" with the nomenclature suggested by this reviewer throughout the ms and supplement.

Referee comment: P6L8: Rephrase (and italicize genus and species names): 'The outer shell layer of studied *K. rhytiphora* specimens is ... near the ventral margin'

Answer: edited as suggested.

Referee comment: P6L10: "in agreement with previous studies**…**" This phrasing means that the other species studied by Carré and Soldati and colleagues lived in Australian waters. Rephrase.

Answer: edited to clarify that these are literature examples from the southern hemisphere but not from Australia.

Referee comment: P6L11: "growth periods": delete "periods"

Answer: edited as suggested.

Referee comment: P6L13: "troughs" Odd phrasing. Something like this is better: 'Cyclic changes in greyscale near the ventral margin correlate strongly with tidal cycles, i.e., light grey and dark grey portion fall together with full and new moon cycles, respectively. The main problem is that you do not provide any evidence for the timing of shell growth! Where is the evidence that the dark and light portions really have formed during new and full moon? This is an interpretation at most, and as such belong to the Discussion section (where you need to refer to previous studies of intertidal bivalves which found narrower increments and thicker growth lines formed during spring tides, and these portions then appear darker than shell portions formed during neap tides when viewed at lower magnification and under reflected light. More suitable Refs: Evans 1972, Schöne 2008, Hallmann et al. 2009)

Answer: We have moved these paragraphs to the discussion and have edited the text as suggested.

Referee comment: P6L16-25 also needs to be moved to Discussion. Only keep descriptive part here. You have no evidence that these grey bands formed on a circalunidian basis, but you can certainly interpret them as such based on previous work.

Answer: Moved as suggested.

Referee comment: Timing of shell growth: You later noticed that you observed 6 lines in portions formed in tanks during 6 days suggesting that at least these growth patterns are circadian. However, you have no evidence that the same applies to shell portions formed in nature. Given that the specimens lived in the intertidal (please provide details on tidal regime: diurnal or semidiurnal, tidal range etc.), it is reasonable to assume that they have formed circalunidian growth patterns. Here, please stick to descriptions, not interpretation.

Answer: Agreed – we revised this part and added that the shell areas formed in the natural environment perhaps follow circalunidian growth patterns.

Referee comment: P6L25: 'in two other specimens', not "on two other specimens"

Answer: edited as suggested.

Referee comment: Section 3.2: Title is more suitable for Discussion. – This section should be expanded as it is an essential component of the ms and forms the basis for your hypothesis on element incorporation. Describe Table 1 in much more detail. Report molar ratios as well. Compute and tell reader by how much the Sr levels increased in the shell when exposed to 18 times higher Sr levels in water. This will then show that the Sr levels in oOSL increased by 18 to 20 times, whereas the iOSL only by ca. 14 times. This needs to be discussed later.

Answer: The content of Section 3.2 "Validation of Sr incorporation" is a result and is thus indeed suitable for this section (no change). We now provide molar ratios in Table 2 and discuss these now is this section as well. We have also added the differences between labelled and unlabelled shell as well as the amount that Sr in the shell increases during labelling conditions.

Referee comment: P7L2: ". . .were identical within uncertainty": i.e., they have remained invariant, stayed the same? I suggest you rephrase this to avoid confusion.

Answer: This is geochemically the correct terminology and means within *analytical* uncertainty.

Referee comment: Title Section 3.3: Section heading should inform about content of section, not which method has been used.

Answer: This section contains the Raman spectroscopy results and is, as such, correctly titled.

Referee comment: P7L31: "This species develops annual growth checks" On what evidence is this statement based? How did you analyze when the shell portions formed? Likely correct, but pure speculation... or is there previous work on this species?

Answer: Referee has below accepted our reasoning for the time gauge. No change necessary.

Referee comment: Section 3.4.3: Interesting information, but what is the purpose of having this measured and reported?

Answer: The amounts and composition of the organic moiety are not well known, particularly for non-nacreous shells. We are presenting new results here with the purpose of closing this knowledge gap. We note that the referee finds these results interesting.

Referee comment: P9L10: crystallographically

Answer: "crystallographic preferred orientation (CPO)" is the correct name of this type of data. No change.

Referee comment: P9 "Calcification Rates" includes density, not synonymous to growth rate! Calcification rate = amount CaCO3 precipitated per time interval per area; this is not what you mean.

Answer: see our answer above.

Referee comment: P9L28-29: "Due to the geometry of first order prisms without- and inward bending in cross-sections,. . ." No sentence

Answer: This sentence has been edited for grammar and clarity.

Referee comment: P10L2: Provide image showing where you determined increment widths, or even better trace two growth lines to show that growth in oOSL is faster than in iOSL due to shell geometry.

Answer: These images are already in the supplement.

Referee comment: P10L4-5: quite complicated phrasing: absolute growth rates vary among specimens

Answer: edited as suggested.

Referee comment: P10L5: grew, on average, 5.6... same for the other "on average": separate by comma and place before number

Answer: edited as suggested.

Referee comment: P10L12: "Also, rates tend to decrease effectively with increasing distance to the ventral margins (Fig. 4A).": Unclear what you mean and purpose of mentioning this. You need to trace fortnights in Figure 4A to support your statement.

Answer: We have edited this part for clarity.

Referee comment: "bivalve species": you listed genera not species rephrase: ... structure of other bivalves, e.g., Pinna..., the aragonitic...

Answer: edited as suggested.

Referee comment: P10L23, P11L5, P11L23: "In *K. rhytiphora* the first order prisms" comma after species name

Answer: changed as suggested.

Referee comment: P11L10: Equally-sized (adverbial usage)

Answer: changed as suggested.

Referee comment: P11L30: Since you did not capitalize "aragonite", you should also use lower case here (except for the acronym/abbreviated form). Besides that, you used lower case in the Abstract.

Answer: changed as suggested.

Referee comment: P12L30: replace "the outside of their shells" by 'outer shell surface (Fig. 1), and'

Answer: changed as suggested.

Referee comment: P13L1: "growth time": Firstly, you have no evidence that these growth checks formed annually. Secondly, no bivalve grows 365 days. Note also that such ornamentation patterns do not agree with growth patterns in other species, and likely this is a coincident and only true for shell portions near the ventral margin in the studied specimens. Rephrase.

Answer: Referee comment below retracts this one – hence no change. However, we seized this opportunity to emphasize that we are aware that no bivalve grows 365 days with the same rate but that this was a reasonable estimate that yielded a result in the correct dimension.

Referee comment: P13L12-13: Perfect! This is your time gauge. It verifies the circa daily nature of these growth features and could further be used to support your hypothesis of fortnightly growth bundles appearing as greyscale changes.

Answer: Thank you we have added a sentence suggesting the use of the increments to support our hypothesis of fortnightly growth bundles.

Referee comment: P13L14-15: replace "higher" with 'faster', "short" with 'narrow', "longer" with 'broader'

Answer: changed as suggested.

Referee comment: P13L15: "day": An interesting question that you need to discuss is that these are probably circadian (24h) periods entrained by the 12/12 light/dark cycle experimental conditions. The adjustment interval was probably long enough that the natural, tide-entrained shell formation cyclicality (resulting in circalunidian, 24.8h, periods) vanished. Under natural conditions though, you would need to have circatidal (12.4h) and circalunidian increments, because otherwise your interpretation of the other 48 or 50 dark cycles representing fortnight periods would not hold true.

Answer: We agree and have added this to the discussion of our data in the revised version of the ms.

Referee comment: P13L22: "We suggest a diel physiologically controlled variation of calcification" Not sure exactly what you mean. Circadian clock controlling growth/calcification rate? This has been reported previously elsewhere.

Answer: This part has been rewritten for clarity.

Referee comment: P13L29 "have no lag"? Well, this depends on the temporal scale you are looking at. Where is the evidence that there was no gradual increase in shell Sr levels during the course of minutes or so? Diffusion of Sr through the mantle epithelium takes at least some time.

Answer: See answer above – we edited this sentence for better temporal accuracy.

Referee comment: P13P29"physiological processes involving Sr incorporation", rephrase: 'physiological processes controlling Sr incorporation'

Answer: changed as suggested.

Referee comment: P13L30-31: I do not think that the implications provided are supporting an ACC mediated growth of shell in bivalves.

Answer: We rephrased this sentence accordingly.

Referee comment: P13L33-34: "A fundamental observation of this study is that the calcification front runs evenly across all structural units and architectural orders of the shell independently of the current growth rate. This" But this is known and no a new finding of this study!

Answer: We disagree. This was not known for compound composite prismatic and crossed-acicular shells and is an entirely new finding (we emphasised this aspect now more clearly in the revised version of the ms). We note the lack of literature evidence for the referee's statement here.

Referee comment: P14L1: "show the labels to cut across the different architectural building blocks": could also occur if extrapallial space is gel-filled or epithelial cells are in direct contact with shell

Answer: Agreed.

Referee comment: P14L2 "where the label would rather follow a zig-zag trend between fully labelled and unlabelled units" Impossible to understand what you intend to say here. Rephrase please. Do you mean that the growth front is uncoupled from the ultrastructures? This is known as well: In freshwater bivalves the large prisms continue to grow over many years and daily growth lines cross them perpendicularly (studies by Dunca, Mutvei etc.).

Answer: This part of the text has been clarified as we do not refer to the uncoupling of the ultrastructure. We disagree that this was already known. This was not known for compound composite prismatic and crossed-acicular shells.

Referee comment: P14L2-4: "This is clearly visible from the sharply defined change between labelled and unlabelled shell areas (Fig. 4B and D), as well as from the cyclic variations in short-term growth rates (discussed above). Our" Likewise hard to understand

Answer: has been revised.

Referee comment: P14L10 "active selective transport consuming Ca2+-ATPase enzymes": transport consumes energy which is provided by ATP, and the enzyme that accomplishes the transportation is the Ca ATPase. Rephrase.

Answer: This part of the ms was omitted as we agreed on shortening this chapter.

Referee comment: P14L14-15 "We observed virtually identical enrichment factors for Ca and Sr (CaShell/CaSeawater and SrShell/SrSeawater) in labelled and ambient conditions (Table 3).": Interesting point

of view! But this does not mean anything else than Sr/Ca shell increases proportionately to that of Sr/Ca seawater, and this has already been shown by Zhao et al. (2017), which you did not cite.

Answer: This section has been extensively rewritten according to insightful comments from reviewer 2 and the citation has been incorporated.

Referee comment: P14L15: "Sr-ion transport is independent from. . . " if so, the energy demand of the bivalve increases in order to keep the Sr out of the shell. Do you see a decrease in growth rate during Sr enrichment as opposed to 'normal' Sr levels in water?

Answer: This section was deleted.

Referee comment: P14L16-17: "Sr ion would be at the expense of a Ca ion": Not really clear what you mean; Since this is the essence of your paper, you need to describe this more clearly and convincingly. Why exactly can transport mechanism 1 not be true?

Answer: This section was cut upon revision.

Referee comment: P14L17: Replace "Sr-enrichment" by 'shell Sr concentrations'

Answer: changed as suggested.

Referee comment: P14L18: Ca/Sr: please also or only report Sr/Ca

Answer: We now use Sr/Ca ratios in the ms.

Referee comment: P14L19-20: Replace "Hence, the strong enrichment of Ca from seawater to shell" by 'strong enrichment of Ca in shell'

Answer: changed as suggested.

Referee comment: P14L25: "Ca to be transported as ACC-nanogranules to the calcification front (Loste et al., 2004; Addadi et al., 2006; Jacob et al., 2011; Zhang and Xu, 2013)." Check if all cited studied were using bivalves (not gastropods or other taxa), and which ultrastructures were analzed, report this here.

Answer: The point is here that ACC-nanogranules have been observed across ALL TAXA to be part of a common principle of biomineralization. We would be happy to refer the referee to review studies who do exactly what is suggested here, but which is beyond the scope of our study.

Referee comment: Section 4.5: Here you discuss more (and different stuff) than what the heading implies.

Answer: We disagree. All aspects and 'stuff' discussed here belong under this heading.

Referee comment: P15L12: italicize genus and species names

Answer: edited as suggested.

Referee comment: P15L14-15: "a systematic change in growth increments during Sr-enriched periods": Do you mean 'growth increment widths'? You need to highlight here again that food levels and other extrinsic factors that could potentially have affected growth rate remained unchanged during the experiment, and you would have expected invariant increment widths if Sr had no effect on growth rate... see comment further above on relationship between growth rate and Sr exclusion from shell

Answer: We have revised this part of the manuscript (yes we meant widths) and have added the information suggested by this reviewer.

Referee comment: P15L18: Replace "calcification" by 'growth rate'

Answer: Replaced with "growth".

Referee comment: P15L23-24: "Reduced growth rates in aquaculture conditions cannot be explained by ontogenetic trends alone but result from missing tidal cycles." Sorry, but this is pure speculation and likely wrong. Much more likely is that you did not provide proper food and the animals did not really 'like' the tank conditions.

Answer: It is well known that shallow water depths lead to reduced growth rates in intertidal bivalves and, hence, the aquaculture experiments (in which tides are difficult to produce) can be understood to have similar but prolonged effects comparable the low water levels in nature. We have added a citation and edited this sentence for clarity. Also the possibility of a switch from circalunidian to lunar days has been now mentioned and the sentence was toned down to show that we speculate this to be true. The reviewer's suggestion that we haven't provided "proper food" is completely groundless as we have purchased special diet that is distributed as one of the most high quality shell fish diets commercially available and that has been used in other studies and commercial aquacultures. Also, although growth rates were downregulated compared to nature they were stable within the experimental period as shown in this study (e.g. Table 2).

Referee comment: P15L30: 'nanometer'?

Answer: British English, no change.

Referee comment: More comments in pdf with annotated figures and tables.

Answer see below.

**Reviewer Comments from the Supplementary Information:**

Referee comment: [To Fig. 1] I recommend removal of boxes around letters A, B, C...

Answer: We thank the referee for this suggestion. No change.

Referee comment: [To Fig. 1] FYI: The inner shell layer is formed inside the brown areas, whereas the brown section and portions outside thereof largely belong to the iOSL; the oOSL is likely not seen in this image.

Answer: Agreed as already clarified above, shell layers have been renamed.

Referee comment: [To Fig. 1] The pallial line is the small indentation ca. 1.5 cm away from the ventral margin on the inner shell surface. The pallial line strikes out again at the cardinal tooth (hinge; likewise a small kink developed). The inner shell layer is formed inside the portion delimited by the pallial line 'strikeouts'.

Answer: This is correct.

Referee comment: [To Fig. 1] Outer (A) and inner shell surface (B)...

Answer: edited as suggested.

Referee comment: [To Fig. 1] Denote: Arrows cannot be summers!

Answer: edited as suggested.

Referee comment: [To Fig. 1] Low or high values? dark or light grey?

Answer: clarified.

Referee comment: [To Fig. 1] Temporal alignment is purely speculative. Though I believe it is correct, you did not provide any convincing support this.

Answer: See referee comment above where they backtrack on this comment

Referee comment: [To Fig. 2] Captions should explain things. You fail to say what the pink and blue actually means: Sr enrichment in shell during immserion in Sr-enriched (give molar value) tank water and Sr shell levels during times of normal marine Sr conditions of 8.9 mmol/mol.

Answer: Agreed and clarified.

Referee comment: [To Fig. 2] Indicate DOG.

Answer: Edited as suggested.

Referee comment: [To Fig. 2] That are the inner and outer portions of the OUTER SHELL LAYER; B is not the inner shell layer!

Answer: Agreed and edited accordingly.

Referee comment: [To Fig. 2] This does not look polished, but etched.

Answer: This is a polished surface that received a final physical and chemical polishing step as explained in our methods section.

Referee comment: [To Fig. 2] You need to provide a schematic of the shell to show where exactly the images in Figures 2-5 were taken (hinge or ventral margin, where in the ventral margin?). As outlined below you only sampled the inner and outer portions of the outer shell layer, but not actually the inner shell layer.

Answer: In our first caption we state "All cross-sections in this study are prepared as radial sections along the maximum growth axis unless otherwise specified."

Referee comment: [To Fig. 3] Denote DOG

Answer: Edited as suggested. We have added and/or edited all DOG errors in the ms and supplement as well to make it easier to distinguish between the overall growth direction of a shell and local growth directions as visualised by the label to make it less confusing for the reader.

Referee comment: [To Fig. 4] That's incorrect! You are still in the outer shell layer here. These are just two different portions of the outer shell layer. The inner shell layer is way back (below in this image) and starts where the myostracum intersects with the inner shell surface (= aka pallial line) and then ends somewhere at the hinge portion. Shell material is added laminarily along the entire inner shell surface beyong the pallial line and results in thickening of ontogenetically younger shell portions (near the umbo and hinge). The inner portion of the outer shell layer actually does the as the outer portion: contributes to size increase of the entire shell, but in addtion also contributes to thickening. I recommend to look at the Figure in Schöne (2013) for morphological details of a veneroid.

Answer: Already addressed above.

Referee comment: [To Fig. 5] I am lost here. Is this image mirrored and upside down? Can you just present it in the same way as Figure 4? And please describe here again which portion on the two sides of the dotted line is the inner (left of line?) and outer portion of the outer shell layer (right of line).

Answer: We have clarified the orientation of the map by adding a schematic of the shell tip to all EBSD maps in the ms and the supplement (as well as in some other figures to maintain clarity. We chose to present other EBSD maps in ms and supplement that we think are easier for the reader to understand. We have simplified the figures also by reducing the number of white arrows that indicate the overall shell growth direction to one as is common practice in other publication and use the pole figures and appearance of the Sr-label in the underlying BSE images to point out the local growth directions in the corresponding EBSD maps.

Referee comment: [To Fig. 6] agree with what? each other? Rephrase. Within each shell layer, growth rates are unaffected by exposure to higher Sr content in water. How have you actually tested this mathematically? Student t-test? Normal distibution test done?

Answer: The information that the four different increments in 6C are within errors is based on their positions of the diagram. We have rewritten this part of the caption to make it more accessible to the reader.

Referee comment: Table 1: without considering Ca... 18-20x oOSL, 13-14x iOSL, Why different? Why does only Sr level in oOSL change proportionately to Sr increase in water?

Answer:  Firstly it is a general observation that values are higher for oOSL compared to iOSL as we state in the ms. It could be speculated that this is caused by the different overall growth rates between the two shell layers combined with the curvature of the shell.

Referee comment: Table 2: I do not understand "or". Which of the data in this column are 12d and 6d? For direct comparison of data, provide daily growth rates for all data.

Answer: The referee may have missed the explanation already provided (coded by italicised and normal text). We have now used a clearer coding (bold and normal). Daily growth are now provided as suggested.

**Referee #2:**

We thank this referee for their constructive comments, which will provided a helpful basis for the revision of our ms.

We appreciate that the referee feels that the way of presenting our large dataset is adequate. The referee noted that the ms is more on the descriptive side, and this is indeed a main in our work as this shell architectural type has never been studied at the micro to nano scale before. We see the work presented in this ms as a baseline for further work carried out on this type of shell architecture in the future, which necessarily needs to build on a detailed description.

In the following we are listing the referee's comments followed by our answer.

Referee comment: Description and interpretation of the data which related to crystallography and biomineralization seems to be OK, however, discussion about elemental transportation was based on very weak evidence thus problematic. Especially, the evidences the authors based on is (1) fluctuation of gray contrast observed at the growth portion during the Sr-enriched labelling experiments obtained by BSE image, even though BSE contrast is unreliable method for quantifying Sr concentration, and (2) similar enrichment factor (Shell/Seawater ratios) in labelled and non-labelled conditions in both ultrastructural layers aquired by EPMA analysis, while the way for presentation of this enrichment factor is not adequate for discussing the element transport. Because most of the discussion regarding biomineralization is good quality, and because the length of the MS is already enough, so I recommend to simply delete the contents related to element transportation.

Answer: We agree that the discussion about elemental transportation is a more speculative part of the manuscript and has been rewritten considerably. We also agree that BSE imaging per se is an unreliable method of quantification. This is exactly the reason why we went to great length to calibrate the grey scale of the BSE imaging by combining it with quantitative electron microprobe measurements using Wavelength

Dispersive Spectrometry (WDS) with the same instrument and in the same session as carrying out the BSE imaging. Such calibration enables direct comparison of the grey scales in the BSE images with the quantitative data using WDS in the Electron Microprobe. Further down in the text, the referee agrees with us on this point. We have added a sentence to the Results that explains this to the readers.

Referee comment: I would like to also suggest to add a new schematic drawing for summarizing the biomineralization and shell formation mechanisms obtained by this study. SEM and EBSD pictures are of course very nice, but they are sometimes too complicated for readers. A simplified drawing will be very helpful for readers to grasp the main conclusion of this MS.

Answer: We have added a schematic to the revised version of the ms as well as a new supplementary figure.

Referee comment: The authors not only examined the pulsed Sr-labelled portion of the shell, but also examined the shell comprehensively, so I recommend changing the title."

Answer: We believe the current title already reflects the 'comprehensive examination of the shell' as stated by the referee. The purpose of the label is to provide markers in time for the study of the entire shell. We will however reorganise and add the term 'shell' to the title to specify the aim of the study. We edited the title to be: '**Insights into architecture, growth dynamics, and biomineralization from pulsed Sr-labelled Katelysia rhytiphora shells (Mollusca, Bivalvia)**'.

Referee comment: P1, L24, L26-27, as mentioned above and below, the discussion of the element transportation is based on too weak evidence, so I recommend to deleting this part.

Answer: These sub structures were also observed in micro-Raman maps and reflect true changes in Sr concentration. We discuss this in more detail below (see referee comment P13, L13). However, we edited the abstract to make this point clearer to the reader.

Referee comment: P3, L31, More detailed information of labelled seawater circulation is necessary. Did the authors use a single batch of seawater, or prepare labelled seawater every time for changing the water? How robust was the stability of the Sr concentration? The seawater renewing was performed constantly or done at once? Because the authors did not provide seawater composition, the Sr fluctuation, if exist, is suspicious. Changes in Sr/Ca ratio in seawater can easily produce Sr/Ca fluctuation in the shell. This is very important and critical for the discussion for the elemental transport mechanism.

Answer: For labelling, seawater was enriched in Sr by adding 4.380 g Sr-hexachloride per 10 l of seawater and was freshly prepared each time the water had to be renewed. Renewing the water was done at the start and in the middle of each labelling period. As we used natural seawater from two large seawater storage tanks of 10,000 ltr capacity each and a high precision balance, precise to the third digit, we consider the Sr concentration data robust. We have added this to the materials and methods section.

Referee comment: P12, L1-17, I would suggest adding simulation data of Young's stiffness for two test cases, (1) Single aragonitic crystal, and (2) The same crystal arrangement, but have a random orientation of the crystals. Is it possible? The comparison between (1) and (2) will provide the contribution of complex 3D construction of multi-order unit of crystal arrangement, and that of between (2) and the results presented in the MS will provide a contribution of control of crystal orientation by bivalve, is this right? I am not familiar with the stiffness simulation, so I am not completely sure that this suggestion is pointing or not.

Answer: We have now included the Young's modulus for a single crystal. We have also included a reference to an earlier publication from our group that shows the Young's modulus for an aragonite single crystal as reference (Agbaje et al 2017). However, depicting a randomly oriented fabric in a pole figure means that the aragonite crystallographic axes will be randomly oriented. Therefore, the elastic properties of the crystal would be averaged and the fabric would be isotropic. A pole figure depicting an isotropic orientation would show an even distribution across the entire pole figure and would therefore be very uninformative to the reader. We added a sentence describing that a sample with random crystal orientation would lead to isotropic results.

*References for this answer:*

O.B.A. Agbaje, R. Wirth, L.F.G. Morales, K. Shirai, T. Watanabe, M. Kosnik, D.E. Jacob (2017). Architecture of crossed-lamellar bivalve shells: The Southern Giant Clam (*Tirana dears*, Roding, 1798). R. Soc. Open Sci. 4: 170622. http://dx.doi.org/10.1098/rsos.170622

Referee comment: P13, L13, the "bright grey areas" must not be caused "by variation in Sr concentration". It is OK to say that the contrast between labelled and non-labelled part is caused by the Sr concentration changes, because this is validated by Sr/Ca analysis by NanoSIMS and EPMA. However, the variation within the labelled portion was not be assured. Can you see this fluctuation also in Sr/Ca map? The contrast of BSE image is not only induced by Sr concentration but also by density (mass number) and topography. As the authors discussed, organic concentration can even change the contrast of BSE. If the authors want to discuss Sr concentration variation, they should be based on Sr analysis, not on BSE image. According to this, the evidence for the discussion at P13, L19-23 relies on very weak evidence. Additionally, the authors did not provide Sr and Ca composition of seawater, so it is difficult to exclude the possibility that this variation is attributed to the changes in seawater composition.

Answer: The resolution of the Sr/Ca maps obtained by NanoSIMS unfortunately do not allow to observe any variation at this spatial scale. As argued above, the Sr and Ca composition of the water, particularly during the labelling periods, is constant within analytical uncertainty and can thus be excluded as a source of grey scale variability in BSE. Neither did we observe growth irregularities (e.g. organic components) in these shell layers. Topography and edge effects would not result in such regular patterns of grey scales only within this very sharply defined layer as observed here. The fine grey banding, however, also shows up in the Raman maps (e.g. Figs.3, S2) and as Raman is not sensitive to electron density effects this would exclude these as a cause for the banding. Furthermore, deconvolution of the Raman signal is consistent with variation in Sr concentration as underlying cause for the grayscale banding observed in BSE, as increased Sr concentration in aragonite results in peak broadening (Fig. 3) and peak shift of the main carbonate band to lower wavenumbers (Fig. S2) (Alia et al., 1997, O'Donnell et al., 2008, Ruschel et al., 2012). This is direct evidence for the correlation of lighter grey scales in BSE with higher Sr concentrations in the aragonitic shell. Upon revision of the manuscript we will clarify this connection between BSE and Raman analysis more.

*References for this answer:*

Alia, J. M., Mera, Y. D. de, Edwards, H. G.M., Martín, P. G., and Andres, S. L.: FT-Raman and infrared spectroscopic study of aragonite-strontianite (CaxSr1− xCO3) solid solution, Spectrochimica Acta Part A: Molecular and Biomolecular Spectroscopy, 53, 2347–2362, 1997

O'Donnell, M. D., Fredholm, Y., Rouffignac, A. de, and Hill, R. G.: Structural analysis of a series of strontium-substituted apatites, Acta Biomaterialia, 4, 1455–1464, 2008.

Ruschel, K., Nasdala, L., Kronz, A., Hanchar, J. M., Többens, D. M., Škoda, R., Finger, F., and Möller, A.: A Raman spectroscopic study on the structural disorder of monazite–(Ce), Mineralogy and Petrology, 105, 41–55, 2012.

Referee comment: P14, L8-29, "4.4 Revisiting the Concept of Ion Transport Pathways". I recommend omitting this section because this section seems to be based on very weak evidence as mentioned above comments. In addition to the unreliability of BSE as Sr indicator, similar "enrichment factors for Ca and Sr (Ca-shell/Ca-seawater and Sr-shell/Sr-seawater" is not an appropriate parameter for discussing the elemental fractionation. This should be discussed by distribution coefficient (Sr/Ca-shell)/(Sr/Ca-seawater). Judging from the data in Table3, the data does not seem to satisfy enough robustness for discussing this topic. The authors also ignore fractionation between EPF (if exist) and carbonate. This can also produce low Sr/Ca ratio in the shell, without changing the EPF composition. No evidence was also presented for justifying the ACC formation obtained in this study. So, overall this section is not supported by the original data, thus should be omitted.

Answer: After reflecting on both reviewers' comments that this section is very speculative, we felt it was best to cut this section considerably and to focus only on discussing the distribution coefficients (Sr/Ca(shell)/Sr/Ca(seawater)) which are now presented in Table 3. Including possible fractionation by any potential EPF is extremely speculative and is therefore not warranted, however, we reflected the possibility of the potential fractionation by EPF (if existing) and toned down the assertive tone of this section.

Referee comment: P16, L1-6, Conclusion. The second conclusion is OK, but the first and third conclusions were not supported by the data presented in this MS, because of the reasons as mentioned above.

Answer: We have listed four conclusions and believe the referee is referring to conclusion two and four as being too speculative. We have re-written and modified these parts of the conclusion also with regards to referee 1's comments.

Referee comment: Minor comments: P2, L5-10, Organic macromolecules itself can also control trace element incorporation. See, Stephenson A. E., DeYoreo J. J., Wu L., Wu K. J., Hoyer J. and Dove P. M. (2008) Peptides enhance magnesium signature in calcite: insights into origins of vital effects. Science 322, 724– 727 Wang D. B., Wallace A. F., De Yoreo J. J. and Dove P. M. (2009) Carboxylated molecules regulate magnesium content of amorphous calcium carbonates during calcification. Proc. Natl. Acad. Sci. U.S.A. 106, 21511– 21516.

Answer: Thank you for pointing out the omission of these important works that we have now cited in the new version of the ms.

Referee comment: P4, L14, Magnification is not necessary, because it will be ultimately depends on print or screen size.

Answer: Thank you, mention of the magnification factor has been omitted in the revised version.

Referee comment: P4, L26, What is "Phenom XL"? P5, L27, "DREMEL tool" is not adequate. Maybe you should provide information of producer company, or use general name?

Answer: Phenom XL is the product name of the SEM used in this study. Similarly, a DREMEL tool is the official name of this tool. The term SEM is mentioned in the same sentence with Phenom XL. We added the term 'power tool' after 'DREMEL' to make this clearer.

[revised manuscript text omitted]